# PTENα functions as an immune suppressor and promotes immune resistance in *PTEN*-mutant cancer

Yizhe Sun [1,4], Dan Lu [1,4✉], Yue Yin[1], Jia Song[1], Yang Liu[1], Wenyan Hao[1], Fang Qi[1], Guangze Zhang[1], Xin Zhang[1], Liang Liu[1], Zhiqiang Lin[1], Hui Liang[1], Xuyang Zhao[1], Yan Jin[1] & Yuxin Yin [1,2,3✉]

*PTEN* is frequently mutated in human cancers and *PTEN* mutants promote tumor progression and metastasis. *PTEN* mutations have been implicated in immune regulation, however, the underlying mechanism is largely unknown. Here, we report that PTENα, the isoform of PTEN, remains active in cancer bearing stop-gained *PTEN* mutations. Through counteraction of CD8[+] T cell-mediated cytotoxicity, PTENα leads to T cell dysfunction and accelerates immune-resistant cancer progression. Clinical analysis further uncovers that PTENα-active mutations suppress host immune responses and result in poor prognosis in cancer as relative to PTENα-inactive mutations. Furthermore, germline deletion of *Ptenα* in mice increases cell susceptibility to immune attack through augmenting stress granule formation and limiting synthesis of peroxidases, leading to massive oxidative cell death and severe inflammatory damage. We propose that PTENα protects tumor from T cell killing and thus PTENα is a potential target in antitumor immunotherapy.

---

[1] Institute of Systems Biomedicine, Department of Pathology, School of Basic Medical Sciences, Beijing Key Laboratory of Tumor Systems Biology, Peking University Health Science Center, Beijing, China. [2] Peking-Tsinghua Center for Life Sciences, Peking University Health Science Center, Beijing, China. [3] Institute of Precision Medicine, Peking University Shenzhen Hospital, Shenzhen, China. [4] These authors contributed equally: Yizhe Sun, Dan Lu. ✉email: taotao@bjmu.edu.cn; yinyuxin@hsc.pku.edu.cn

The importance of *phosphatase and tensin homolog deleted on chromosome ten* (*PTEN*) in tumorigenesis is underscored by its frequent mutations, including missense, nonsense, frameshift, and large deletions[1,2]. The tumor-suppressive activity of PTEN largely depends on its lipid phosphatase activity, which antagonizes PI3K/AKT activation[1]. Accordingly, a significant fraction of PTEN missense mutations is identified in human cancers, which selectively abolish its phosphatase activity[3]. Additionally, more than half of PTEN mutations result in protein truncation and instability, which are functionally comparable to *PTEN* loss[1,4]. In spite of the dysregulation of PI3K-AKT signaling in cancers with *PTEN* truncated or missense mutations, the outcomes of tumors with different types of *PTEN* mutations are distinct[5,6]. Therefore, an in-depth understanding of the biological functions of various PTEN mutations in tumorigenesis is still in need, and crucial for cancer treatment.

Through restoring or enhancing the effector function of CD8$^+$ tumor-infiltrating T cells, immunotherapy has revolutionized the treatment of patients with advanced-stage cancers[7]. However, this therapy is limited to a minority of patients with cancer, and the clinical efficacy is often compromised along with tumor development[7,8]. Previous studies have revealed that tumor cells can hijack strategies developed to limit immune responses, thereby affecting priming, activation, and recruitment of T cells, causing tumor immune escape[9]. Apart from these processes of T cells, effective anti-tumor immunity also requires cytotoxic T lymphocytes (CTLs)-mediated cell killing[10]. Researches have shown that the function of CTLs depends on the ROS production in the target cells, and recent studies further uncover that CTLs cause different types of cell death depending on the context of the target cells[11–13]. These phenomena lead to the hypothesis that, instead of affecting T cell function, tumor-intrinsic factors may directly affect the susceptibility of tumor cells to CTLs. To date, researches on tumor immune escape still mainly focus on the effects of tumor cells on T cells, while little is known about whether tumor cells can escape from CTLs-mediated killing through modulating their own metabolic state.

PTENα (also known as PTEN-L) is the first identified isoform of PTEN, which contains an additional disordered 173-amino acid N terminus[14,15]. This region confers functions of PTENα that are distinct from those of PTEN[14,15]. The role of PTENα in tumors is complicated, which limits tumor growth in glioblastoma cells, while promotes carcinogenesis in liver cancer[14,16]. Up to now, the effects of PTENα in anti-tumor immunity remain elusive. In this study, we find that PTENα acts as an immune suppressor that restricts CD8$^+$ T cell response against *PTEN* mutant cancers, consequently resulting in poor outcome of the disease. Rather than directly affecting T cell priming or differentiation, PTENα promotes the cell resistance to T cell-mediated cytotoxicity and results in tumor immune escape. Reciprocally, loss of PTENα down-regulates translation of proteins associated with the oxidation–reduction process, and potentiates ferroptosis upon incubation with cytotoxic T cells, exacerbating inflammatory responses and promoting immune clearance. Our data thus demonstrate that PTENα acts as a suppressor of T cell cytotoxicity-induced oxidative cell death, and highlight the importance of targeting PTENα in antitumor immune therapy.

## Results

### PTENα remains active in *PTEN*-mutant cancer.
Loss-of-function mutations in the *PTEN* gene are frequent in human cancer[3]. In addition to phosphatase-inactive mutations, abundant stop-gained mutations were detected in exons encoding the C-terminal domain of PTEN (Fig. 1a). Through induction of

PTEN instability, stop-gained mutations restrict the function of phosphatase PTEN, whose effects are largely identical to those of phosphatase-inactive mutations[1]. However, through analyzing the survival of patients with tumor-bearing *PTEN* mutations, we found that the outcome of patients with stop-gained mutations was poorer than those with phosphatase-inactive mutations (Fig. 1b and supplementary Fig. 1a). We next analyzed differentially expressed genes in cancers with these two-type mutations based on the matched diagnosis age (40–60 years), aneuploidy score (<3), and neoplasm histologic grade (G3). Utilizing gene set enrichment analysis (GSEA), we found that genes related to adaptive immune response were selectively enriched in cancers with phosphatase-inactive mutations as compared with those with stop-gained mutations (Fig. 1c). Similar results were also detected between the two-type mutations in a broader context (Fig. 1d). These results lead to the hypothesis that, aside from tumor-promoting effects, stop-gained mutations of *PTEN* restrict host immune response against cancer.

As an isoform of PTEN, PTENα is initially translated from a CUG codon upstream of and in-frame with the coding region of canonical PTEN[15]. As shown in Supplementary Fig. 1b, phosphatase-inactive mutations impaired the phosphatase activity of PTEN and its isoform PTENα. To assess the effects of stop-gained mutations on PTEN isoforms, we analyzed the protein stability of the PTEN family with or without stop-gained mutations. As shown in Fig. 1e, f and Supplementary Fig. 1c, in contrast to the identical half-life between wild-type PTEN and PTENα, stop-gained mutations accelerated the degradation of PTEN rather than PTENα. To further confirm this result, we utilized the Jurkat and MOLT4 cells, which have been reported to carry stop-gained *PTEN* mutations[17]. In accordance with the bioinformatics data, we found that MOLT4 cells contain PTEN$^{K267Rfs*9}$ mutation and Jurkat cells contain PTEN$^{R234Afs*1}$ and PTEN$^{L247*}$ mutations by DNA-sequencing analysis (Supplementary Fig. 1d). We then employed an anti-PTEN antibody to assess the status of endogenous PTEN and PTENα in Jurkat and MOLT4 cells. As shown in Supplementary Fig. 1d, compared with wild-type PTEN and PTENα in HEK-293T, truncated PTENα rather than PTEN can be detected in both Jurkat and MOLT4 cells. Our data thus indicate that PTENα remains active in cancer with stop-gained *PTEN* mutations (Supplementary Fig. 1b).

### PTENα promotes tumor immune escape.
To investigate the role of PTENα in tumorigenesis, we used CRISPR/Cas9 technology to knock out the endogenous PTEN in the B16-F10 cell line, which also abolished the expression of PTENα, and subsequently re-enforced the expression of PTENα in *Pten$^{-/-}$* cell line. As shown in Supplementary Fig. 2a–c, ectopic expression of PTENα elicited little effects on cell proliferation in vitro. In light of the impaired immune response in cancer with stop-gained mutations, we thus employed irradiated B16-F10 cell line as tumor vaccine to assess the role of PTENα in modulation of host immune response against cancer (Fig. 2a). As shown in Fig. 2b, in contrast to the identical volume of tumor with or without PTENα in unimmunized C57BL/6 mice, overexpression of PTENα weakened the effectiveness of cancer vaccination. Through analysis of tumor-infiltrating lymphocytes (TIL) by flow cytometry, less amounts of CD45$^+$ immune cells were detected in PTENα-expressing tumors as relative to control tumors (Fig. 2c).

To study the status of immune cells in PTENα-expressing tumors, we applied scRNA-seq methods to investigate CD45$^+$ immune cells isolated from PTENα-expressing or control tumors with two replicates. To remove batch-effect, data integration and batch-effect correction were applied to the data. Utilizing graph-based clustering to analyze the tumor-infiltrating immune cells,

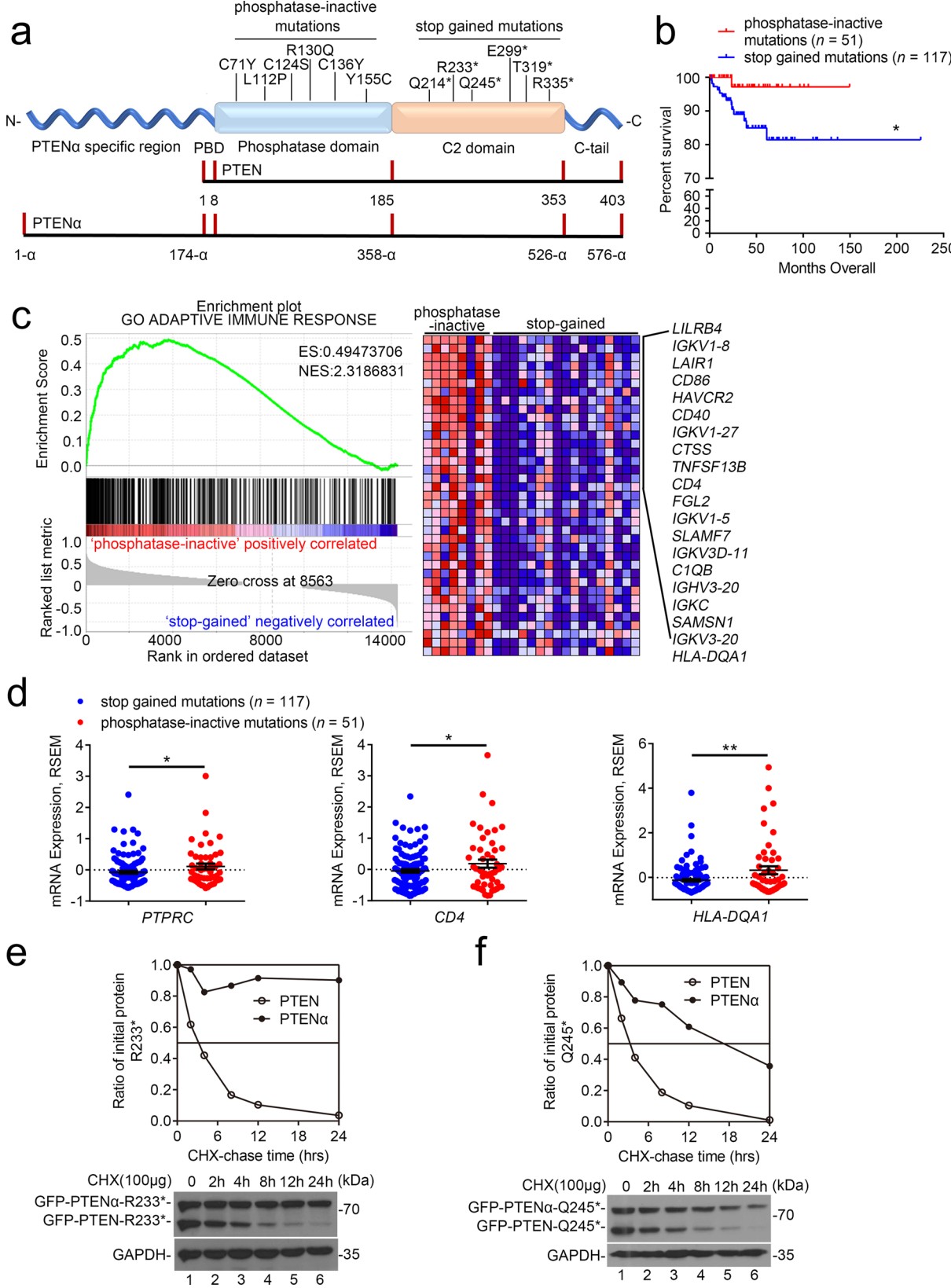

we identified 12 clusters of immune cells for 10× data. We then defined the clusters based on the exclusive expression of canonical marker genes. As shown in Supplementary Fig. 2d, e, and Supplementary Table 1, major immune cell types in cancers including CD4$^+$ T, CD8$^+$ T, Treg, NK, monocyte, macrophages, neutrophil, B, plasma B, plasmacytoid dendritic cells (pDCs), and conventional DCs (cDCs). Compared with more than 80% of lymphoid cells in the control tumor, the percentage of T

**Fig. 1 PTENα remains active in *PTEN*-mutant cancers. a** Cartoons of structures of PTEN and PTENα. Domains that are contained by PTEN and PTENα are shown in the simplified drawing, respectively. Stop gained mutations include all the nonsense or frameshift mutations terminated after H[185] (last amino acid of phosphatase tensin-type domain of PTEN). Phosphatase-inactive mutations are from the UniProt database. Some of the two kinds of mutations are indicated in the graph. **b** Survival curve of uterine corpus endometrial carcinoma (UCEC) patients with *PTEN* mutations (Stop gained mutations, $n = 117$; Phosphatase-inactive mutations, $n = 51$, *$P = 0.0377$). Clinical data were acquired from the TCGA database. **c** GSEA of RNA-seq data of UCEC patients with matched diagnosis age (40–60 years), aneuploidy score (<3), and neoplasm histologic grade (G3). The data were acquired from the TCGA database. ES is an abbreviation of Enrichment Score, and NES represents Normalized Enrichment Score. **d** Expression levels of immune surveillance-related genes in uterine corpus endometrial carcinoma (UCEC) patients with indicated *PTEN* mutations (Stop gained mutations, $n = 117$; Phosphatase-inactive mutations, $n = 51$, mean ± s.e.m., *$P$ (PTPRC) = 0.0400, *$P$ (CD4) = 0.0498, **$P = 0.0022$). **e** and **f** Immunoblot analysis of expression of GFP-tagged PTEN and PTENα carrying indicated mutations in HEK293T cells treated with 100 μg/ml CHX for indicated hours. An anti-GFP antibody was used. Gray values of PTEN and PTENα relative to GAPDH were determined, using for the line chart. Statistical significance was assessed by Log-rank (Mantel–Cox) test (**b**) or two-tailed unpaired Student's *t* test without multiple comparisons test (**d**). Data are representative of two (**e** and **f**) independent experiments with similar results. Source data are provided as a Source Data file. See also Supplementary Fig. 1.

lymphocytes and B lymphocytes were significantly reduced in PTENα-expressing tumors (Supplementary Fig. 2f). Considering the importance of CD8$^+$ T cells in host antitumor immunity, we performed unsupervised clustering and identified four clusters for CD8$^+$ T cells (Fig. 2d and Supplementary Fig. 3a). In contrast to the other three clusters, the fourth cluster of CD8$^+$ T cells, CD8T-Ctla4, expressed high levels of exhaustion markers *Clta4*, *Pdcd1*, and *Lag3*, thus representing exhausted CD8$^+$ T cells (Fig. 2e). As expected, the percentage of C4_CD8T-Ctla4 was remarkably increased in PTENα-expressing tumors as compared with control tumors (Fig. 2f, g). Consistent with scRNA-seq results, subsequent flow cytometry revealed that overexpression of PTENα limited the amounts of infiltrated lymphocytes, and enhanced PD-1 and TIM-3 expression on CD8$^+$ tumor-infiltrating T cells (supplementary Fig. 3b–f). Furthermore, we also measured the production of IFNγ in the tumor-infiltrating CD8$^+$ T cells in the treatment of PMA/ionomycin. As shown in Supplementary Fig. 3g, the production of IFNγ was weakened in CD8$^+$ T cells from PTENα-expressing tumors relative to those in control tumors.

To further confirm the immune regulatory role of PTENα in cancer, we performed the adoptive T-cell transfer assay. Mock or PTENα-expressing B16-*Pten*$^{-/-}$ cells were infected with LCMV-Cl13, and subcutaneously injected into nude mice. 10 days later, the activated CD8$^+$ T cells were harvested from spleen in mice infected with LCMV-Cl13 on day-7 post infection, and subsequently transferred into the tumor-bearing nude mice. Similar procedures were repeated on 14-day and 18-day in B16 tumor-bearing mice (Fig. 2h). As shown in Fig. 2i, viral infection hardly influenced tumor growth of Mock or PTENα-expressing B16 cells in nude mice, and ectopic expression of PTENα elicited no effects on cell proliferation, which is consistent to the data in C57BL/6 mice (Fig. 2i). Moreover, the transferred CD8$^+$ T cells exhibited significantly antitumor effects on Mock rather than PTENα-expressing B16 cells in nude mice (Fig. 2i). Notably, through analysis of CD8$^+$ T cells infiltrating in spleen or tumors, we found that the amount of adoptive transferred CD8$^+$ T cells were identical between the nude mice with PTENα-expressing B16 cells and those with Mock B16 cells (Supplementary Fig. 3h). Taken together, our data demonstrate that PTENα blocks T cell-mediated cancer eradication and promotes cancer immune escape.

Furthermore, we also investigated the effects of PTENα bearing stop-gained mutations in cancer immune escape. Analog to the immunosuppressive effects of wild-type PTENα, overexpression of PTENα$^{CTG-R233*}$ also reduced the effectiveness of tumor vaccine and facilitated tumor growth (Supplementary Fig. 3i). To ascertain whether the immunosuppressive role of PTENα is dependent on its secretion, we also employed the vector of PTENα$^{CTG-\Delta6A}$ that has been reported to abolish secretion of

PTENα[14]. Utilizing the cancer vaccine model, we found that overexpression of PTENα$^{CTG-\Delta6A}$ also promoted cancer immune escape, which is identical to the effects of PTENα$^{CTG-R233*}$, indicating that the secretion characteristics of PTENα hardly affects its immunoregulatory function (Supplementary Fig. 3i). Considering that PTENα mutations also exhibit clinical relevance in colon cancers, we employed CT26 cells (murine colorectal cancer cell line) to assess the role of PTENα in tumor vaccination model. We first used shRNA to knock down the endogenous PTEN expression in CT26 cells, and subsequently stably transfected with the vector encoding Mock or PTENα (Supplementary Fig. 3j). In accordance with the data in B16 cells, ectopic expression of PTENα elicited immunosuppressive effects and promoted tumor development in immunized mice (supplementary Fig. 3k). Together, our data thus indicate that the presence of PTENα in cancers subverts host immune attack.

**PTENα promotes T cell exhaustion in a T cell-extrinsic manner.** Since the presence of PTENα in cancers drives CD8$^+$ T cell dysfunction, we employed *Ptenα*$^{-/-}$ mice to further study the role of PTENα in the modulation of T cell fate. Although PTENα was expressed in both lymphoid and non-lymphoid organs (Supplementary Fig. 4a), *Ptenα*$^{-/-}$ mice developed normally, and no spontaneous tumor was detected within 2-year old. In order to determine whether loss of PTENα affects the proliferation, differentiation, and survival of T cells, we firstly employed a CFSE-staining assay to measure the T cell expansion during activation. As shown in Supplementary Fig. 4b, c, deletion of PTENα elicited little effects on T cell expansion during T cell priming. Besides, identical ratio of iTreg cell (CD4$^+$CD25$^+$Foxp3$^+$), Th1 (CD4$^+$IFNγ$^+$), Th17 (CD4$^+$IL-17A$^+$) and Tr1 (CD4$^+$IL-10$^+$) cells were induced under various T-cell polarizing conditions between wild-type and *Ptenα*$^{-/-}$ mice (Supplementary Fig. 4d). Finally, we used Annexin V/7-AAD staining to assess cell death during T cell activation. As shown in Supplementary Fig. 4e, no significant difference was detected between wild-type and *Ptenα*$^{-/-}$ T cells.

We next utilized lymphocytic choriomeningitis virus clone 13 (LCMV-Cl13) to induce T cell exhaustion. As shown in Fig. 3a, b, high viral loads were found in the lungs in wild-type mice, whereas *Ptenα*$^{-/-}$ mice showed a much-reduced viral burden on day 30 post infection. Compared with wild-type mice, the frequency of virus-specific CD8$^+$T cells (GP$_{33-41}$$^+$) was increased in the lungs from *Ptenα*$^{-/-}$ mice (Fig. 3c and Supplementary Fig. 5a). Consistent with published studies[18], we observed that high levels of inhibitory receptors including PD-1, TIM-3, and LAG-3 were expressed on the surface of virus-specific wild-type CD8$^+$T cells rather than *Ptenα*$^{-/-}$ CD8$^+$T cells during chronic infection (Fig. 3d). Moreover, we stimulated CD8$^+$GP$_{33-41}$$^+$T cells with virus-specific peptides to induce the production of effector cytokines in wild-type and *Ptenα*$^{-/-}$ T cells. As shown in

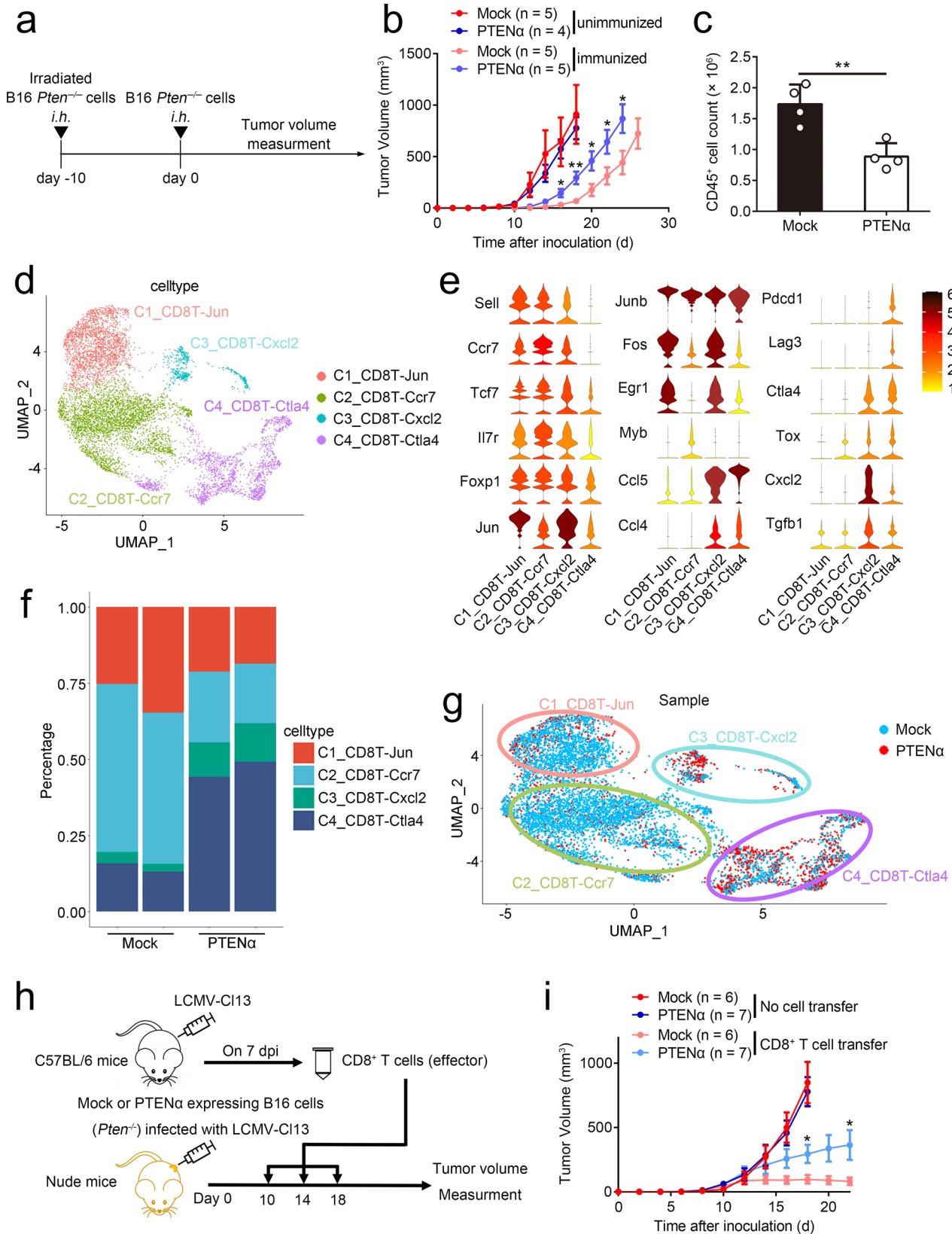

Supplementary Fig. 5b, $Ptenα^{-/-}$ T cells produced greater amounts of effector cytokines than wild-type T cells upon exposure to viral peptide. Similar results were also observed in virus-specific T cells from spleens (Supplementary Fig. 5c–e). These data thus indicate that deletion of PTENα blocks T cell dysfunction.

We next sought to determine whether T cell polarization is dependent on T cell-intrinsic PTENα. To this end, we used a bone marrow (BM) transplantation model in which $Ptenα^{-/-}$ or wild-type mice were lethally irradiated and transplanted with wild-type or $Ptenα^{-/-}$ BM and subsequently infected with LCMV-Cl13. As shown in Fig. 3e, on day 30 post-infection, the numbers of virus-

**Fig. 2 PTENα promotes tumor immune escape. a–g** Cancer vaccination model of Mock or PTENα transfected $Pten^{-/-}$ B16 cells. **a** Schematic diagram of the cancer vaccine model. **b** Tumor volumes of C57BL/6 mice subjected to the cancer vaccination model were monitored overtime. The unimmunized group refers to mice without irradiated B16 cells immunization. The statistical significances between the immunized groups were indicated (PTENα-unimmunized, $n = 4$ mice; other groups, $n = 5$ mice, mean ± s.e.m., *$P$ (d16) = 0.0254, *$P$ (d20) = 0.0339, *$P$ (d22) = 0.0487, *$P$ (d24) = 0.0450, **$P = 0.0077$). **c** The tumor-infiltrating lymphocytes (TILs) were isolated when the tumor volumes of the Mock-immunized group reach 200 mm³, and the cells were subjected to flow cytometry analysis. The cell counts of CD45$^+$ cells were used for statistical analysis ($n = 4$ mice, mean ± SD, **$P = 0.0044$). Gating strategies were shown in Supplementary Fig. 3b. **d–g** TILs were isolated when the tumor volumes of Mock-immunized group reach 200 mm³, and CD45$^+$ cells were sorted, using for 10× scRNA-seq with two replicates. 25,098 cells were identified as 12 clusters utilizing graph-based clustering. **d** Cd8T cells were further analyzed and identified as 4 clusters utilizing graph-based clustering. **e** Violin plot of differentiated genes in the 4 clusters of Cd8T cells. The color of each violin is the average expression value of a given gene. **f** and **g** Umap and proportion of the 4 clusters of Cd8T cells were shown. **h** and **i** Adoptive T cell transfer model of Mock or PTENα expressing B16-$Pten^{-/-}$ cells. **h** An illustration of mice treatment. **i** Tumor volumes was monitored over time (Mock, $n = 6$ mice; PTENα, $n = 7$ mice, mean ± s.e.m., *$P$ (d18) = 0.0360, *$P$ (d22) = 0.0481). Statistical significances between CD8$^+$ T cells transferred groups were shown. Statistical significance was assessed by a two-tailed unpaired Student's $t$ test without multiple comparisons test (**b**, **c**, **i**). Data are representative of two (**b** and **i**) or three (**c**) independent experiments with similar results. Source data are provided as a Source Data file. See also Supplementary Figs. 2 and 3.

specific T cells were identical between $Ptenα^{-/-}$ recipients of wild-type BM and those of $Ptenα^{-/-}$ BM. Of note, compared with wild-type donor mice, greater amounts of virus-specific T cells were detected in the two types of $Ptenα^{-/-}$ recipient mice (Fig. 3e). Conversely, LCMV-Cl13 infection resulted in lower amounts of virus-specific T cells in wild-type recipients, no matter which kind of BM is adoptive transferred (Fig. 3f). Consistent results were also detected by assessment of inhibitory receptor expression and effector cytokine production (Supplementary Fig. 6a–d). Combined with the data that PTENα is also expressed in non-lymphoid organs (Supplementary Fig. 4a), our data thus demonstrate that T cell-extrinsic PTENα drives T cell exhaustion.

**PTENα promotes cell resistance to T cell cytotoxicity**. To investigate the mechanism by which PTENα promotes T cell exhaustion, we analyze the epithelial–immune cell interaction. As shown in Fig. 4a, loss of PTENα triggered stronger inflammatory response in the early stage of higher titers of LCMV-Cl13 infection, causing massive congestion and edema in the lungs of $Ptenα^{-/-}$ mice upon viral infection. Moreover, greater amounts of death cells were detected in $Ptenα^{-/-}$ lungs assessed by TUNEL staining (Fig. 4b). Besides, we performed mass spectrometry analysis of tissue interstitial fluid (TIF) in the lungs during inflammation, and found that a series of nuclear proteins were released in extracellular matrix of lungs from $Ptenα^{-/-}$ mice at the early stage of viral infection, which acts as danger-associated molecular patterns (DAMPs) and activates immune response (Fig. 4c). Consistently, loss of PTENα increased the amounts of HSP70, HSP90, HMGB1, and eATP (extracellular ATP) in TIF in the lung during viral infection (Figs. 4d, e). These results thus indicate that deletion of PTENα promotes immunogenic cell death upon LCMV infection.

Since LCMV-Cl13 is a non-cytolytic virus that hardly induces cell death in vitro[19], we hypothesized that T cell-mediated cytotoxicity mainly contributed to the severe cell death. To this end, we performed in vitro killing assay to investigate whether PTENα affects target cell responses to CTLs. We pulsed Mock or PTENα expressing B16-$Pten^{-/-}$ cells with LCMV-GP$_{33-41}$ peptide and incubated with CD8$^+$ T cells isolated from spleens of C57BL/6 mice untreated or infected with LCMV-Cl13. Notably, incubation of CD8$^+$ T cells from the infected mice induced significantly lower cell lysis rates in PTENα-expressing B16 cells relative to those control cells (Fig. 4f). Moreover, compared with PTENα-expressing cells, the T cell-mediated cytotoxicity of Mock B16-$Pten^{-/-}$ cells induced high level of ATP release and intracellular ROS accumulation, which indicated that loss of PTENα resulted in non-apoptotic cell death under immune attack (Fig. 4g, h). Similar results were also detected in $Ptenα^{+/+}$ and $Ptenα^{-/-}$ MEFs (Supplementary Fig. 7a–c). Our

data thus demonstrate that CTLs triggers immunogenic death of $Ptenα^{-/-}$ cells that exacerbates inflammatory damage upon exposure to pathogens. The existence of PTENα correspondingly limits antigen clearance, resulting in T cell exhaustion.

**PTENα restricts oxidative cell death**. Previous studies have revealed that ROS is essential for CTLs-mediated cytotoxicity[11]. In consideration of the higher ROS level in PTENα-deletion cells induced by CTLs, we treated $Ptenα^{+/+}$ and $Ptenα^{-/-}$ MEFs with hydrogen peroxide ($H_2O_2$) to mimic CTLs stimulation and investigated the nature of cell death caused by PTENα deficiency. As shown in Supplementary Fig. 8a, b, PTENα deficiency markedly increased cell susceptibility to ROS-mediated cell death. Moreover, using a series of cell death inhibitors, we found that the ferroptosis inhibitor ferrostatin-1 (Fer-1) was the only one completely abolishing the cell death induced by $H_2O_2$ (Supplementary Fig. 8c). Consistent results were also observed in Mock or PTENα-expressing $Pten^{-/-}$ B16 cells (Fig. 5a–c). Besides, treatment of iron chelator deferomamine (DFO) and α-TOC (a most abundant form of vitamin E) significantly rescued $Ptenα^{-/-}$ MEFs from $H_2O_2$-induced cell death (Supplementary Fig. 8d, e). Reciprocally, supplementation of the ferroptosis activator erastin and RSL3 significantly promoted cell death compared with other cell death agonists did (Supplementary Fig. 8f, g). To further confirm that loss of PTENα selectively induces ferroptosis, we measured the status of GPX4, which is a key regulator of ferroptosis. As shown in Supplementary Fig. 8h, the protein levels of GPX4 were reduced in lungs from $Ptenα^{-/-}$ mice compared with wild-type mice upon exposure to the virus.

In order to ascertain that suppression of ferroptosis mainly contributes to the oncogenic function of PTENα, we used α-TOC which is a radical scavenger and lipid peroxidation inhibitor, to suppress ferroptosis commitment. Considering only a portion of α-TOC dosage can reach the tumor by oral administration and its water-insoluble characteristic, we exploited a molecular-matched strategy to prepare α-TOC-loading TPGS nanoparticles (NP-VE) with extremely high drug loading levels (up to 10 mg/ml)[20], which can effectively limit ROS production (Supplementary Fig. 8i, j). In vitro assays showed that α-TOC treatment weakened the difference of cell death between PTENα-expressing cells and controls cells exposure to cytotoxic T cells (Fig. 5d). Besides, we also used BODIPY-C11 dye to analyze the level of lipid ROS. Analog to the intracellular ROS level, the presence of PTENα limited the level of lipid ROS in B16 cells ($Pten^{-/-}$) under T cell attack (Fig. 5e). Moreover, treatment of NP-VE in the cancer vaccine model promoted the progression of Mock-expressing B16-$Pten^{-/-}$ cells to a similar level of PTENα expressing B16-$Pten^{-/-}$ cells (Fig. 5f). Flow cytometry analysis of the infiltrating

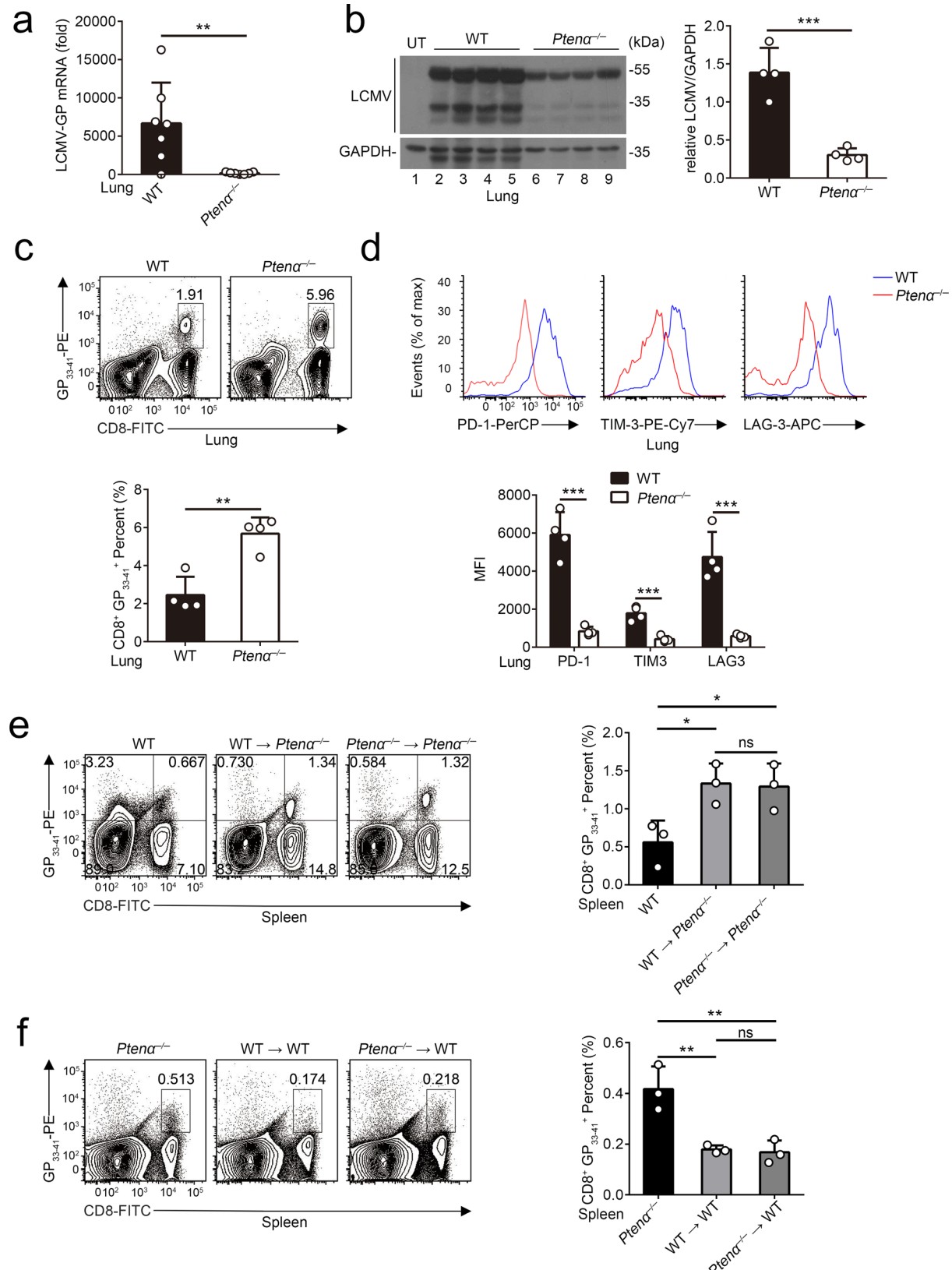

immune cells revealed that injection of NP-VE restricted the amount of CD45$^+$ immune cells infiltrating in tumors as compared with untreated Mock tumors (Supplementary Fig. 9a). Besides, the presence of PTENα exhibited little effects on the recruitment of immune cells in tumors treated with NP-VE (Supplementary Fig. 9a), suggesting that blockade of oxidative cell

death is essential for the immunosuppressive effects of PTENα. Further flow cytometry analysis demonstrated that NP-VE treatment or PTENα presence restricted both T and B cell recruitment in tumors (Supplementary Fig. 9b, c). In accordance with the suppressive effects of NP-VE treatment on T cell recruitment, increased expression of PD1 and reduced production

**Fig. 3 T cell-extrinsic PTENα promotes exhausted T cell formation. a–d** Wild-type and *Ptenα*^−/−^ mice were i.v. infected with $5 \times 10^5$ PFU LCMV-Cl13. On day 30 post-infection, the mice were sacrificed, and the lungs of the mice were harvested. **a, b** The mRNA and protein levels of LCMV-GP in the lungs were analyzed with qRT-PCR (**a**) (*n* = 7 mice, mean ± SD, **P = 0.0071) or immunoblot analysis (**b**) with anti-LCMV antibody, respectively. Gray values of LCMV proteins were determined, using for statistical analysis (**b**) (*n* = 4 mice, mean ± SD, ***P = 0.0007). UT untreatment. **c** Lymphocytes isolated from lungs of wild-type and *Ptenα*^−/−^ mice on day 30 post-infection were subjected to flow cytometry, and the frequencies of CD8^+^GP$_{33-41}$^+^ cells were analyzed (*n* = 4 mice, mean ± SD, **P = 0.0023). Gating strategies were shown in Supplementary Fig. 5a. **d** Flow cytometric analysis of the expression of PD-1, TIM-3 and LAG-3 on CD8^+^GP$_{33-41}$^+^ cells from lungs of wild-type and *Ptenα*^−/−^ mice on day 30 post-infection (*n* = 4 mice, mean ± SD, ***P (PD-1) = 0.0002, ***P (TIM-3) = 0.0005, ***P (LAG-3) = 0.0007). Gating strategies were shown in Supplementary Fig. 5a. **e** Bone marrow from wild-type and *Ptenα*^−/−^ mice were transplanted into *Ptenα*^−/−^ mice, respectively. 30 days after transplantation, mice were i.v. infected with $5 \times 10^5$ PFU LCMV-Cl13. On day 30 post-infection, spleens were harvested from the mice, and lymphocytes were isolated. Percentages of CD8^+^GP$_{33-41}$^+^ cells were determined by flow cytometry analysis (*n* = 3 mice, mean ± SD, ns not significant, *P (WT vs. WT → *Ptenα*^−/−^) = 0.0373, *P (WT vs. *Ptenα*^−/−^ → *Ptenα*^−/−^) = 0.0456). Gating strategies were shown in Supplementary Fig. 5a. Unstained refers to samples that are unstained with PD-1, TIM-3, or LAG3. **f** Wild-type mice were adoptive transferred with BM from wild-type or *Ptenα*^−/−^ mice, and infected with $5 \times 10^5$ PFU LCMV-Cl13 on day 30 post-transplantation. 30 days post-infection, splenocytes were harvested from the mice, and the percentages of CD8^+^GP$_{33-41}$^+^ cells were determined by flow cytometry analysis (*n* = 3 mice, mean ± SD, ns, not significant, **P (*Ptenα*^−/−^ vs. WT → WT) = 0.0061, **P (*Ptenα*^−/−^ vs. *Ptenα*^−/−^ → WT) = 0.0049). Gating strategies were shown in Supplementary Fig. 5a. Statistical significance was assessed by two-tailed unpaired Student's *t* test (**a–d**) or one-way ANOVA followed by Tukey's multiple comparisons test (**e** and **f**). Data are representative of two (**a–f**) independent experiments. Source data are provided as a Source Data file. See also Supplementary Figs. 4–6.

of IFNγ were also detected in CD8^+^ T cells isolated from mock tumors treated with NP-VE (Supplementary Fig. 9d, e). Taken together, our data demonstrate that PTENα increases cell resistance to CTLs-induced immunogenic cell death and consequently leads to cancer immune escape.

**PTENα blocks stress granules formation during oxidative stress.** To investigate the molecular mechanism by which PTENα modulates cell response to immune attack, we performed pull-down assays with FLAG-tagged proteins coupled with mass spectrometry under the condition of $H_2O_2$ treatment. Interestingly, we noticed that majority of PTENα-associated proteins were involved in stress granules assembly (G3BP1, G3BP2, etc.) and formation (eIF2α, PCBP1, PCBP2, ZC3HAV1, etc.) (Fig. 6a). Subsequent co-immunoprecipitation and immunofluorescence assays further confirmed the association of PTENα with stress granules components (Fig. 6b, c). To determine whether PTEN also participates in this process, we co-transfected PTEN or PTENα with eIF2α in the presence or absence of $H_2O_2$ treatment. Compared with the stimulatory effect of $H_2O_2$ treatment on the association of PTENα with eIF2α (Fig. 6d, lane 3–4 versus lane 1–2), PTEN did not interact with stress granules-associated proteins (Fig. 6d, lane 7–8 versus lane 5–6 and Fig. 6e, lane 3 versus lane 1 and 2).

To determine whether PTENα influences stress granules formation, we transfected mock or PTENα into *Pten*^−/−^ B16-cells followed by stimulating with $H_2O_2$, which is also a known inducer of stress granules. In spite of extensive co-localization of PTENα with intrinsic G3BP1 detected by immunofluorescence assay, overexpression of PTENα significantly inhibited $H_2O_2$-induced cytoplasmic foci formation compared with cells transfected with empty vectors (Fig. 6f, g). Together, these results indicate that PTENα selectively interacts with stress granules-associated proteins and blocks stress granules formation under stress conditions.

**PTENα maintains protein synthesis of peroxidases.** The phosphorylation of eIF2α is crucial for stress granules formation[21]. Given that PTENα possesses the canonical phosphatase activity and interacts with eIF2α, we hypothesized that PTENα modulates eIF2α phosphorylation under multiple kinds of stress conditions. As anticipated, stimulated by $H_2O_2$, overexpression of PTENα in *Pten*^−/−^ B16 cells inhibited eIF2α phosphorylation (Fig. 7a, lanes 4 and 6 versus lanes 3 and 5). Similar results were also observed in MEFs (Supplementary Fig. 10a, lanes 4 and 6 versus lanes 3 and 5).

Additionally, unlike PTENα, enforced expression of PTEN or the phosphatase-dead PTENα-C297S mutant did not significantly influence the eIF2α phosphorylation status in *Pten*^−/−^ MEFs upon $H_2O_2$ exposure (Supplementary Fig. 10a, lanes 4 and 6 versus lanes 3 and 5). Besides, we also performed in vitro phosphatase assay and found that PTENα directly dephosphorylates eIF2α in vitro (Fig. 7b, lane 3 versus lane 2). Our data thus confirm that PTENα acts as a phosphatase on eIF2α.

Since eIF2α phosphorylation restricts host translational machinery[21], we incubated *Ptenα*^+/+^ and *Ptenα*^−/−^ MEFs in medium containing ^35^S isotope-labeled methionine (^35^S-Met) and then measured ^35^S-labeled protein synthesis. Relative to *Ptenα*^+/+^ MEFs, a substantial decrease in protein synthesis was observed in *Ptenα*^−/−^ MEFs after exposure to $H_2O_2$ or TG (Fig. 7c). Consistent with these findings, puromycin incorporation (as a measure of protein synthesis) into nascent cells was also significantly decreased under stress conditions in *Ptenα*^−/−^ MEFs compared with *Ptenα*^+/+^ MEFs, as revealed by immunofluorescence and immunoblotting analysis (Supplementary Fig. 10b, lanes 4–6 versus lanes 1–3, and Supplementary Fig. 10c). Similar results were also detected in Mock or PTENα-expressing *Pten*^−/−^ B16 cells (Fig. 7d, e, lanes 4–6 versus lanes 1–3). Together, our data thus indicate that deletion of PTENα promotes eIF2α-mediated translation inhibition.

Stress granules are known to be related with selective translation[22–24]. To determine which proteins are regulated by eIF2α phosphorylation and consequent stress granules assembly in *Ptenα*^−/−^ MEFs, we performed a mass spectrometry analysis of *Ptenα*^+/+^ and *Ptenα*^−/−^ MEFs under oxidative stress. As shown in Fig. 7f and Supplementary Fig. 10d, loss of PTENα selectively affected the synthesis of proteins associated with translation and oxidation–reduction processes. Ensued immunoblotting analysis confirmed that protein level of GPX4 rather than its mRNA level was significantly decreased in PTENα-null cells under oxidative stress condition (Fig. 7g, lane 4 and lane 6 versus lane 3 and lane 5, Supplementary Fig. 10e, lane 4 and lane 6 versus lane 3 and lane 5, and Supplementary Fig. 10f). Accordingly, intracellular ROS levels in $H_2O_2$-treated PTENα-expressing B16 cells were diminished relative to control cells (Fig. 7h).

To further confirm the essential role of Gpx4 in PTENα-mediated tumor immune escape, we used shRNA to knock down the endogenous Gpx4 expression in *Pten*^−/−^ B16 cells, which was subsequently stable transfected with PTENα. Employing the murine tumor vaccination model, loss of Gpx4 neutralized the immunosuppressive effects of PTENα and remarkably increased

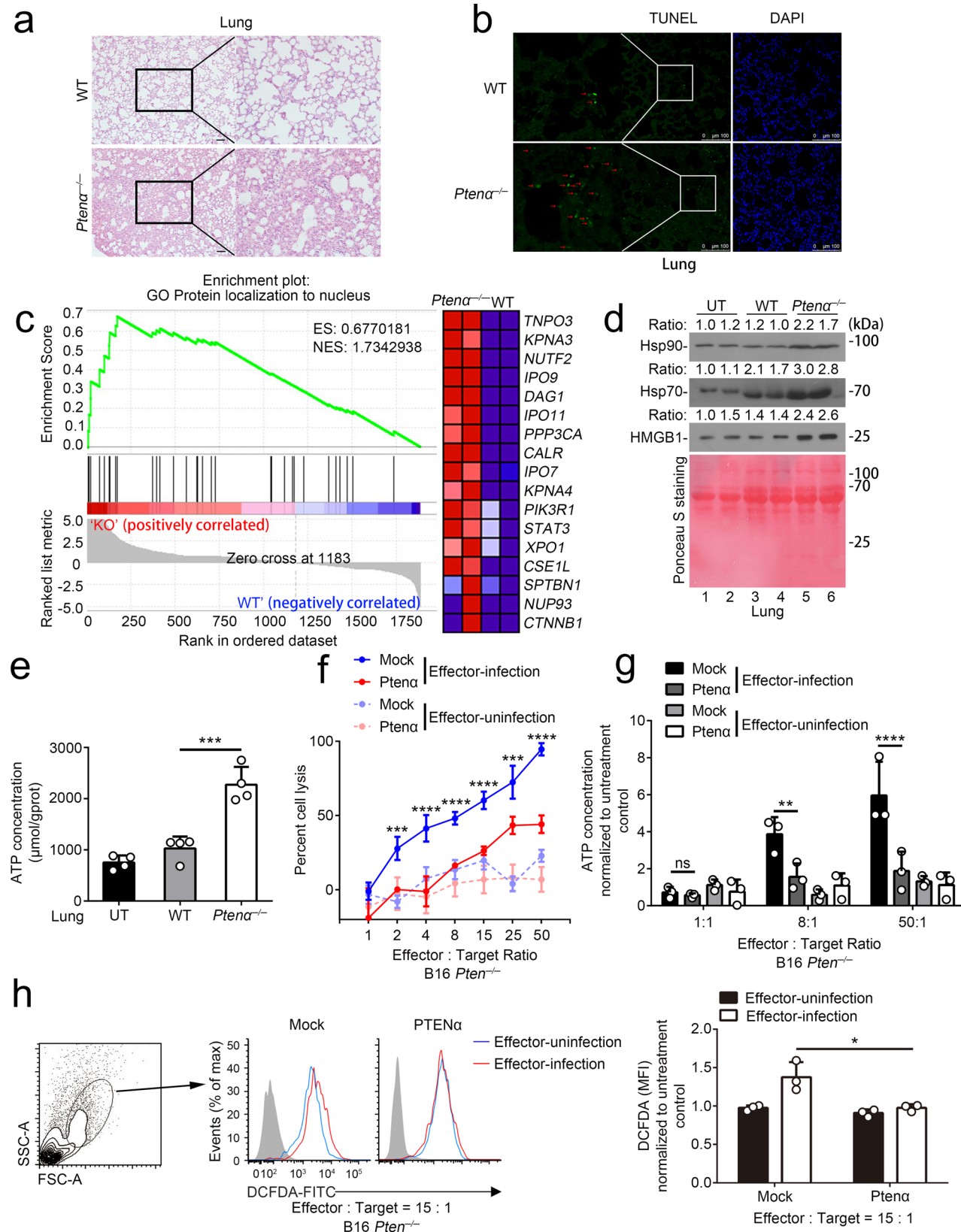

the effectiveness of tumor vaccination (Fig. 7i). Our data thus demonstrate that Gpx4 largely contributes to the stimulatory role of PTENα in cancer immune escape.

Collectively, we identify PTENα acts as phosphatase of eIF2α, which in turn promotes synthesis of peroxidases and blocks oxidative cell death, eventually ameliorating cancer immune evasion.

**Fig. 4 PTENα promotes cell resistance to T cell cytotoxicity. a–e** Wild-type and $Ptenα^{-/-}$ mice were i.v. infected with $1 \times 10^6$ PFU LCMV-Cl13. On day 7 post-infection, lungs were harvested from the mice. **a** The lungs were subjected to hematoxylin and eosin (H&E) staining. The image is representative of four mice with similar results. Scale bar: 200 μm. **b** TUNEL staining of lungs from wild-type and $Ptenα^{-/-}$ mice on day 7 post-infection. The image is representative of four mice with similar results. The red arrow indicates TUNEL positive cells. **c** The tissue interstitial fluids were isolated from the lungs and subjected to MS analysis. The differential proteins were analyzed using GSEA with GO gene sets. ES is an abbreviation of enrichment score, and NES represents normalized enrichment score. KO refers to $Ptenα^{-/-}$. Each sample is a mixture of fluids from three mice. **d** and **e** The interstitial fluids were isolated from the lungs, and used for immunoblot analysis with indicated antibodies (**d**), and ATP content assay (**e**) ($n = 4$ mixed samples from three mice, mean ± SD, ***$P = 0.0002$). For immunoblot analysis, equal amounts of loading proteins were showed by the Ponceau S staining. The relative ratios of Hsp70, Hsp90, and HMGB1 to Ponceau S-stained total proteins were indicated. UT, untreatment. **f–h** In vitro cytotoxicity assay of B16 cells. Mock or PTENα expressing $Pten^{-/-}$ B16 cells pulsed with $GP_{33-41}$ peptide were used as target cells. Effector cells and target cells were incubated at indicated ratios. Effector-uninfection refers to T cells from uninfected mice and acts as a negative control. **f** 20 h post-incubation, B16 cells were washed to remove lymphocytes and counted using CCK-8 ($n = 3$ cell cultures, mean ± SD, ***$P$ (2) $= 0.0006$, ***$P$ (25) $= 0.0003$, ****$P < 0.0001$). **g** Culture medium was collected 20 h post-incubation and centrifuged to remove cells and debris. The supernatant was subjected to ATP content assay ($n = 3$ cell cultures, mean ± SD, ns, not significant, **$P = 0.0078$, ****$P < 0.0001$). Untreatment control refers to pulsed B16 cells without effector incubation. **h** 12 h post-incubation, B16 cells were harvested and stained with DCFDA, using for flow cytometry analysis. Cells with higher intensity of FSC and SSC were gated and considered as B16 cells. Mean fluorescence intensity (MFI) of DCFDA was used for statistical analysis ($n = 3$ cell cultures, mean ± SD, *$P = 0.0255$). Untreatment control refers to pulsed B16 cells without effector incubation. Statistical significance was assessed by two-tailed unpaired Student's $t$ test (**h**) or one (**e**) or two (**f** and **g**)-way ANOVA followed by Tukey's multiple comparisons test. Data are representative of two (**a**, **b**, **d–h**) independent experiments. Source data are provided as a Source Data file. See also Supplementary Fig. 7.

## Discussion

Despite the wide application of immunotherapy in the treatment of cancers, only a subset of patients responds to the treatment, and the clinical efficacy is often compromised along with tumor development[10]. Notably, a recent study uncovers that *PTEN* mutations are frequently observed in immune-resistant cancers[25]. Although the combination of immunotherapies and PI3K-AKT pathway inhibitors elicits partially effective to reinvigorate tumor-infiltrating T cells[26], the immunosuppressive tumor environment can hardly be transformed, which suggests that other signaling pathways are also triggered in addition to PTEN dysfunction. Here we identify that presence of PTENα in *PTEN*-mutant cancers counteracts immune attack and promotes tumor immune escape (Supplementary Fig. 10 g). Rather than affecting T cell priming, tumor-intrinsic PTENα restricts T cell-mediated cytotoxicity through limiting ferroptosis, which attenuates immunogenic cell death and maintains immunosuppressive niche. Our data thus indicate that PTENα can be a potential target of immune therapy to regulate the susceptibility of tumor cells to immune responses.

As an N-terminally extended isoform of PTEN, PTENα translates through a CUG codon upstream of and in-frame with the coding region of canonical PTEN[15]. Accompanied with PTEN, the phosphatase-inactive mutations also impair the phosphatase activity of PTENα. However, we find that the stop-gained mutations, which result in PTEN instability and degradation, hardly affect PTENα stabilization. Notably, the presence of stop-gained mutations of PTEN is correlated with less T-cell infiltration and worse patient outcome, suggesting the oncogenic role of PTENα in tumor development. Interestingly, the role of PTENα in tumors seems complicated due to the opposite effects of PTENα on tumor growth in glioblastoma and liver cancer[14,16]. Although clinical analysis reveals that sustained expression of PTENα in liver cancer tissue is correlated with accelerated tumor progression, both types of research use the model of nude mice bearing tumor xenograft and fail to investigate the role of PTENα in anti-tumor immunity[14,16]. Herein, we identify the immunosuppressive function of PTENα, which attenuates cancer cell susceptibility to T cell-mediated cytotoxicity through limiting oxidative cell death, thereby promoting cancer escape from immune eradication.

Among the inhibitory effects of PTENα on host antitumor immune response, we regard its negative role in oxidative cell death as the core mechanism in vivo. First, the tumor vaccination model reveals that tumor-intrinsic rather than T cell-intrinsic PTENα mainly contributes to cancer immune evasion. Second, both in vitro killing assay and T cell adoptive transfer model demonstrate that reduction of immunogenic cell death and ensued DAMPs release is the cause of limited recruitment of immune cell. Third, chronic viral infection and BMT models demonstrate that reduced cell death and consequently persistent antigenic stimulation induced by PTENα drive exhausted T cell formation. Above all, our results demonstrated that, through protecting tumor cells from T cell killing, PTENα limits the infiltration of leukocytes in tumors and impairs T cell effector function, eventually leading to tumor immune escape.

The physiological function of PTENα remains largely unknown. Here we demonstrate that, as a phosphatase of eIF2α, PTENα maintains host protein synthesis, and limits stress granule formation. Stress granules have a great impact on cells' adaption to environmental changes through selective modulation of host protein synthesis under various stress conditions[23,24]. We found that PTENα restricts stress granule formation and maintains translation of proteins associated with oxidation–reduction processes, which in turn limits oxidative cell death and ameliorates the inflammatory response by blocking DAMPs release. In accordance with viral infection, it has been reported that *Pseudomonas aeruginosa* infection triggers severe immunopathology in mice without PTEN-L (PTENα)[27]. Our work thus supports the notion that the presence of PTENα is critical for host protection from pathogen-induced tissue damage. In *PTEN* mutant cancer, this process is hijacked, thereby promoting the adaption of tumor cells to T cell cytotoxicity, resulting in tumor immune escape.

Ferroptosis has been implicated in a variety of pathological disorders, such as neurodegenerative events, ischemia/reperfusion injury, and bacterial infection[28]. Besides, a previous study has revealed that CTLs can kill cancer cells by inducing ferroptotic cell death[12]. A recent study also uncovers that the sensitivity of tumor cells to oxidative stress and ferroptosis limits their metastasis through blood[29]. In this study, our results identify the inhibitory role of PTENα in the modulation of ferroptosis. Through neutralizing intracellular ROS, PTENα enhanced cancer cell resistance against T cell killing. Moreover, the presence of PTENα promoted cancer metastasis through blood. Therefore, these results further confirm the importance of PTENα in cancer immune escape and highlight the potential of targeting PTENα to be a promising way in cancer treatment.

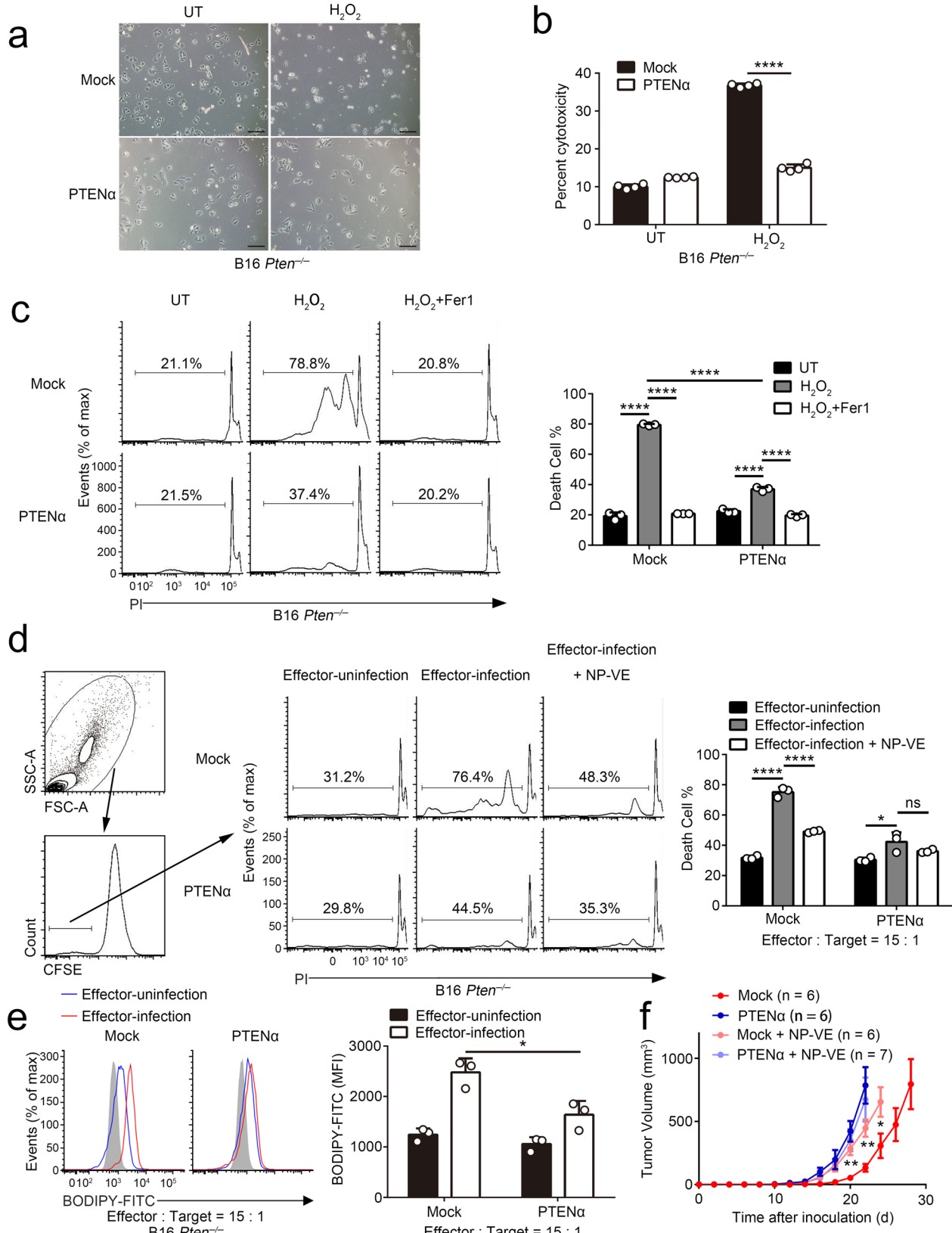

## Methods

**Mice.** *Ptenα⁻/⁻* mice (C57BL/6J background) were generated and reported in our previous study[30]. Balb/c and Balb/c nude mice were purchased from Charles River Laboratories. 6–8 weeks old male mice were used for the study. All animals were maintained under a specific pathogen-free condition at the Department of Laboratory Animal Science of Peking University Health Science Center and reared in standard conditions with controlled temperature (20–26 °C), humidity (40–70%),

and 12/12-hour dark/light cycle. The animal experimental protocols were approved by the Ethics Committee of Peking University Health Science Center, and the animal testing and research comply with all relevant ethical regulations.

**Cells.** B16-F10, HEK-293T, Molt4, Jurkat, CT26, and HeLa cells were from American Type Culture Collection (ATCC). MEFs were obtained through mincing

**Fig. 5 PTENα restricts oxidative cell death. a** Microscopy analysis of $H_2O_2$ (500 μM) treated Mock or PTENα-expressing B16-$Pten^{-/-}$ cells. UT untreatment. Scale bar: 100 μm. **b** Mock or PTENα-expressing B16-$Pten^{-/-}$ cells were treated with 500 μM $H_2O_2$ and the percent cytotoxicity was assessed using LDH release assay ($n = 4$ cell cultures, mean ± SD, ****$P < 0.0001$). UT untreatment. **c** Mock or PTENα-expressing B16-$Pten^{-/-}$ cells were treated with 500 μM $H_2O_2$ in the absence or presence of 1 μM Fer1. Cells were stained with propidium iodide (PI), followed by flow cytometric analysis. All cells were gated. The death cell rates were used for statistical analysis ($n = 3$ cell cultures, mean ± SD, ****$P < 0.0001$). UT untreatment. **d** and **e** In vitro cytotoxicity assay for B16 cells. Mock or PTENα-expressing B16-$Pten^{-/-}$ cells pulsed with GP$_{33-41}$ peptide were used as target cells. Effector cells and target cells were incubated at indicated ratios. Effector-uninfection refers to non-antigen-specific T cells and acts as a negative control. **d** Before incubation, CD8$^+$ T cells were stained with CFSE, then the effector cells and target cells were incubated in the presence or absence of 20 mM NP-VE. 20 h post-incubation, the cells were harvested and subjected to PI staining. Death cells in the CFSE$^-$ cells were assessed by flow cytometric analysis ($n = 3$ cell cultures, mean ± SD, ns not significant, *$P = 0.0188$, ****$P < 0.0001$). Gating strategies were shown. **e** 12 h post-incubation, cells were harvested and stained with BODIPY-C11, followed by flow cytometric analysis. The gating strategy was identical to that in Fig. 4h. MFI of the target cells was used for statistical analysis ($n = 3$ cell cultures, mean ± SD, *$P = 0.0207$). **f** Mock or PTENα transfected $Pten^{-/-}$ B16 cells were subjected to the cancer vaccination model and subcutaneously injected in the presence or absence of 50 μl of 5 M NP-VE. Tumor volumes were monitored over time, and the statistical significances between the Mock groups were indicated (PTENα + NP-VE, $n = 7$ mice; other groups, $n = 6$ mice, mean ± s.e.m., *$P = 0.0465$, **$P$ (d20) = 0.0015, **$P$ (d20) = 0.0019). Statistical significance was assessed by two-tailed unpaired Student's $t$ test (**b**, **e**, **f**) or one (**d**) or two (**c**)-way ANOVA followed by Tukey's multiple comparisons test. Data are representative of two (**a-f**) independent experiments. Source data are provided as a Source Data file. See also Supplementary Figs. 8 and 9.

---

13.5-day embryos, digesting with trypsin, and filtering through a 70 μm strainer. For the construction of B16-$Pten^{-/-}$ cell lines, the CRISPR-Cas9 system was used. In brief, the guide sequence "TGTGCATATTTATTGCATCG" targeting $Pten$ was cloning into the PX330 plasmid. The PX330 plasmid was co-expressed with Cas9 in the B16 cells and selected with G418. The cells were seeded in a 96-well plate at the density of 50 cells per well. After 2–3 weeks of expansion, the clones were harvested and examined with sequencing and western blot.

**Antibodies and reagents**. Following commercial antibodies were used in this study. clone number, catalog number and dilutions were shown in turn: anti-PTEN (138G6, 9559, 1:1000), anti-eIF2α (D7D3, 5324, 1:1000) and antibody to phosphorylated eIF2α (Ser51) (119A11, 3597, 1:1000) (all from CST); anti-GAPDH (1C4, KM9002T, 1:5000) and anti-GFP (9F6, KM8009, 1:5000) (both from Sungenebiotech); anti-FLAG (M2, F3165, 1:5000, Sigma-Aldrich); anti-α-tubulin (2F9, M175-3, 1:5000), H-2D$^b$ LCMV gp33 Tetramer-KAVYNFATC (TB-5002-1, 1:250) (both from MBL); anti-puromycin (17H1, MABE341, 1:1000, merck-millipore); anti-GPX4 (E-12, sc-166570, 1:1000) and anti-LCMV (M104, sc-57894, 1:100) (all from Santa Cruz); anti-Hsp90 (OTI4C10, TA-12, 1:1000, ZSGB-BIO); anti-N-PTEN (EPR23729-4, ab260011, 1:1000), anti-HMGB1 (EPR3507, ab79823, 1:1000) and anti-Hsp70 (EPR16892, ab181606, 1:1000) (all from abcam); anti-IL-17A (eBio17B7, 12-7177-81, 1:250), anti-IL-10 (JES5-16E3, 12-7101-41, 1:250), anti-Foxp3 (FJK-16s, 17-5773-82, 1:250), anti-PD-1 (J43, 46-9985-82, 1:250), anti-TIM3 (RMT3-23, 25-5870-82, 1:250), anti-LAG3 (C9B7W, 17-2231-82, 1:250), anti-CD45 (30-F11, 25-0451-81, 1:250), anti-CD25 (PC61.5, 12-0251-83, 1:250), antibody to IFN-γ (XMG1.2, 17-7311-82, 1:250) and antibody to TNF-α (MP6-XT22, 25-7321-82, 1:250) (all from eBioscience); Anti-CD4 (GK1.5, 35-0041-U500, 1:250) and anti-CD8 (53-6.7, 35-0081-U500 and 20-0081-U100, 1:250) (both from Tonbo); anti-G3BP1 (A14836, 1:1000, ABclonal); Anti-CD3ε (145-2C11, 100301, 2 μg/ml), anti-CD28 (37.51, 102101, 1 μg/ml), anti-IL-4 (11B11, 504101, 10 μg/ml) and anti-IFNγ (XMG1.2, 505801, 10 μg/ml) (all from BioLegend).

The reagents used in this study were as follow z-VAD-Fmk, Erastin, RSL3 and ferrostatin-1 (Fer-1), Selleck; $H_2O_2$, Cycloheximide (CHX), propidium iodide (PI), N-Acetyl-L-cysteine (NAC), Deferoxamine (DFO), and α-tocopherol (α-Toc) Sigma; H2DCFDA, GeneCopoeia; Thapsigargin (TG) and Necrostatin-1 (Nec-1), abcam; L-$^{35}$S-Methionine, PerkinElmer; puromycin, ACROS; Recombinant mouse TNF-α, Biolegend; Annexin V and 7-AAD, ebioscience; CFSE, Tonbo; BODIPY-C11, Invitrogen.

**Short hairpin RNA infection assay**. pLKO.1-puro was used to knock down gene expression. Sequence for $Gpx4$ shRNA interference was 5′-CCGGGGAGCCCAT TCCTGAACCTTTCTCGAGAAAGGTTCAGGAATGGGCTCCTTTTTG-3′, and sequence for $Pten$ shRNA interference was 5′- CCGGGGGTAAATACGTTCTT CATACCTCGAGGTATGAAGAACGTATTTACCCTTTTTG-3′.

**Cell proliferation assay**. Different numbers of cells were seeded into the 96-well plates by the gradient. Then the cells were subjected to a Cell counting Kit-8 (CCK-8, Dojindo) assay following the manufacturer's instructions to draw the stand curve. For assessment of cell proliferation, the B16 cells were seeded into a 96-well plate at the density of 1000 cells per well in 100 μl DMEM medium containing 10% (vol/vol) fetal bovine serum (FBS). The cells were cultured at 37 °C and measured using CCK-8 daily. The cell numbers were determined by comparing them with the stand curve.

**Colony formation assay**. B16 cells were seeded in a six-well plate at the density of 100 cells per well and cultured at 37 °C for 7 days. After fixing in methanol for 20 min and washing with PBS twice, the cells were stained with Crystal Violet

Staining Solution (Beyotime) at room temperature for 20 min, using for photographing.

**Melanoma mouse model**. For the cancer vaccination model, $1 \times 10^5$ B16-F10 cells were irradiated (40 Gy), and cultured at 37 °C overnight. Then the dying cells were collected and injected subcutaneously into the left flank of C57BL/6 mice. 10 days post-injection, $1 \times 10^6$ B16 cells were re-injected into the right flank of the mice through subcutaneous injection. The tumor volumes were monitored over time.

For adoptive T cell transfer assay, B16 cells were infected with 1 MOI LCMV-Cl13 for 24 h, and subcutaneous injected into the right flank of Balb/c nude mice. Meanwhile, C57BL/6 mice were intravenous (i.v.) infected with $1 \times 10^6$ LCMV-Cl13, and CD8$^+$ T cells were isolated from the spleen of the mice on day 7 post infection. Then the CD8$^+$ T cells were transferred into the tumor-bearing nude mice on day 10, 14, and 18 post tumor inoculation. Tumor volumes were monitored overtime.

**Viral infection**. For LCMV chronic infection mice model, 6–8-week-old mice were infected by i.v. injection with indicated titers of LCMV-Cl13.

**Quantitative real-time PCR (qRT-PCR)**. TRIzol reagents (invitrogen) were used to isolate total RNA from cells or tissues, followed by reverse transcribing with GoScript™ Reverse Transcription System (Promega). TransStart Top Green qPCRSuperMix (TransGen Biotech) was used for real-time PCR. qRT-PCR was performed using Applied Biosystems 7500 Fast & 7500 Real-Time PCR System and 7500 Software v2.3 (Applied Biosystems) (all primers are listed in Supplementary Table 2).

**Bone marrow transplantation (BMT)**. To obtain bone marrow cells, donor mice were sacrificed, and the femurs and tibias were harvested. The bone marrow canals were washed out with RPMI1640 medium containing 1% (vol/vol) FBS, and the bone marrow cells were collected, followed by red blood cell (RBC) lysis. Recipient mice were lethally irradiated (896 cGy/mouse), and i.v. injected with $5 \times 10^6$ bone marrow cells. Experiments on transplanted mice were performed after a latency of 30 days to ensure bone marrow reconstitution.

**In vitro cytotoxicity assay**. C57BL/6 mice were untreated or i.v. infected with $1 \times 10^6$ PFU LCMV-Cl13. On day 7 post infection, lymphocytes were isolated from spleens of the infected mice, and the CD8$^+$ T cells were isolated using Magni-Sort™ Mouse CD8 Positive Selection Kit (eBioscience). The CD8$^+$ T cells were incubated with target cells pulsed with 2 μg/ml GP$_{33-41}$ peptide at indicated ratios. 8 h post incubation, target cells were collected and stained with DCFDA, using for flow cytometry analysis. For percent cell lysis rate, target cells were washed to remove lymphocytes, and live cells were measured using Cell counting Kit-8 (CCK-8, Dojindo) following the manufacturer's instructions.

**Detection of ROS production**. ROS-sensitive fluorescent probe 2′,7′-dichlor-odihydrofluorescein diacetate (H2DCFDA) was used to detect generation of ROS in cultured cells. Briefly, cells were washed with PBS, followed by staining with 100 mM DCFDA for 30 min at 37 °C. After washing with FACS (PBS + 1% (vol/vol) FBS), stained cells were subjected to flow cytometry analysis.

For detection of lipid peroxidation, cells were collected, and stained with 2 μM BODIPY-C11. Lipid ROS level was determined by flow cytometry analysis with excitation at 488 nm.

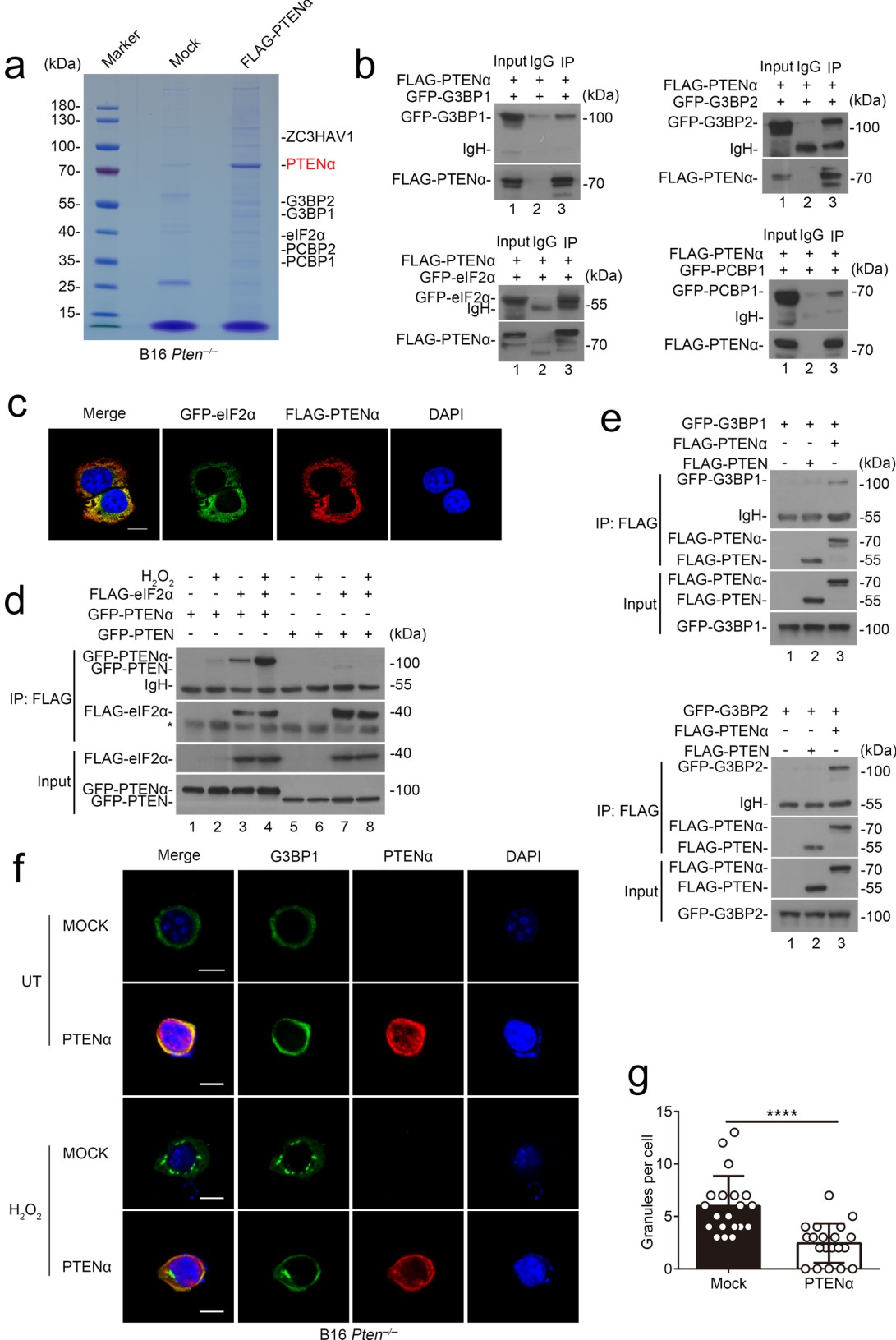

**TUNEL staining**. For the detection of Terminal deoxynucleotidyl transferase (TdT)-mediated dUTP nick end labeling (TUNEL) positive cells, the in-situ cell death detection kit (Roche) was used according to the manufacturer's instructions.

**Confocal microscopy**. Cells were seeded on cover glasses. 12 h later, cells were fixed with 100% acetone, permeabilized with 0.5% (vol/vol) Triton X-100 and then blocked using 1% (wt/vol) BSA dissolved in PBS. Primary antibody was applied for at least 1 h,

**Fig. 6 PTENα blocks stress granules formation during oxidative stress. a** MS analysis of PTENα associated proteins by FLAG pull-down assay in *Pten*$^{-/-}$ B16 cells treated with 100 μM H$_2$O$_2$. Proteins that interact with PTENα are indicated on the right (outlined text). **b** HEK293T cells were co-transfected with FLAG-tagged PTENα and GFP-tagged indicated vectors, followed by 100 μM H$_2$O$_2$ treatment for 3 h. Cell lysates were immunoprecipitated with anti-FLAG antibodies or normal mouse IgG. The immunoprecipitated proteins were subjected to immunoblot analysis with anti-GFP antibodies. **c** HeLa cells were co-transfected with FLAG-tagged PTENα and GFP-tagged eIF2α vectors, and interaction between the proteins was shown by confocal fluorescence microscopy. DAPI was used to indicate nuclear. Scale bars, 10 μm. **d** HEK293T cells were co-transfected with FLAG-tagged eIF2α and GFP-tagged PTEN or PTENα vectors in the absence or presence of H$_2$O$_2$ (100 μM). Cell lysates were immunoprecipitated with anti-FLAG antibody and analyzed by immunoblot with anti-GFP antibody. '*' refers to unspecific band. **e** HEK293T cells were transfected to express FLAG-tagged PTEN or PTENα, together with GFP-tagged G3BPs, followed by treatment of 100 μM H$_2$O$_2$ for 3 h. Cell lysates were immunoprecipitated with anti-FLAG antibody and assessed by immunoblot analysis with anti-GFP antibody. **f** and **g** *Pten*$^{-/-}$ B16 cells were transfected with Mock or FLAG-tagged PTENα vectors, followed by confocal fluorescence microscopy analysis in the presence or absence of 100 μM H$_2$O$_2$ with anti-G3BP1 and anti-PTEN (138G6) antibodies (**f**). Granules in the cells were counted, using for statistical analysis (**g**) ($n = 20$ cells, mean ± SD, ****$P < 0.0001$). Scale bars, 10 μm. UT, untreatment. Statistical significance was assessed by a two-tailed unpaired Student's *t* test (**g**). Data are representative of two (**b-i**) independent experiments. Source data are provided as a Source Data file.

followed by incubation with fluorophore-conjugated secondary antibody for 1 h. DAPI (BioDee Biotechnology) was used to indicate nuclear, and cells were evaluated with fluorescence microscopy using NIS-Elements AR Analysis 4.20.00 64 bit software (Nikon). NIS Elements Viewer 4.20 was used to analyze the confocal microscopy data.

**LDH release assay**. The CytoTox 96R Non-Radioactive Cytotoxicity Assay (Promega) was used to detect lactate dehydrogenase (LDH) release and cell death rates. In brief, cells were seeded in 96-well plates and treated with indicated reagents. The experimental LDH release and maximum LDH release were determined according to the manufacturer's instructions, and the percentage of cytotoxicity was calculated by dividing experimental LDH release by maximum LDH release.

**Interstitial fluid isolation**. Tissue was harvested from mice. Bulk tissue was placed on a 40 μm cell strainer, followed by centrifuging at $45 \times g$ for 5 min at 4 °C to remove surface fluid. Then the tissue was centrifuged at $500 \times g$ for 10 min at 4 °C, and the interstitial fluid was collected. After spinning at $10,000 \times g$ for 10 min at 4 °C to remove insoluble particles, the supernatant was subjected to immunoblot analysis and ATP content assay.

**Ponceau S staining**. Protein samples were subjected to immunoblot analysis. After the electrophoresis and transmembrane, polyvinylidene fluoride membrane was stained by Ponceau S staining solution (Beyotime) for 30 min. The membrane was washed and used for scanning.

**ATP content assay**. ATP contents in the interstitial fluids were detected using ATP assay kit (Nanjing Jiancheng) according to the manufacturer's instructions.

**Propidiumiodide (PI) staining**. Cultured cells were digested and fixed with 100% Ethanol, followed by staining with 1.1 mg/ml PI containing 1% (vol/vol) Triton-X100. Stained cells were used for flow cytometry analysis. For cell cycle analysis, peaks of cells with diploid DNA content were analyzed. Determination of cell death was performed by analyzing the sub-G1 peak.

**Hematoxylin and eosin (H&E) staining**. Tissue specimens were fixed in 10% (vol/vol) neutral buffered formalin followed by paraffin-sectioning and H&E staining.

**Preparation of lymphocytes**. Lymphocytes in spleen were isolated by grinding the tissues in PBS containing 1% (vol/vol) FBS, followed by filtering through a 75 μm strainer. To isolate lymphocytes infiltrating in the lung, minced tissues were digested with 0.5 mg/ml Collagenase D (Roche) and 25 μg/ml DNase I (Sigma) at 37 °C for 30 min. Then the tissues were grinded, and filtered through a 75 μm strainer. A 40% (10 ml)/70% (5 ml) Percoll gradient (GE Healthcare) was used to isolate the tissues by centrifuging $800 \times g$ for 20 min. Cells at the inter-layer were collected and counted for further operation. For isolation of TIL, tumors were minced, grinded and filtered just like the lungs mentioned above. Then the cells were resuspended in 5 ml FACS buffer, adding to a 40% (5 ml)/80% (5 ml) Percoll gradient. The gradient was centrifuged at $400 \times g$ for 45 min, and the cells at the inter-layer of 40%/80% Percoll gradient was collected for further operation.

**In vitro T cell differentiation assay**. Naïve CD4$^+$ T (CD4$^+$CD44$^{low}$) cells from spleens of C57BL/6 mice were sorted, and activated with plate-bound 2 μg/ml anti-CD3 (145-2C11; BioLegend) and 1 μg/ml anti-CD28 (37.51; BioLegend) antibodies. The cells were treated with various cytokines or antibodies for polarization. For Th1 cell differentiation, 10 μg/ml anti-IL-4 (11B11; BioLegend) antibody and 10 ng/ml IL-12 (R&D) were used. For Th17 polarization, 20 ng/ml IL-6 (R&D),

5 ng/ml TGF-β (R&D), 10 μg/ml anti-IFN-γ (XMG1.2; BioLegend) and 10 μg/ml anti-IL-4 antibodies were used. For Tr1 cell differentiation, cells were treated with 50 ng/ml IL-27 (eBioscience), 10 μg/ml anti-IFN-γ and 10 μg/ml anti-IL-4 antibodies. For iTreg polarization, 1 ng/ml TGF-β, 4 ng/ml IL-2 (R&D), 10 μg/ml anti-IFN-γ and 10 μg/ml anti-IL-4 antibodies were used. The cells were cultured at 37 °C for 48 or 72 h, followed by flow cytometry analysis.

For detection of activation induced cell death, Naïve CD4$^+$ T cells were activated with plate-bound 2 μg/ml anti-CD3 and 1 μg/ml anti-CD28 antibodies. 72 h later, cells were collected, and subjected to Annexin V/7-AAD staining.

**CFSE staining**. For CFSE staining, splenocytes were harvested, and washed twice with PBS, followed by staining with CFSE at dark for 10 min. Then the cells were washed twice with PBS containing 5% (w/w) FBS, and subjected to flow cytometry analysis.

**Annexin V/7-AAD staining**. Annexin V/7-AAD staining was performed using Annexin V Apoptosis Detection Kit (eBioscience), following the manufacturer's instructions.

**Flow cytometry**. For surface staining, cells were incubated with specific antibodies for 30 min at room temperature. The flow cytometry analyzer (BD Biosciences), FACSuite Software Bundle v1.0 (BD Biosciences), and FACSDiva Software v6.1 (BD Biosciences) were used for acquiring and analyzing the cells. The FACS data were analyzed with FlowJo v7.6.1 software. To perform intracellular cytokine staining, cells were stimulated with 100 ng/ml PMA and 500 ng/ml ionomycin, or 2 μg/ml GP$_{33-41}$ peptide together with GolgiPlug and GolgiStop (BD) for 5 h. Subsequently, cells were fixed and permeabilized, followed by staining with specific anti-cytokine antibodies.

**Pulldown assay and mass spectrometry**. H1299 cells were seeded and transfected with FLAG-tagged-PTENα. The whole-cell extracts were incubated with FLAG-beads (Sigma-Aldrich), followed by washing four times using PBS-N (PBS with 0.1% (vol/vol) NP40). Immunoprecipitated proteins were eluted using the 3× FLAG-peptide and subjected to SDS–PAGE. After staining with Coomassie Brilliant blue, the gel was excised and subjected to in-gel trypsin digestion and dried. 10 μl 0.1% formic acid was used to dissolve the peptides, and the peptides were sampled onto a 100 μm × 10 cm fused silica emitter packed with reversed-phase ReproSil-Pur C18-AQ resin (3 μm and 120 Å; Ammerbuch). Linear gradients of 5–32% acetonitrile in 0.1% formic acid were used to elute the sample at a flow rate of 300 nl/min for 50 min. An LTQ Orbitrap Elite mass spectrometer (Thermo-Fisher) equipped with a nanoelectrospray ion source (ProxeonBiosystems) was used to acquire the mass spectra data. Collision-induced dissociation (normalized collision energy, 35%; activation Q, 0.250; activation time, 10 ms) was used to perform fragmentation in the LTQ with a target value of 3000 ions. To search the raw files, the SEQUEST engine against a database from the Uniprot protein sequence database was used. Parameters were set as follows: protein modifications were set to carbamidomethylation (C) (fixed), oxidation (M) (variable), and phosphorylation (S, T, Y) (variable); the enzyme specificity was set to trypsin; a maximum missed cleavages were set to 2; the precursor ion mass tolerance was set to 10 ppm, and MS/MS tolerance was set to 0.5 Da.

**Co-immunoprecipitation and immunoblot analysis**. HEK293T cells were lysed by Co-IP lysis buffer containing 150 mM NaCl, 1 mM EDTA, 20 mM Tris–HCl pH 8.0, 10% (vol/vol) glycerol, 0.5% (vol/vol) NP40, and Protease inhibitor Cocktail (Roche). Cell lysates were subjected to SDS-PAGE and immunoblot analysis. For the co-immunoprecipitation assay, IgG or the appropriate antibodies were added into the cell lysates, followed by incubation with Protein A/G beads (EMD

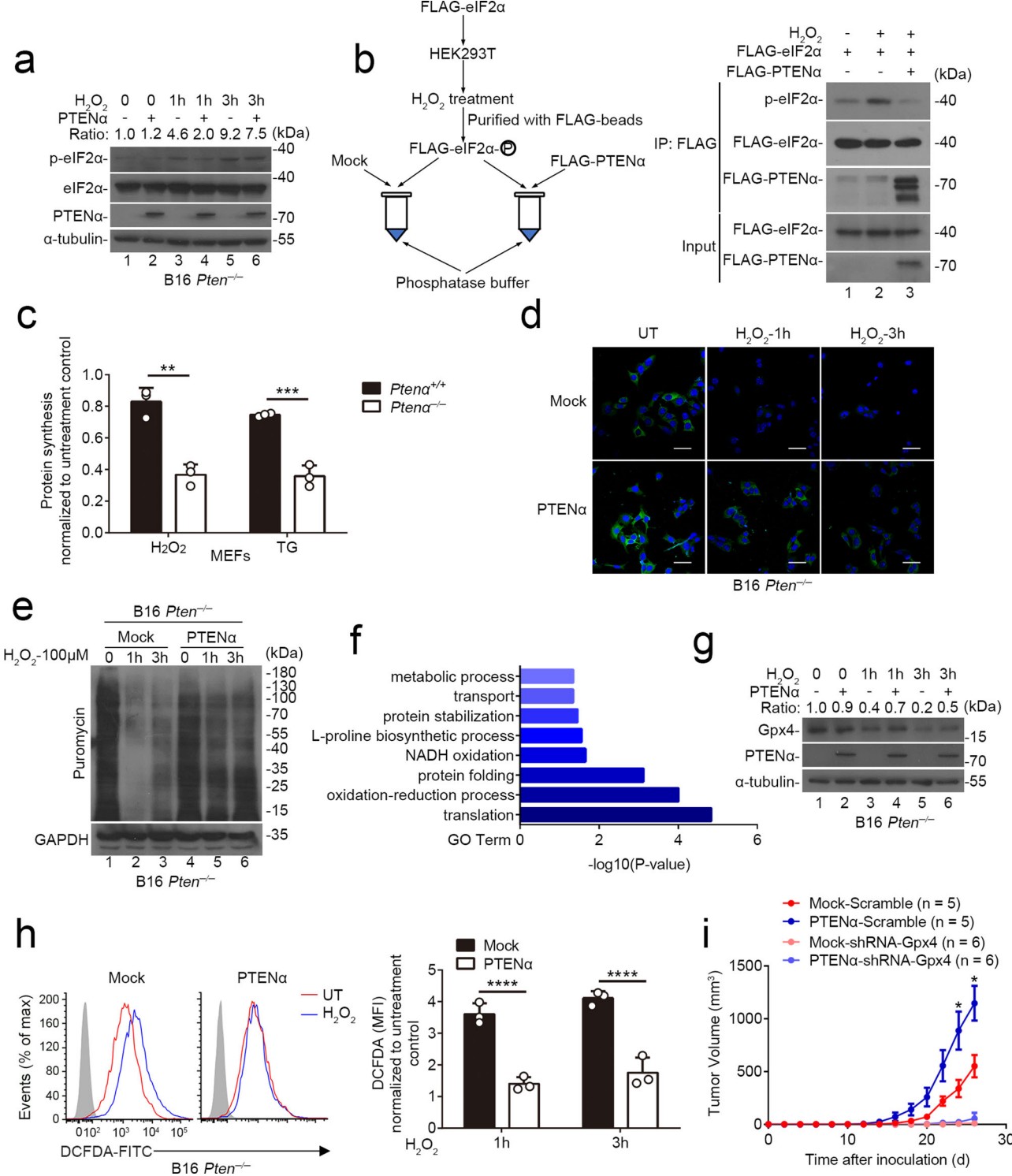

Millipore) for at least 4 h. The precipitants were washed with PBS-N (PBS with 0.1% (vol/vol) NP40) four times, using for immune blotting analysis.

Puromycin-labeled proteins were identified with immunoblot analysis or confocal microscopy.

**In vitro phosphatase assay**. HEK293T cells were transfected to express FLAG-tagged eIF2α and treated with 100 μM $H_2O_2$ to induce phosphorylation of eIF2α. Then the phosphorylated eIF2α was precipitated with FLAG beads, and incubated with precipitated FLAG-tagged PTENα in phosphatase buffer (20 mM Hepes pH = 7.2, 1 mM DTT, 1 mM $MgCl_2$ and 1 mM EDTA) for one hour. The proteins were eluted with FLAG peptide and subjected to immunoblot analysis.

**$^{35}$S-metabolic labeling**. $Ptenα^{+/+}$ and $Ptenα^{-/-}$ MEFs ($5 \times 10^5$) were treated with TG (500 nM) or $H_2O_2$ (100 μM) for 2 h. After starvation with methionine-free DMEM medium containing 2% (vol/vol) dialyzed fetal calf serum for 30 min, 20 μCi [$^{35}$S]-labeled methionine were added. 1 h later, cells were lysed, and labeled proteins were subjected to SDS–PAGE analysis. $^{35}$S-labeled proteins were detected with a PhosphorImager.

**Puromycin incorporation assay**. Indicated cells were treated with $H_2O_2$ (100 μM) for indicated hours, followed by pulsing with puromycin (50 μg/mL) for 30 min.

**Single-cell RNA sequencing sample preparation and data analysis**. For sample preparation, Mock or PTENα transfected $Pten^{-/-}$ B16 cells were subjected to the

**Fig. 7 PTENα maintains protein synthesis of peroxidases. a** Mock or PTENα expressing $Pten^{-/-}$ B16 cells were treated with 100 μM $H_2O_2$ for indicated hours, followed by immunoblot analysis with anti-eIF2α and anti-phospho-eIF2α antibodies. Gray values of total eIF2α and phosphorylated eIF2α were determined, and the relative ratio of phosphorylated eIF2α to total eIF2α was indicated. **b** In vitro phosphatase assay of FLAG-tagged PTENα on phosphorylated eIF2α. Phosphorylation level of eIF2α was assessed by immunoblot analysis with anti-phospho-eIF2α antibodies. **c** $^{35}$S-metabolic labeling analysis of $Ptenα^{+/+}$ and $Ptenα^{-/-}$ MEFs treated with 100 μM $H_2O_2$ or 500 nM TG for 3 h. Gray values of autoradiographs were determined and used for statistical analysis (pooled samples, $n = 3$ independent experiments, mean ± SD, **$P = 0.0020$, ***$P = 0.0007$). **d** and **e** Puromycin incorporation assay of Mock or PTENα expressing $Pten^{-/-}$ B16 cells treated with 100 μM $H_2O_2$ for indicated hours. Puromycin-labeled proteins were identified with confocal fluorescence microscopy (**d**) and immunoblot analysis (**e**). Scale bars, 100 μm. UT untreatment. **f** $Ptenα^{+/+}$ and $Ptenα^{-/-}$ MEFs were treated with 100 μM $H_2O_2$, followed by mass spectrometry analysis. Proteins that were significantly decreased in $Ptenα^{-/-}$ MEFs were analyzed using DAVID with Gene Ontology (GO) terms. **g** Mock or PTENα expressing $Pten^{-/-}$ B16 cells were treated with 100 μM $H_2O_2$ for indicated hours, and expression level of GPX4 was assessed by immunoblot analysis with anti-GPX4 antibody. Gray values of GPX4 and α-tubulin were determined, and the relative ratio of GPX4 to α-tubulin was indicated. **h** Flow cytometry analysis of Mock or PTENα expressing $Pten^{-/-}$ B16 cells treated with $H_2O_2$ (100 μM) for indicated hours and stained with DCFDA. Mean fluorescence intensities of DCFDA were used for statistical analysis ($n = 3$ cell cultures, mean ± SD, ****$P < 0.0001$). All live cells were gated. UT, untreatment. **i** Endogenous Gpx4 expression in B16-$Pten^{-/-}$ cells was knockdown using shRNA targeting Gpx4 or scramble shRNA. Then the cells were transfected to express Mock or PTENα and subjected to cancer vaccination model. The tumor volumes were monitored over time (Scramble, $n = 5$ mice; shRNA, $n = 6$ mice, mean ± s.e.m., *$P$(d24) = 0.0238, *$P$(d26) = 0.0158). Statistical significances between scramble groups were shown. Statistical significance was assessed by a two-tailed unpaired Student's $t$ test (**c**, **h**, **i**). Data are representative of two (**a**, **b**, **d**, **e**, **g–i**) independent experiments or pooled from three (**c**) independent experiments. Source data are provided as a Source Data file. See also Supplementary Fig. 10.

---

cancer vaccination model. The tumors were collected when the tumor volumes of Mock groups reached 200 mm³. The TILs were isolated, and CD45⁺ 7AAD⁻ cells were sorted, using for scRNA-seq with two replicates. Each sample is a mix of cells from two mice. Library construction and 10× scRNA-seq were performed by Berry Genomics. Sequencing was performed in different sequencing lanes in Illumina NovaSeq 6000.

For raw data processing and quality control, Cell Ranger (version 6.0.1) was used to produce a raw unique molecular identifier (UMI) count matrix through processes including cellular barcodes demultiplex, mapping reads to the transcriptome and generating quantitative matrix. The matrix was converted into a Seurat object by the R package Seurat (version 4.0.1). Low-quality cells with UMI numbers >5000 or <300, or have over 10% mitochondrial-derived UMI counts were removed. 25,098 remained single cells were applied in downstream analyses.

Subsequently, the count matrix was normalized with the SCTransform function. Then we performed integration on the four datasets. In this process, potential Anchors were created with the FindIntegrationAnchors function using top 3000 variable genes, and IntegrateData function was used to integrate data and create a new matrix with 3000 features, in which potential batch effect was regressed out.

To reduce the dimensionality of the scRNA-Seq dataset, principal component analysis (PCA) was performed. Top 30 PCs were used to perform the downstream analysis with Elbowplot function. Subsequently, t-distributed stochastic neighbor embedding (tSNE) and uniform manifold approximation and projection (UMAP) were performed. The FindClusters function was used to divide all cells into 22 clusters with resolution set as 0.6, and the FindAllmarkers function was performed to identify preferentially expressed genes in clusters. Based on these genes, the clusters were identified as 12 cell types, and visualized with 2D tSNE plots, and the heatmap plot was made with the DoHeatmap function. The proportion of each cell type in the four samples were counted by the R package dplyr (1.0.6), and the bar graph were drawn using R package ggplot2 (3.3.3).

For further analysis of the subpopulation of Cd8T cells, all Cd8T cells were divided into four clusters with the FindClusters function, with resolution set as 0.08. The FindAllMarkers function was performed to identify preferentially expressed genes in the clusters. Based on these genes, four clusters were identified as four cell types, and visualized with 2D UMAP plot. Heatmap plot was made using the DoHeatmap function, and the proportion of each cell type in the Cd8T cells were counted by the R package dplyr (1.0.6). The bar graph was made using the R package ggplot2 (3.3.3). We also drawn the violin plots to show the expression of important genes in the Cd8T cells with the VlnPlot function.

**Bioinformatics.** Mutations, mRNA expression data and clinical data were from cBioportal[31,32]. Stop gained mutations includes all the stop-gained and frameshift mutations early terminated after H[185] (last amino acid of phosphatase tensin-type domain of PTEN). Loss of phosphatase activity mutations were from UniProt database.

For gene-set enrichment analysis (GSEA), differentially expressed genes or proteins were analyzed using applications from Broad Institute 21 (http://www.broad.mit.edu/gsea/software/software_index.html) with default parameters. GO gene sets were downloaded from the 23 Broad Institute Molecular Signature Database (MSigDB).

**Statistics and reproducibility.** Prism GraphPad software v6.01 was used for analysis. The statistical significance between different groups was calculated with an unpaired Student's $t$-test or log-rank (Mantel–Cox) test. To compare three or more means, one or two-way ANOVA followed by Tukey's multiple comparisons test was used. $P < 0.05$ was considered significant. All experiments were independently replicated at least two times and similar results were generated.

**Reporting summary**. Further information on research design is available in the Nature Research Reporting Summary linked to this article.

## Data availability

The mass spectrometry proteome data generated in this study have been deposited in the Integrated Proteome Resources (iProX), an official member of ProteomeXchange Consortium, with the accession number IPX0002523000 and IPX0003269000. The scRNAseq data have been deposited in the GEO database with the accession code GSE178258. Sequencing data of PTEN in Molt4 and Jurkat cells have been deposited in the NCBI Genbank nucleotide database under accession code MZ615337, MZ615338 and MZ615339. Phosphatase-inactive PTEN mutations are from Uniprot database (https://www.uniprot.org/uniprot/P60484). Clinical data of the tumor patients carrying PTEN mutations from the TCGA (UCEC, PanCancer Atlas) and Metastatic Colorectal Cancer databases (MSKCC, Cancer Cell 2018[33]) can be acquired from cBioPortal (https://www.cbioportal.org). GO gene-sets are from MSigDB (http://www.gsea-msigdb.org/gsea/downloads.jsp). The remaining data are available within the Article, Supplementary Information or Source Data file. Source data are provided with this paper.

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

## Acknowledgements

This work was supported by grants including the National Key Research and Development Program of China (Grant 2016YFA0500302 to Y.Y.), the National Natural Science Foundation of China (Key grants 82030081 and 81874235 to Y.Y., and grant 82022032 and 81991505 to D.L.), the Lam Chung Nin Foundation for Systems Biomedicine, Clinical Medicine Plus X-Young Scholars Project, Peking University, the fundamental research funds for the Central Universities (No. PKU2021LCXQ026 to D.L.), and the Fundamental Research Funds for the Central Universities (No. BMU2018YJ003 to D.L.).

## Author contributions

Y.S., D.L. and Yuxin Y. conceived the study and designed experiments. Y.S. and D.L. performed the most of experiments and analyzed the data. J.S., W.H., Yue Y. and F.Q. assisted in some experiments. Y.J. provided technical assistance. Xuyang Z. performed mass spectrometry analysis. Xin Z. and G.Z. analyzed the sc-RNAseq data. L.L., Z.L., Y.J., Y.L. and H.L. provided reagents. D.L. and Yuxin Y. supervised the research. Y.S., D.L. and Yuxin Y. wrote the paper.

## Competing interests

The authors declare no competing interests.
