## [Peer Review File · Nature Communications]

Reviewers' Comments:

Reviewer #1:

Remarks to the Author:

The manuscript entitled "PTENa functions as an immune suppressor and promotes immune resistance in PTEN-mutant cancer" by Sun et al. reports that PTENa, as an isoform of PTEN, remains active in tumor with certain types of mutation called "stop-gained mutations". PTENa in tumor cells was shown to modulate host antitumor immune response. Using a whole-body knockout mouse, PTENa was shown to protect host from inflammation, but has no effect on T cells themselves. PTENa KO cells are more sensitive to T cell-mediated killing and H₂O₂-induced cell death. Lastly, PTENa was shown to regulate stress granules formation and protein synthesis of peroxidases. Overall, the functions of PENTa in vitro and in vivo are extensively studied here, and some phenomenon observed is very interesting and with high novelty, including the involvement of PTENa in tumor immunity and the regulation of PTENa on stress granules. However, the results from different parts are not logically and firmly connected to draw the conclusion that stated in the title. Here are major concerns.

1. Experimental models especially cell lines are not consistent across different figures. The manuscript starts with the phenomenon that PTENa may regulate tumor immunity. Bioinformatics are done in uterine corpus endometrial carcinoma (UCEC) but animal model is mouse melanoma B16. And then T cell killing and H₂O₂ cytotoxicity experiments are done in MEF cells. Later, at molecular level, the interactions between PTENa and stress granules are done in H1299, HEK293 or Hela cells (Fig 6). Lastly, figure 7 is done in PC3 and MEF cells. Therefore, it seems that each figure tells a different story and the causality among them is unclear, since the mechanism observed in each cell model may not be applied to all others.
2. Logical issue. Fig.1 shows that PTENa involves in cancer immunity through comparing PTEN^{-/-} PTENa ^{-/-} vs PTEN^{-/-} PTENa ^{+/+} B16 tumor cells, but later the functions of PTENa in host inflammation, T cell cytotoxicity and oxidative cell death are studied by comparing WT (PTEN^{+/+} PTENa ^{+/+}) vs PTEN^{+/+} PTENa ^{-/-}. There are logical gap between different parts. Further, the mechanism behind PTEN regulation on tumor immunity is still not convinced. The author should manipulate eIF2a or GPX4 expression in PTEN^{-/-} PTENa ^{+/+} B16 tumor cell to test their effects on tumor growth and tumor immune response.
3. The effect of PTENa on ferroptotic cell death is not conclusive. PTENa KO MEFs cells are more sensitive to T cell-mediated killing and H₂O₂-induced oxidative cell death. But whether oxidative cell death contributes to T-cell cytotoxicity in this model is unknown. And lipid ROS, as a hallmark of ferroptosis, should also be quantified in target cells.

There are also several concerns on the following figures and results.

1. In Fig. 1B and 1C, how about the survival of patients without PTEN mutations in UCEC? Will it be different from patients with stop-gained mutations?
2. In addition to UCEC, what is the correlation between PTEN mutations and patient survival in other types of cancer?
3. In Fig. 1C, why are only 4 immune related genes showed? How about other classical genes such as CD8, PD-1, IFN γ and TNF α ? In this case, immune response related gene signatures will be better.
4. Too few animals to get the conclusion for Fig. 1F. Further, instead of vaccine model, cancer immunotherapy models including anti-PD-1 or adoptive T-cell transfer (such as OT-I model) should be used to indicate the immune escape.
5. T-cell mediated killing effect is shown difference only when the ratio of effector vs. tumor reaches to 50:1(Fg.4C-F), suggesting effector cells (T cells) may not induce direct killing on target cells (MEF). Further, cell death in MEF cells should be measured directly using PI staining.
6. In Fig.4H, it seems no difference for HMGB1 release, which is a classical DAMPs. This need be explained. In addition, the change of HMGB1 is not mentioned in the text.
7. Statistics cannot be done in Fig. 4I and Fig.7H if there are only two samples in each group.
8. In Fig. 5, GPX4 expression should be analyzed in PTENa WT and KO MEF cells, not the whole lung.
9. It's better to quantify the amount of stress granules across different groups showed in Fig.6E.
10. It seems that the baseline level of ROS is different between WT and KO cells.

Reviewer #2:

Remarks to the Author:

Sun et al. present novel findings regarding the role of the PTEN isoform PTEN- α (aka PTEN-Long or PTEN-L) in regulating the immune response to cancers in a way that may have a meaningful impact on our understanding of pathogenesis. The authors show data in support of a hypothesis that the ability of T cells to kill tumor cells is suppressed by alpha expression in the tumor, which inhibits cell death due to oxidation through the regulation of protein translation. The idea is supported by data showing that truncating or stop mutations of the PTEN gene lead to loss of the PTEN protein due to reduced half-life without producing a loss of PTEN-alpha. While this is an interesting idea and there is some preliminary data to support it, the authors need to increase the rigor and the depth of the studies to make a convincing case. For instance there is no data showing that the stop mutations show increased protein expression at the endogenous level relative to missense mutations or that the stop mutations function as well as wild type alpha to suppress T cell killing or other immune modulatory activities.

Below are some other points that should be addressed.

The authors should indicate explicitly that PTEN-alpha is the same as the widely accepted name PTEN-L. This will be a great help for the literature. It would be best to use both names in the paper to avoid confusion.

Figure 1

In 1A, "stop gained" mutations should be defined. What are they? Does this include frameshift mutations?

1D and E. Half-life of stop mutations are performed with GFP fusions. There is no data showing effect of missense mutations using this expression system. These same experiments should be performed with untagged expression vectors as the GFP tag is large and can affect protein half-life.

What about the effect on steady state levels in actual tumors?

There is no data showing that the long half-life stop gain mutations can cause an altered inflammation when expressed compared to wild type Pten α . This should be experimentally tested.

B16 Pten $-/-$ melanoma cells generated and used in this study should be characterized to determine if there are off target mutations generated as a result of CRISPR modification.

The level of expression of exogenous Pten- α should be compared to the endogenous level that is normally expressed in B16 melanoma studies. It is quite possible that these studies are expressing extremely high levels of the protein.

The ability of Pten- α to exert its effect on the microenvironment may be due to its secretion. The investigators should generate mutations of Pten- α that cannot be secreted to determine their effect.

Fig 2. The Pten α mutant mice that are presented in this paper have not been fully characterized. How was the knockout performed? What effect does it have on Pten and Pten- α expression? Off target integration could lead to some of the findings presented, so ruling out these effects is needed to be sure that the effects seen are due to the intended genetic manipulation. Also, the possibility that the knockout leads paradoxically to increased Pten protein or an increase in the expression of other isoforms like M, N, etc, should be examined.

- Prior work by others (PMID: 29246444) has shown that mutation of Pten α in mice causes a hyper-inflammatory response in the lung after an inflammatory stimulus. The authors should discuss their work as it relates to the prior literature as it supports the author's findings.

- TNF- α should be one of the cytokines measured

Figure 3. The effects on T cell function are quite interesting. The authors refer to prior literature without citing it.

Figure 6

What domains of PTEN α are needed to interact with and regulate stress granule formation? Can PTEN do the same thing?

The authors claim that PTEN α is a phosphatase for eIF2 α . However, there is no direct evidence showing that PTEN acts as a protein phosphatase for eIF2 α . This claim should be removed from the paper.

Reviewer #3:

Remarks to the Author:

The manuscript titled "PTEN α functions as an immune suppressor and promotes immune resistance in PTEN-mutant cancer" is from one of the first groups that characterized the function of PTEN α (PTEN-long), a new PTEN isoform. By perturbing the expression of PTEN α in tumor cells and murine tissues, here, the authors aimed to dissect potential immunoregulatory roles of PTEN- α in tumor setting. The authors demonstrated distinct impacts of different mutations on the stability of PTEN isoforms and reduced infiltration of immune cells in PTEN α -expressing B16 tumors. Additionally, the authors used Ptena $^{-/-}$ mice to characterize the phenotype of this strain in DSS-induced colitis and LCMV infection setting. These results are interesting and highlight the involvement of PTEN α in regulating host immune responses. However, the authors failed to provide sufficient data to support several key conclusions. Moreover, the authors didn't comprehensively evaluate the activity of PTEN-controlled cell signaling pathways in the genetically modified cells and strains. Therefore, the mechanisms by which PTEN α regulates host immune responses remain largely unclear. The paper also suffers from loose logic connections between sections in the results part and lack of justifications of multiple models used in this paper.

The major concerns are:

1. According to the title and the descriptions in the introduction section, the paper should be centered on the immunoregulatory roles of PTEN α in tumor setting. Except the figure 1, the results of the remaining six figures are from either inflammation or viral infection models. Especially, the authors concluded that the immunological importance of PTEN α is T- cell extrinsic, as shown in the figures 1F-H. It is unclear why the authors didn't focused on characterizing the impact of PTEN α expression on tumor immune microenvironment or tumor sensitivity to T cell killings. Defining the underlying mechanisms by which PTEN α -expressing B16 tumors display immune resistant phenotype are critical and more related to their title. The conclusion that PTEN α promotes cell resistance to T cell cytotoxicity is primarily based on the phenotype of LCMV-infected MEF cells. However, tumor cells have much more dysregulated intrinsic pathways when compared with MEF cells.
2. Based on their previous work published in the 2017 Cell Reports paper, the authors substituted "the two CTG codons of mouse Ptena (CTG347 ... CTG362, which are equivalent to Homo sapiens CTG513) with GGA" to generate the "PTEN α - $^{-/-}$ mice" used in this paper. In the published references, Ptena was reported to be expressed by tumor cells and brain tissues. To better explain how knockout of Ptena enhances ConA-induced hepatitis and DSS-induced colitis, and reduced susceptibility of LCMV-infected MEF to T cell cytotoxicity, the authors should characterize the level of Ptena expression in different types of normal cells or tissues, particularly in liver, colon and immune cells. If these tissues/cells express Ptena, the impact of Ptena knockout on the activity of signal pathways controlled by PTEN in these tissues/cells should be addressed, too.
3. It has been reported that PTEN α can be secreted from cells and has the potential to antagonize the PI3K signaling in adjacent cells. Given that the function from secreted PTEN α can also shape tumor immune microenvironment, the author should determine whether overexpression or knockout of PTEN α in cells or mice change the level of secreted PTEN α .

4. Several experiments are missing important experimental aims. In the figure 1C, the authors only compared the difference between patients with stop gained mutations and phosphatase-inactive mutations. The authors should include the group without any PTEN mutations. The supplementary Figure 4 (especially for S4A) is important figure. The authors should consider to include it in Figure 3. Additionally, the author should provide the data from wild-type mice receiving the BM transfer from either WT or PTEN^Δ mice in S4A. Moreover, does PTEN^Δ knockout alter the proliferation and survival of cytotoxic T cells and helper T cells? Lastly, why was the analysis of figure 1B limited in UCEC patients?

Reviewers' comments:

Reviewer #1 (Remarks to the Author): with expertise in ferroptosis and immune resistance

The manuscript entitled “PTENa functions as an immune suppressor and promotes immune resistance in PTEN-mutant cancer” by Sun et al. reports that PTENa, as an isoform of PTEN, remains active in tumor with certain types of mutation called “stop-gained mutations”. PTENa in tumor cells was shown to modulate host antitumor immune response. Using a whole-body knockout mouse, PTENa was shown to protect host from inflammation, but has no effect on T cells themselves. PTENa KO cells are more sensitive to T cell-mediated killing and H₂O₂-induced cell death. Lastly, PTENa was shown to regulate stress granules formation and protein synthesis of peroxidases. Overall, the functions of PENTa in vitro and in vivo are extensively studied here, and some phenomenon observed is very interesting and with high novelty, including the involvement of PTENa in tumor immunity and the regulation of PTENa on stress granules. However, the results from different parts are not logically and firmly connected to draw the conclusion that stated in the title. Here are major concerns.

1. Experimental models especially cell lines are not consistent across different figures. The manuscript starts with the phenomenon that PTENa may regulate tumor immunity. Bioinformatics are done in uterine corpus endometrial carcinoma (UCEC) but animal model is mouse melanoma B16. And then T cell killing and H₂O₂ cytotoxicity experiments are done in MEF cells. Later, at molecular level, the interactions between PTENa and stress granules are done in H1299, HEK293 or Hela cells (Fig 6). Lastly, figure 7 is done in PC3 and MEF cells. Therefore, it seems that each figure tells a different story and the causality among them is unclear, since the mechanism observed in each cell model may not be applied to all others.

[Response] According to these suggestions, we have repeated all the experiments with B16 cells. As shown in **Figure 4F** in the revised manuscript, enforced expression

of PTEN α in *Pten*^{-/-} B16 cells attenuated T cell-mediated cytotoxicity. Moreover, the amount of released ATP and the intracellular ROS level from PTEN α -expressing B16 cells (*Pten*^{-/-}) were decreased as compared with control cells did (**Fig. 4G and 4H** in the revised manuscript). Our data thus support the notion that PTEN α promotes cancer cell resistance to T cell killing through blocking DAMPs release.

To further confirm the regulatory role of PTEN α in cancer cell response to oxidative stress, we used H₂O₂ to treat B16 cells (*Pten*^{-/-}) in presence or absence of PTEN α . As shown in **Figure 5A-C** in the revised manuscript, enforced expression of PTEN α increased cancer cell resistance to oxidative cell death. Furthermore, we used *Pten*^{-/-} B16 cells to analyze the interactome of PTEN α . Consistent to the data in lung cancer cells (H1299), a series of key components of stress granule including eIF2 α , G3BPs and PCBP1 were associated with PTEN α in B16 cells (*Pten*^{-/-}) (**Fig. 6A** in the revised manuscript). The association of PTEN α with G3BP1 in B16 cells was further confirmed by confocal assay (**Fig. 6E** in the revised manuscript). Moreover, similar to MEFs cells, enforced expression of PTEN α inhibited phosphorylation of eIF2 α in B16 cells (**Fig. 7A** in the revised manuscript). To determine whether PTEN α can also promotes protein synthesis in B16 cells, we employed puromycin incorporation assay in B16 cells under the treatment of H₂O₂. As shown in **Figure 7D and 7E** in the revised manuscript, enforced expression of PTEN α in B16 cells ameliorated the translational inhibition induced by oxidative stress, which was analogue to the data using MEFs cells. Additionally, we also measured the levels of GPX4 and intracellular ROS in B16 cells. As shown in **Figure 7G** in the revised manuscript, PTEN α presence maintained GPX4 expression in B16 cells under oxidative stress. Conversely, the level of intracellular ROS was decreased in PTEN α -expressing B16 cells as relative to control cells (**Fig. 7H** in the revised manuscript). Collectively, our data demonstrate that PTEN α restricts oxidative cell death and blocks T cell-mediated eradication of cancer.

2. Logical issue. Fig.1 shows that PTEN α involves in cancer immunity through comparing PTEN^{-/-} PTEN α ^{-/-} vs PTEN^{-/-} PTEN α ^{+/+} B16 tumor cells, but later the functions of PTEN α in host inflammation, T cell cytotoxicity and oxidative cell death are studied by comparing WT (PTEN^{+/+} PTEN α ^{+/+}) vs PTEN^{+/+} PTEN α ^{-/-}. There are logical gap between different parts. Further, the mechanism behind PTEN regulation on tumor immunity is still not convinced. The author should manipulate eIF2 α or GPX4 expression in PTEN^{-/-} PTEN α ^{+/+} B16 tumor cell to test their effects on tumor growth and tumor immune response.

[Response] The following efforts were made to address this concern. As shown in **Figure 4F** in the revised manuscript, enforced expression of PTEN α in *Pten*^{-/-} B16 cells attenuated T cell-mediated cytotoxicity. Moreover, the amount of released ATP and the intracellular ROS level from PTEN α -expressing B16 cells (*Pten*^{-/-}) were decreased as compared with control cells did (**Fig. 4G and 4H** in the revised manuscript). Consistently, we also observed diminished oxidative cell death in PTEN α overexpressed B16 cells (**Fig. 5A-C** in the revised manuscript). Furthermore, we analyzed the status of eIF2 α and Gpx4 in *Pten*^{-/-} B16 cells with or without

PTEN α . As shown in **Figure 7A** and **7G** in the revised manuscript, enforced expression of PTEN α promoted eIF2 α dephosphorylation and maintained Gpx4 expression in B16 cells (*Pten*^{-/-}) under oxidative stress.

In order to ascertain that suppression of ferroptosis mainly contributes to the oncogenic function of PTEN α , we used α -TOC (most abundant form of vitamin E) that is radical scavenger and lipid peroxidation inhibitor, to suppress ferroptosis commitment. Considering only a portion of α -TOC dosage can reach the tumor by oral administration and its water insoluble characteristic, we exploited a molecular-matched strategy to prepare α -TOC-loading TPGS nanoparticles (NP-VE) with extremely high drug loading levels (up to 10 mg/ml) (PMID: 33393230). Both *in vitro* and *in vivo* assays showed that α -TOC treatment weakened the difference of cell death between PTEN α -expressing cells and controls cells exposure to immune attack (**Fig. 5D** and **5F** in the revised manuscript). Our data thus demonstrate that PTEN α limits ferroptotic cell death and attenuates cancer immune response.

3. The effect of PTEN α on ferroptotic cell death is not conclusive. PTEN α KO MEFs cells are more sensitive to T cell-mediated killing and H2O2-induced oxidative cell death. But whether oxidative cell death contributes to T-cell cytotoxicity in this model is unknown. And lipid ROS, as a hallmark of ferroptosis, should also be quantified in target cells.

[Response] We thank the reviewer for this question. To determine whether oxidative cell death contributes to T cell-mediated cytotoxicity, we co-cultured the virus-specific CD8⁺ T cells (Effector) with viral-peptide pulsed PTEN α -expressing cells or controls cells (Target). As shown in **Figure 5D** in the revised manuscript, enforced expression of PTEN α impaired T cell attack against cancer cells. Notably, supplementation of α -TOC-loading TPGS nanoparticles (NP-VE) weakened the cell death in control cells rather than PTEN α -expressing cells, which further confirm that the regulatory role of PTEN α on ferroptotic cell death is critical for its effects on cancer immune escape (**Fig. 5D** in the revised manuscript). Besides, we also used BODIPY-C11 dye to analyze the level of lipid ROS. Analogue to the intracellular ROS level, the presence of PTEN α limited the level of lipid ROS in B16 cells (*Pten*^{-/-}) under T cell attack (**Fig. 5E** in the revised manuscript).

There are also several concerns on the following figures and results.

1. In Fig. 1B and 1C, how about the survival of patients without PTEN mutations in UCEC? Will it be different from patients with stop-gained mutations?

[Response] As suggested, we have analyzed the survival of patients with or without *PTEN* mutations in UCECs. Unexpectedly, TCGA database revealed that patients with *PTEN* mutations had a better prognosis than those with wild-type *PTEN* (**Response Fig. 1A**). It is reported that the frequency of somatic mutations in *PTEN* and *P53* are mutually exclusive in cancer progression (PMID: 12379854). We thus compared the DNA mutations in cancers with or without *PTEN* mutations. As shown in **Response Figure 1B**, the frequency of p53 mutation was significant higher in wild-type *PTEN*

cancers than those in *PTEN*-mutant cancers. Moreover, the mutation of *p53* is associated with a worse prognosis of patients with cancer (**Response Fig. 1C**). We therefore speculate that the mutually exclusive recurrent *PTEN* and *p53* mutations results in the better prognosis of patients with *PTEN* mutations.

To avoid the effects due to *p53* mutations, we interrogated *PTEN* mutation data from Metastatic Colorectal Cancer database (MSKCC, Cancer Cell 2018) and Breast Cancer database (METABRIC, Nature 2012 & Nat Commun 2016). As shown in **Figure S1A** in the revised manuscript and **Response Fig. 1D**, compared with patients without *PTEN* mutations, the outcome of patients with stop-gained mutations were worse. Notably, less effects of mutually exclusive mutations between *p53* and *PTEN* were detected in these databases (**Response Fig. 1E and 1F**).

Response Figure 1. The relationship of *PTEN* mutations with the outcome of patients with UCEC.

(A) Clinical data of uterine corpus endometrial carcinoma (UCEC) patients with *PTEN* mutations were acquired from the TCGA database, and used for drawing of survival curve (*PTEN-mutant*, n = 352; *PTEN-wildtype*, n = 176, ***p* < 0.01).

(B) Analysis of DNA mutation frequency between *PTEN-mutant* and *PTEN-wt* cancers from TCGA database.

(C) Clinical data of uterine corpus endometrial carcinoma (UCEC) patients with *p53* mutations were acquired from the TCGA database, and used for drawing of survival curve (*TP53-mutant*, n = 194; *TP53-wildtype*, n = 334, ***p* < 0.01).

(D) Clinical data of breast cancer (BRCA) patients with *PTEN* mutations were acquired from the Breast Cancer database (METABRIC, Nature 2012 & Nat Commun 2016), and used for drawing of survival curve (WT, n = 1850; stop-gained mutations, n = 25; phosphatase-inactive mutations, n = 6, **p* < 0.05).

(E) Analysis of DNA mutation frequency between *PTEN-mutant* and *PTEN-wt* cancers from Metastatic Colorectal Cancer database (MSKCC, Cancer Cell 2018).

(F) Analysis of DNA mutation frequency between *PTEN-mutant* and *PTEN-wt*

cancers from Breast Cancer database (METABRIC, Nature 2012 & Nat Commun 2016).

2. In addition to UCEC, what is the correlation between PTEN mutations and patient survival in other types of cancer?

[Response] Compared with the amount of UCEC patients with *PTEN* mutations, the numbers of patients with other types of cancers carrying *PTEN* mutations in TCGA database are limited, which are no more than 10 cases with phosphatase-inactive or stop-gained mutations. To further confirm this result, we interrogated *PTEN* mutations data from metastatic colorectal cancer database. As shown in **Figure S1A** in the revised manuscript, the outcome of patients with stop-gained mutations tend to be worse than those with phosphatase-inactive mutations or wild-type *PTEN*.

3. In Fig. 1C, why are only 4 immune related genes showed? How about other classical genes such as CD8, PD-1, IFN γ and TNF α ? In this case, immune response related gene signatures will be better.

[Response] To comprehensively analyze the differentially expressed genes in cancers with these two-type mutations, we firstly screened 8 patients with phosphatase-inactive mutation and 17 patients with stop-gained mutation from TCGA database based on the clinical criteria such as matched diagnosis age (40-60 yrs), aneuploidy score (<3) and neoplasm histologic grade (G3). Utilizing Gene Set Enrichment Analysis (GSEA), we found that genes related with activation of immune response or adaptive immune response were selectively enriched in cancers with phosphatase-inactive mutation as compared with those with stop-gained mutations (**Fig. 1C and S1B** in the revised manuscript). Our data thus indicate that stop-gained *PTEN* mutations is involved in cancer immunosuppression.

4. Too few animals to get the conclusion for Fig. 1F. Further, instead of vaccine model, cancer immunotherapy models including anti-PD-1 or adoptive T-cell transfer (such as OT-I model) should be used to indicate the immune escape.

[Response] We appreciate this suggestion and repeated this experiment. As shown in **Figure 5F** in the revised manuscript, overexpression of PTEN α restricted host immune response and facilitated tumor development. Notably, we observed that less amounts of CD45⁺ immune cells were detected in PTEN α -expressing tumors as relative to control tumors by flow cytometry (**Fig. 2C** in the revised manuscript).

To further study the status of immune cells in PTEN α -expressing tumor, we applied scRNA-seq methods to investigate CD45⁺ immune cells isolated from PTEN α -expressing or control tumors. Utilizing graph-based clustering to analyze the tumor-infiltrating immune cells, we identified 10 clusters for 10x data including 8605 cells. We then defined the clusters based on the exclusive expression of canonical marker genes. As shown in **Figure 2D** and **Figure S2D** in the revised manuscript,

major immune cell types in cancers including CD4⁺ T, Treg, CD8⁺ T, NK, monocyte, neutrophil, myeloid derived suppressor cells (MDSCs), B, plasma B and pDCs. Compared with more than 70% of lymphoid cells in control tumor, the percentage of T lymphocytes and B lymphocytes were significantly reduced in PTEN α -expressing tumors (**Fig. 2E** and **2F** in the revised manuscript). Conversely, the ratio of innate immune cells, especially MDSCs, was increased in PTEN α -expressing tumors as relative to control tumors (**Fig. 2E** and **2F** in the revised manuscript). In light of the importance of CD8⁺ T cells in host antitumor immunity, we thus analyzed the differentially expressed genes in CD8⁺ T cells from PTEN α -expressing or control tumors. As shown in **Figure 2G** in the revised manuscript, in contrast to high level of genes related with T cell exhaustion including *Dusp2*, *Hif1a*, *Pdcd1*, *Havcr2* and *Lag3* in CD8⁺ T cells from PTEN α -expressing tumor, genes related with memory T cell formation including *Ccr7*, *Tcf7*, *Sell* and *Il7r* were selectively upregulated in CD8⁺ T cells from control tumor. Consistent with scRNA-seq results, subsequent flow cytometry assay revealed that overexpression of PTEN α limited the amounts of CD4⁺ and CD8⁺ infiltrated T cells, and enhanced PD-1 expression on CD8⁺ tumor-infiltrating T cells (**Fig. 2H** and **2I** in the revised manuscript). Our data thus indicate that PTEN α remains active in PTEN mutant cancers and subverts host immune attack.

To further confirm the conclusion for Fig. 1F, we repeated the cancer vaccine model with PTEN α ^{CTG-R233*}-expressing B16 cells (*Pten*^{-/-}). As shown in **Figure S2E and S2F** in the revised manuscript, overexpression of PTEN α ^{CTG-R233*} impaired the stability of PTEN rather than PTEN α . Analogue to the immunosuppressive effects of wild-type PTEN α , overexpression of PTEN α ^{CTG-R233*} reduced the effectiveness of tumor vaccine and promoted tumor growth (**Fig. 2J** in the revised manuscript). Furthermore, we also rescued the expression of PTEN with or without PTEN α in B16 cells (*Pten*^{-/-}) (**Fig. S2E and S2F** in the revised manuscript). As shown in **Figure 2K** in the revised manuscript, PTEN α ^{CTG}-expressing tumors developed faster than PTEN-expressing tumors, which further support the notion that PTEN α elicits immunosuppressive function and facilitates cancer immune escape.

In addition to the cancer vaccine model, we also employed B16 lung metastasis model. As shown in **Figure S2G and S2H** in the manuscript, enforced expression of PTEN α increased pulmonary metastasis of B16 cells. In light of the inhibitory role of PTEN α in modulation of ferroptosis, our data thus support the work of Ubellacker (PMID: 32814895) that the sensitivity of tumor cells to oxidative stress limits their metastasis through blood.

Finally, we employed bone marrow (BM) transplantation model to analyze the role of PTEN α in modulation of immune response. As shown in **Figure 3E** in the revised manuscript, 30 days following LCMV Clone 13 infection, compared with wild-type donor mice, greater amounts of virus-specific T cells were detected in *Ptena*^{-/-} recipient mice adoptive transferred with wild-type BM or those of *Ptena*^{-/-} BM. Consistent with the greater amount of virus-specific T cells, lower level of inhibitory receptors and enhanced effector cytokine production were also detected in virus-specific T cells from *Ptena*^{-/-} recipient mice as relative to their littermate

control did (**Fig. S5A and S5C** in the revised manuscript). Conversely, the amount of virus-specific T cells from WT recipient mice adoptive transferred with WT or *Ptena*^{-/-} BM was limited, which expressed higher level of inhibitory receptors and less effector cytokine production as compared with those from *Ptena*^{-/-} donor mice (**Fig. 3F, S5B and S5D** in the revised manuscript). Our data thus demonstrate that PTEN α exerts immunosuppressive effects in a T cell-extrinsic manner.

5. T-cell mediated killing effect is shown difference only when the ratio of effector vs. tumor reaches to 50:1(Fg.4C-F), suggesting effector cells (T cells) may not induce direct killing on target cells (MEF). Further, cell death in MEF cells should be measured directly using PI staining.

[Response] We thank the reviewer for this suggestion. In the revised manuscript, we have repeated the *in vitro* T cell cytotoxicity assay with B16 cells (*Pten*^{-/-}). As shown in **Figure 4F** in the revised manuscript, enforced expression of PTEN α significantly attenuated T cell killing when the ratio of effector vs. target reach to 2:1. Besides, lower level of released ATP and ROS production were also detected in PTEN α -expressing B16 cells as compared with control cells when the ratio of effector vs. target reach to 8:1 and 15:1, respectively (**Fig. 4G and 4H** in the revised manuscript). Moreover, we have used PI staining to measure the cell death of B16 cells exposure to cytotoxic T cell as shown in **Figure 5D** in the revised manuscript.

6. In Fig.4H, it seems no difference for HMGB1 release, which is a classical DAMPs. This need be explained. In addition, the change of HMGB1 is not mentioned in the text.

[Response] The fact that it seems no difference for HMGB1 release is mainly due to lower amount of protein loading in *Ptena*^{-/-} group. We then employed *ImageJ* software to quantify the level of HMGB1 and found greater amount of HMGB1 releasing from lung tissues of *Ptena*^{-/-} mice (**Response Fig. 2**). To better present our data, we have repeated this experiment. As shown in **Figure 4D** in the revised manuscript, greater amounts of DAMPs including Hsp70, Hsp90 and HMGB1 were detected in lung tissues of *Ptena*^{-/-} mice as compared with those from wild-type mice.

Response Figure 2. Deficiency of PTEN α promotes DAMPs release upon LCMV-Cl13 infection

The interstitial fluids from lungs of wild-type and *Pten* $\alpha^{-/-}$ mice *i.v.* infected with LCMV-C113 were isolated on day 7 post infection. The fluids were used for immunoblot analysis with indicated antibodies. The equal amounts of loading proteins were showed by the Ponceau S staining. The relative ratios of Hsp70, Hsp90 and HMGB1 to Ponceau S stained total proteins were indicated. UT, untreated.

7. Statistics cannot be done in Fig. 4I and Fig.7H if there are only two samples in each group.

[Response] We have repeated these experiments with at least three samples in each group as shown in **Figure 4E** (Figure 4I in the previous manuscript) and **Figure 7H** in the revised manuscript.

8. In Fig. 5, GPX4 expression should be analyzed in PTEN α WT and KO MEF cells, not the whole lung.

[Response] We thank the reviewer for this suggestion. We have measured the expression of Gpx4 in B16 cells (*Pten* $\alpha^{-/-}$). As shown in **Figure 7G** in the revised manuscript, compared with control cells, enforced expression of PTEN α maintained the protein level of Gpx4 in B16 cells under oxidative stress.

9. It's better to quantify the amount of stress granules across different groups showed in Fig.6E.

[Response] As suggested, we have quantified the amount of stress granules across different groups in the revised manuscript. As shown in **Figure 6F** in the revised manuscript, enforced expression of PTEN α blocked stress granule formation under oxidative stress.

10. It seems that the baseline level of ROS is different between WT and KO cells.

[Response] Indeed, the baseline level of ROS in wild-type MEFs is different from those in *Pten* $\alpha^{-/-}$ MEFs. To better study the role of PTEN α in modulation of intracellular ROS, we thus used PTEN α -expressing B16 cells and control cells. As shown in **Figure 7H** in the revised manuscript, enforced expression of PTEN α dampened the upregulation of intracellular ROS induced by H₂O₂.

Reviewer #2 (Remarks to the Author): with expertise in PTEN biology

Sun et al. present novel findings regarding the role of the PTEN isoform PTEN- α (aka PTEN-Long or PTEN-L) in regulating the immune response to cancers in a way that may have a meaningful impact on our understanding of pathogenesis. The authors show data in support of a hypothesis that the ability of T cells to kill tumor cells is suppressed by alpha expression in the tumor, which inhibits cell death due to oxidation through the regulation of protein translation.

The idea is supported by data showing that truncating or stop mutations of the PTEN gene lead to loss of the PTEN protein due to reduced half-life without producing a loss of PTEN-alpha. While this is an interesting idea and there is some preliminary data to support it, the authors need to increase the rigor and the depth of the studies to make a convincing case. For instance there is no data showing that the stop mutations show increased protein expression at the endogenous level relative to missense mutations or that the stop mutations function as well as wild type alpha to suppress T cell killing or other immune modulatory activities.

Below are some other points that should be addressed.

The authors should indicate explicitly that PTEN-alpha is the same as the widely accepted name PTEN-L. This will be a great help for the literature. It would be best to use both names in the paper to avoid confusion.

[Response] We thank the reviewer for this suggestion. We have added *PTEN-L* in our revised manuscript.

Figure 1

In 1A, “stop gained” mutations should be defined. What are they? Does this include frameshift mutations?

[Response] Stop gained mutations includes all the nonsense or frameshift mutations terminated after H185 (last amino acid of phosphatase tensin-type domain of PTEN), which mainly impair the stability of PTEN protein rather than affect its phosphatase activity. In our revised manuscript, we have defined these types of mutation in the legend of **Figure 1A**.

1D and E. Half-life of stop mutations are performed with GFP fusions. There is no data showing effect of missense mutations using this expression system. These same experiments should be performed with untagged expression vectors as the GFP tag is large and can affect protein half-life.

[Response] According to these suggestions. We firstly cloned a series of wild-type or mutant open reading frame (ORF) of *PTEN α* into untagged expression vectors. As shown in **Figure S2E and S2F** in the revised manuscript, compared with *PTEN α ^{CTG}* which expresses wild-type PTEN and *PTEN α* , enforced expression of *PTEN α ^{CTG-R233*}* or *PTEN α ^{CTG-Q298*}* impaired the stability of PTEN rather than *PTEN α* . Conversely, enforced expression of the phosphatase-inactive mutations such as *PTEN α ^{CTG-C124S}* or *PTEN α ^{CTG-R130Q}* hardly affected the protein level of PTEN or *PTEN α* (**Fig. S2E and S2F** in the revised manuscript).

What about the effect on steady state levels in actual tumors?

[**Response**] The following efforts were made to address this concern. Despite high frequency of *PTEN* mutations in cancer, it is still hard to obtain actual tumors bearing with stop-gained mutations (**Response Figure 3**). Through utilizing the Cancer Cell Line Encyclopedia (Broad, 2019) database, we found that both Jurkat and MOLT4 cells contain stop-gained mutations. Subsequent DNA sequencing analysis further confirmed the bioinformatic data that MOLT4 cells contain *PTEN*^{K267Rfs*9} mutation and Jurkat cells contain *PTEN*^{R234Afs*1} and *PTEN*^{L247*} mutations (**Fig. S1E** in the revised manuscript). We then employed anti-N-PTEN antibody to assess the status of endogenous PTEN and PTEN α in Jurkat and MOLT4 cells. As shown in **Figure S1E** in the revised manuscript, compared with wild-type PTEN and PTEN α in HEK-293T, truncated PTEN α rather than PTEN was detected in both Jurkat and MOLT4 cells. Our data thus demonstrate that PTEN α is persistently expressed in *PTEN*-mutant cancers.

Response Figure 3. The frequency of *PTEN* mutation in multiple cancers.

The *PTEN* mutation data from the TCGA database. Percentages of the patients carrying phosphatase-inactive mutations or stop-gained mutations in total tumor patients were calculated, using for drawing the plot.

There is no data showing that the long half-life stop gain mutations can cause an altered inflammation when expressed compared to wild type Ptena. This should be experimentally tested.

[**Response**] To determine whether stop-gained *PTEN* mutations elicit similar effects as wild-type PTEN α did, we employed the cancer vaccine model with PTEN α ^{CTG-R233*}-expressing B16 cells (*Pten*^{-/-}). As shown in **Figure S2F** in the revised manuscript, overexpression of PTEN α ^{CTG-R233*} impaired the stability of PTEN rather than PTEN α . Analogue to the immunosuppressive effects of wild-type PTEN α , overexpression of PTEN α ^{CTG-R233*} reduced the effectiveness of tumor vaccine and promoted tumor growth (**Fig. 2J** in the revised manuscript). Our data thus demonstrate that stop-gain mutations remain PTEN α expression, which in turn elicit immunosuppressive function and facilitate cancer immune escape.

B16 Pten^{-/-} melanoma cells generated and used in this study should be characterized to determine if there are off-target mutations generated as a result of CRISPR modification.

[Response] To test whether there are any potential off-targets in our B16 *Pten^{-/-}* cells, we used the online tools (<https://cm.jefferson.edu/Off-Spotter/>) to find out the transcripts that may be targeted by the sgRNA (TGTGCATATTTATTGCATCG). Then we selected the top ten matched transcripts except for *PTEN* for further analysis. Finally, DNA sequencing was used to verify whether the genome sequence of the corresponding misrecognized gene was altered in B16 *Pten^{-/-}* cells. As shown in **Response Figure 4**, neither genome mutation nor deletion was detected in this homozygous *Pten* knockout cell clone, suggesting that there may not contain putative off-targets in B16 *Pten^{-/-}* cells.

Response Figure 4. DNA sequencing analysis of the top ten matched transcripts in *Pten*^{+/+} B16 and *Pten*^{-/-} B16 cells.

Genomic DNA of the *Pten*^{+/+} B16 and *Pten*^{-/-} B16 cells were extracted and used as the template for amplifying by PCR. The products of amplification were subjected to sequencing analysis. Homo refers to *Pten*^{-/-} B16 cells.

The level of expression of exogenous Pten-α should be compared to the endogenous level that is normally expressed in B16 melanoma studies. It is quite possible that these studies are expressing extremely high levels of the protein.

[Response] We thank the reviewer for this question. Compared with high level of PTENα in PTENα expression vector (ATG:ATG), the expression of PTENα in PTENα^{CTG} expression system (CTG:ATG) approximated its physiological status (**Fig. S2E and S2F** in the revised manuscript). We then transfected the vector of PTENα^{CTG} or PTEN into *Pten*^{-/-} B16 cells. As shown in **Response Figure 5**, the level of exogenous PTENα in *Pten*^{-/-} B16 cells was comparable to the level of endogenous PTENα.

To determine whether the PTENα^{CTG} expression system elicits the identical effects as the PTENα expression system, we used PTEN-expressing B16 cells and PTENα^{CTG}-expressing B16 cells to repeat cancer vaccine model. As shown in **Figure 2K** in the revised manuscript, overexpression of PTENα weakened the effectiveness of cancer vaccination and promoted tumor growth. Our data thus demonstrate that PTENα exerts immunosuppressive effects and facilitates cancer immune escape.

Response Figure 5. The expression of PTEN α and PTEN in B16 cells.

Pten^{-/-} B16 cells were transfected to express PTEN α or PTEN, followed by western blot analysis with anti-PTEN antibody.

The ability of Pten- α to exert its effect on the microenvironment may be due to its secretion. The investigators should generate mutations of Pten- α that cannot be secreted to determine their effect.

[Response] To ascertain whether the immunosuppressive role of PTEN α is dependent on its secretion, we thus employed the vector of PTEN α ^{CTG- Δ 6A} that has been reported to abolish secretion of PTEN α (PMID: 23744781). Utilizing the cancer vaccine model, we found that overexpression of PTEN α ^{CTG- Δ 6A} also promoted cancer immune escape, which is identical to the effects of PTEN α ^{CTG-R233*} (**Fig. 2J** in the revised manuscript). Our data thus demonstrate that the secretion characteristics of PTEN α hardly affects its immunoregulatory effects.

Fig 2. The Ptena mutant mice that are presented in this paper have not been fully characterized. How was the knockout performed? What effect does it have on Pten and Pten- α expression? Off target integration could lead to some of the findings presented, so ruling out these effects is needed to be sure that the effects seen are due to the intended genetic manipulation. Also, the possibility that the knockout leads paradoxically to increased Pten protein or an increase in the expression of other isoforms like M, N, etc, should be examined.

[Response] We thank the reviewer for this suggestion. We have reported the *Ptena* specific knockout mouse model in previous studies (PMID: 28636948 and 30926592), we thus cited the literature of Pan Wang, *et al* in the section of Materials and Methods. Rather than CRISPR-Cas9, we employed homologous recombination technology to establish *Ptena*-specific knockout mice. To further confirm the reliability of *Ptena*^{-/-} mice, we performed Whole-Exon Sequencing (WES) of *Ptena*^{-/-} mouse and its wildtype littermate control. As shown in **Response Table 1 and 2**, no off-target site was detected in *Ptena*^{-/-} mouse.

To ascertain whether loss of *Ptena* can affect the expression of other PTEN isoforms, we used anti-PTEN antibody to assess the protein level of PTEN isoforms in liver, colon and spleen from wildtype or *Ptena*^{-/-} mouse. As shown in **Response Figure 6**, deletion of *Ptena* hardly affected the expression of PTEN β or other PTEN isoforms.

Response Figure 6.

Indicated tissues from WT or *Ptena*^{-/-} mouse were harvested and subjected to western blot analysis with anti-PTEN antibody. * refers to unspecific band.

Response Table 1 and 2.

Mutations in wild-type (Response Table 1) and *Ptena*^{-/-} (Response Table 2) mice determined by whole exome sequencing were listed in the Tables. Tables are in the appendix files.

• *Prior work by others (PMID: 29246444) has shown that mutation of Ptena in mice causes a hyper-inflammatory response in the lung after an inflammatory stimulus. The authors should discuss their work as it relates to the prior literature as it supports the author's findings.*

[Response] We thank the reviewer for this suggestion. In our revised manuscript, we have discussed the work of Riquelme as the reviewer suggested.

• *TNF-α should be one of the cytokines measured*

[Response] As suggested, we measured the mRNA level of *TNFα* in liver or colon tissue from mice bearing Con A-induced hepatitis or DSS-induced colitis. As shown in Response Figure 7, the mRNA levels of *TNFα* were increased in *Ptena*^{-/-} mice as relative to their littermate wild-type controls under inflammatory conditions.

Response Figure 7.

(A) Wild-type and *Ptena*^{-/-} mice were subjected to DSS recovery mice model. The mice were treated with 5% DSS through drinking water for 6 days, followed by DSS-free drinking water treatment. Expression levels of indicated genes in the colons from the mice on day 8 of the DSS model were determined using qRT-PCR analysis

(n = 4 mice, mean ± SD, ** $P < 0.01$).

(B) Wild-type and *Ptena*^{-/-} mice were *i.v.* injected with 25 mg/kg Con A. The livers of the mice were harvested 5 hours post injection. Expression levels of indicated genes in the livers were assessed by qRT-PCR analysis (n = 3 mice, mean ± SD, * $P < 0.05$)

Figure 3. The effects on T cell function are quite interesting. The authors refer to prior literature without citing it.

[Response] As suggested, we have cited the literatures about the effects of chronic infection on T cell function.

Figure 6

What domains of PTEN α are needed to interact with and regulate stress granule formation? Can PTEN do the same thing?

[Response] It is reported that eIF2 α , G3BP1 and G3BP2 are key components of stress granule. We thus co-transfected PTEN or PTEN α with eIF2 α to determine whether PTEN is also involved in stress granule formation. As shown in **Figure 6D** in the revised manuscript, PTEN α rather than PTEN can interact with eIF2 α . Similar results were also detected by co-transfection of G3BP1 or G3BP2 with PTEN or PTEN α , respectively (**Figure S8** in the revised manuscript). Our data thus demonstrate that the N-terminal domain of PTEN α is essential for its association with stress granules.

The authors claim that PTEN α is a phosphatase for eIF2 α . However, there is no direct evidence showing that PTEN acts as a protein phosphatase for eIF2 α . This claim should be removed from the paper.

[Response] To further confirm that PTEN α can act as a *bona fide* phosphatase of eIF2 α , we employed an *in vitro* phosphatase assay (**Fig. 7B** in the revised manuscript). Through the use of anti-FLAG beads, we precipitated phosphorylated eIF2 α from H₂O₂-stimulated HEK293T cells transfected to express eIF2 α -FLAG. To obtain the PTEN α or control protein, we extracted PTEN α and control protein from HEK293T cells transfected with vector encoding PTEN α or empty vector. Both of them were extracted with anti-FLAG beads and eluted with FLAG peptide. Subsequently, PTEN α or control were separately incubated with a fusion of phosphorylated eIF2 α and anti-FLAG beads in phosphatase buffer. We then eluted bound proteins with FLAG peptide and subjected them to Western blot. As shown in **Figure 7B** in the revised manuscript, PTEN α directly interacted with eIF2 α and triggered eIF2 α dephosphorylation.

Reviewer #3 (Remarks to the Author): with expertise in tumor immune resistance

The manuscript titled “PTEN α functions as an immune suppressor and promotes immune resistance in PTEN-mutant cancer” is from one of the first groups that characterized the function of PTEN α (PTEN-long), a new PTEN isoform. By perturbing the expression of PTEN α in tumor cells and murine tissues, here, the authors aimed to dissect potential immunoregulatory roles of PTEN-alpha in tumor setting. The authors demonstrated distinct impacts of different mutations on the stability of PTEN isoforms and reduced infiltration of immune cells in PTEN α -expressing B16 tumors. Additionally, the authors used Ptena $^{-/-}$ mice to characterize the phenotype of this strain in DSS-induced colitis and LCMV infection setting. These results are interesting and highlight the involvement of PTEN α in regulating host immune responses. However, the authors failed to provide sufficient data to support several key conclusions. Moreover, the authors didn't comprehensively evaluate the activity of PTEN-controlled cell signaling pathways in the genetically modified cells and strains.

Therefore, the mechanisms by which PTEN α regulates host immune responses remain largely unclear. The paper also suffers from loose logic connections between sections in the results part and lack of justifications of multiple models used in this paper.

The major concerns are:

1. According to the title and the descriptions in the introduction section, the paper should be centered on the immunoregulatory roles of PTEN α in tumor setting. Except the figure 1, the results of the remaining six figures are from either inflammation or viral infection models. Especially, the authors concluded that the immunological importance of PTEN α is T- cell extrinsic, as shown in the figures 1F-H. It is unclear why the authors didn't focused on characterizing the impact of PTEN α expression on tumor immune microenvironment or tumor sensitivity to T cell killings. Defining the underlying mechanisms by which PTEN α -expressing B16 tumors display immune resistant phenotype are critical and more related to their title. The conclusion that PTEN α promotes cell resistance to T cell cytotoxicity is primarily based on the phenotype of LCMV-infected MEF cells. However, tumor cells have much more dysregulated intrinsic pathways when compared with MEF cells.

[Response] We are grateful for the reviewer's suggestion. In our revised manuscript, we mainly focused on the immunoregulatory role of PTEN α in tumor setting. For instance, we observed that less amounts of CD45 $^{+}$ immune cells were detected in PTEN α -expressing tumors as relative to control tumors by flow cytometry (**Fig. 2C** in the revised manuscript). To further study the status of immune cells in PTEN α -expressing tumor rather than inflammation, we thus substitute scRNA-seq analysis of CD45 $^{+}$ tumor-infiltrating immune cells for the experiments associated with hepatitis or colitis. Utilizing graph-based clustering to analyze the tumor-infiltrating immune cells, we identified 10 clusters for 10x data including 8605 cells. We then defined the clusters based on the exclusive expression of canonical marker genes. As shown in **Figure 2D** and **S2D** in the revised manuscript, major immune cell types in cancers including CD4 $^{+}$ T, Treg, CD8 $^{+}$ T, NK, monocyte, neutrophil, myeloid derived suppressor cells (MDSCs), B, plasma B and plasmacytoid dendritic cells (pDCs). Compared with more than 70% of lymphoid cells in control tumor, the percentages of

T lymphocytes and B lymphocytes were significantly reduced in PTEN α -expressing tumors (**Fig. 2E** and **2F** in the revised manuscript). Conversely, the ratio of innate immune cells, especially MDSCs, was increased in PTEN α -expressing tumors as relative to control tumors (**Fig. 2E** and **2F** in the revised manuscript). In light of the importance of CD8⁺ T cells in host antitumor immunity, we thus analyzed the differentially expressed genes in CD8⁺ T cells from PTEN α -expressing or control tumors. As shown in **Figure 2G**, in contrast to high level of genes related with T cell exhaustion including *Dusp2*, *Hif1a*, *Pdcd1*, *Havcr2* and *Lag3* in CD8⁺ T cells from PTEN α -expressing tumor, genes related with memory T cell formation including *Ccr7*, *Tcf7*, *Sell* and *Il7r* were selectively upregulated in CD8⁺ T cells from control tumor. Consistent with scRNA-seq results, subsequent flow cytometry revealed that overexpression of PTEN α limited the amounts of CD4⁺ and CD8⁺ infiltrated T cells, and enhanced PD-1 expression on CD8⁺ tumor-infiltrating T cells (**Fig. 2H** and **2I** in the revised manuscript). Our data thus indicate that PTEN α remains active in PTEN mutant cancers and subverts host immune attack.

As suggested, we have repeated all the experiments with B16 cells in our revised manuscript. As shown in **Figure 4F** in the revised manuscript, enforced expression of PTEN α in *Pten*^{-/-} B16 cells attenuated T cell-mediated cytotoxicity. Moreover, the amount of released ATP and the intracellular ROS level from PTEN α -expressing B16 cells (*Pten*^{-/-}) were decreased as compared with control cells did (**Fig. 4G** and **H** in the revised manuscript). Our data thus support the notion that PTEN α promotes tumor cell resistance to T cell cytotoxicity through blocking DAMPs release.

To further confirm the regulatory role of PTEN α in cancer cell response to oxidative stress, we used H₂O₂ to treat B16 cells (*Pten*^{-/-}) in presence or absence of PTEN α . As shown in **Figure 5A-C** in the revised manuscript, enforced expression of PTEN α increased cancer cell resistance to oxidative cell death. Furthermore, we used *Pten*^{-/-} B16 cell line to analyze the interactome of PTEN α . Consistent to the data in lung cancer cell line (H1299), a series of key components of stress granule including eIF2 α , G3BPs and PCBP1 were associated with PTEN α in B16 cells (*Pten*^{-/-}) (**Fig. 6A** in the revised manuscript). The association of PTEN α with G3BP1 in B16 cells was further confirmed by confocal assay (**Fig. 6E** in the revised manuscript). Moreover, similar to MEFs cells, enforced expression of PTEN α inhibits phosphorylation of eIF2 α in B16 cells (**Fig. 7A** in the revised manuscript). To determine whether PTEN α can also promotes protein synthesis in B16 cells, we employed puromycin incorporation assay in B16 cells in the treatment of H₂O₂. As shown in **Figure 7D** and **7E** in the revised manuscript, enforced expression of PTEN α in B16 cells ameliorated the translational inhibition induced by oxidative stress, which was analogue to the data using MEFs cells. Additionally, we also measured the levels of GPX4 and intracellular ROS in B16 cells. As shown in **Figure 7G** in the revised manuscript, enforced expression of PTEN α maintained GPX4 expression in B16 cells under oxidative stress. Conversely, the level of intracellular ROS was decreased in PTEN α -expressing B16 cells as relative to control cells (**Fig. 7H** in the revised manuscript).

In addition to the cancer vaccine model, we also employed B16 lung metastasis

model. As shown in **Figure S2G and S2H** in the revised manuscript, enforced expression of PTEN α increased pulmonary metastasis of B16 cells. In light of the inhibitory role of PTEN α in modulation of ferroptosis, our data thus support the work of Ubellacker JM, *et al.* (PMID: 32814895) that the sensitivity of tumor cells to oxidative stress limits their metastasis through blood.

Collectively, our data demonstrate that PTEN α restricts oxidative cell death and blocks T cell-mediated eradication of cancer.

2. Based on their previous work published in the 2017 Cell Reports paper, the authors substituted “the two CTG codons of mouse Pten α (CTG347 ... CTG362, which are equivalent to Homo sapiens CTG513) with GGA” to generate the “PTEN α -/- mice” used in this paper. In the published references, Pten α was reported to be expressed by tumor cells and brain tissues. To better explain how knockout of Pten α enhances ConA-induced hepatitis and DSS-induced colitis, and reduced susceptibility of LCMV-infected MEF to T cell cytotoxicity, the authors should characterize the level of Pten α expression in different types of normal cells or tissues, particularly in liver, colon and immune cells. If these tissues/cells express Pten α , the impact of Pten α knockout on the activity of signal pathways controlled by PTEN in these tissues/cells should be addressed, too.

[Response] We thank the reviewer for this suggestion. In our revised manuscript, we have added the expression data of PTEN α in multiple tissues. As shown in **Figure S3A** in the revised manuscript, the protein level of PTEN α was high in brain, lymph node and intestinal tissues; moderate in spleen, liver and lung; and low in kidney. We then used anti-PTEN antibody to further confirm the genetic deletion of *Pten α* in these tissues (**Response Figure 8A**).

To study whether loss of PTEN α can affect the PI3K-Akt signaling regulated by PTEN, we employed anti-phosph-AKT-S363 and anti-phosph-AKT-T308 antibodies to measure the phosphorylation level of AKT in spleen, liver and colon tissues from *Pten α ^{-/-}* mice and their wild-type littermate controls. As shown in **Response Figure 8B**, deletion of *PTEN α* hardly affected the PI3K-Akt signaling under the quiescent condition.

Response Figure 8.

(A) Indicated tissues from WT or *Ptena*^{-/-} mice were harvested and subjected to immunoblot analysis with anti-PTEN antibody. ‘*’ refers to unspecific band.

(B) Indicated tissues from WT or *Ptena*^{-/-} mice were harvested. Immunoblot analysis was performed using anti-phosph-AKT-S363 and anti-phosph-AKT-T308 antibodies. Gray values of pAKT-S473, pAKT-T308 and AKT were determined, and the ratios were used for statistical analysis (n = 4 mice, mean ± SD, ns, not significant). KO refers to *Ptena*^{-/-} mice.

3. It has been reported that PTENα can be secreted from cells and has the potential to antagonize the PI3K signaling in adjacent cells. Given that the function from secreted PTENα can also shape tumor immune microenvironment, the author should determine whether overexpression or knockout of PTENα in cells or mice change the level of secreted PTENα.

[Response] To ascertain whether the immunosuppressive role of PTENα is dependent on its secretion, we thus employed the vector of PTENα^{CTG-Δ6A} that has been reported to abolish secretion of PTENα (PMID: 23744781). Utilizing the cancer vaccine

model, we found that overexpression of PTEN $\alpha^{\text{CTG-}\Delta 6\text{A}}$ also promoted cancer immune escape, which is identical to the effects of PTEN $\alpha^{\text{CTG-R233}^*}$ (**Fig. 2J** in the revised manuscript). Our data thus demonstrate that the secretion characteristics of PTEN α hardly affects its immunoregulatory effects.

4. Several experiments are missing important experimental aims. In the figure 1C, the authors only compared the difference between patients with stop gained mutations and phosphatase-inactive mutations. The authors should include the group without any PTEN mutations. The supplementary Figure 4 (especially for S4A) is important figure. The authors should consider to include it in Figure 3. Additionally, the author should provide the data from wild-type mice receiving the BM transfer from either WT or PTEN α -/- mice in S4A. Moreover, does PTEN α knockout alter the proliferation and survival of cytotoxic T cells and helper T cells? Lastly, why was the analysis of figure 1B limited in UCEC patients?

[Response] As suggested, we have analyzed the survival of patients with or without *PTEN* mutations in UCECs. Unexpectedly, TCGA database revealed that patients with *PTEN* mutations had a better prognosis than those with wild-type *PTEN* (**Response Fig. 9A**). It is reported that the frequency of somatic mutations in *PTEN* and *P53* are mutually exclusive in cancer progression (PMID: 12379854). We thus compared the DNA mutations in cancers with or without *PTEN* mutations. As shown in **Response Figure 9B**, the frequency of *p53* mutation was significant higher in wild-type *PTEN* cancers than those in *PTEN*-mutant cancers. Moreover, the mutation of *p53* is associated with a worse prognosis of patients with cancer (**Response Fig. 9C**). We therefore speculate that the mutually exclusive recurrent *PTEN* and *p53* mutations results in the better prognosis of patients with *PTEN* mutations.

To avoid the effects due to *p53* mutations, we interrogated *PTEN* mutation data from Metastatic Colorectal Cancer database (MSKCC, Cancer Cell 2018) and Breast Cancer database (METABRIC, Nature 2012 & Nat Commun 2016). As shown in **Figure S1A** in the revised manuscript and **Response Fig. 9D**, compared with patients without *PTEN* mutations, the outcome of patients with stop-gained mutations were worse. Notably, less effects of mutually exclusive mutations between *p53* and *PTEN* were detected in these databases (**Response Fig. 9E and 9F**).

We are grateful for the reviewer's suggestion and we have moved the **Figure S4A** into **Figure 3**.

As suggested, wild-type mice were lethally irradiated and transplanted with wild-type or *Ptena*^{-/-} bone marrow (BM) and subsequently infected with LCMV-C113. As shown in **Response Figure 10A**, 30 days following LCMV Clone 13 infection, compared with *Ptena*^{-/-} donor mice, less amounts of virus-specific T cells were detected in lung from wild-type recipient mice adoptive transferred with wild-type or *Ptena*^{-/-} BM. Consistent with the less amount of virus-specific T cells, higher level of inhibitory receptors and limited effector cytokine production were also detected in virus-specific T cells from wild-type recipient mice as relative to their littermate control did (**Response Figure 10B and 10C**). Similar results were also detected in spleens from *Ptena*^{-/-} donor mice and wild-type recipient mice (**Fig. 3F**,

S5B and S5D in the revised manuscript). Our data thus demonstrate that *PTEN* α exerts immunosuppressive effects in a T cell-extrinsic manner.

In order to determine whether loss of *PTEN* α affects the proliferation and survival of cytotoxic T cells and helper T cells, we firstly employed CFSE staining assay to measure the T cell proliferation during activation. As shown in **Figure S3B** in the revised manuscript, deletion of *PTEN* α elicited little effects on T cell expansion during T cell activation. Besides, identical ratio of Treg cell, Th1, Th17 and Tr1 cells were induced under various T-cell polarizing conditions between wild-type and *Ptena*^{-/-} mice (**Fig. S3C** in the revised manuscript). Finally, we used Annexin V/7-AAD staining to assess cell death during T cell activation. As shown in **Figure S3D** in the revised manuscript, no significant difference was detected between wild-type and *Ptena*^{-/-} T cells. Collectively, our data demonstrate that deletion of T cell-intrinsic *Ptena* hardly affects T cell fate.

Response Figure 9. The relationship of *PTEN* mutations with the outcome of patients with UCEC.

(A) Clinical data of uterine corpus endometrial carcinoma (UCEC) patients with *PTEN* mutations were acquired from the TCGA database, and used for drawing of survival curve (*PTEN*-mutant, n = 352; *PTEN*-wildtype, n = 176, ***p* < 0.01).

(B) Analysis of DNA mutation frequency between *PTEN*-mutant and *PTEN*-wt cancers from TCGA database.

(C) Clinical data of uterine corpus endometrial carcinoma (UCEC) patients with *p53* mutations were acquired from the TCGA database, and used for drawing of survival curve (*TP53*-mutant, n = 194; *TP53*-wildtype, n = 334, ***p* < 0.01).

(D) Clinical data of breast cancer (BRCA) patients with *PTEN* mutations were acquired from the Breast Cancer database (METABRIC, Nature 2012 & Nat Commun 2016), and used for drawing of survival curve (WT, n = 1850; stop gained mutations, n = 25; phosphatase-inactive mutations, n = 6, **p* < 0.05).

(E) Analysis of DNA mutation frequency between *PTEN*-mutant and *PTEN*-wt

cancers from Metastatic Colorectal Cancer database (MSKCC, Cancer Cell 2018).

(F) Analysis of DNA mutation frequency between *PTEN-mutant* and *PTEN-wt* cancers from Breast Cancer database (METABRIC, Nature 2012 & Nat Commun 2016).

Response Figure 10. *PTENα* drives T cell exhaustion in lung in a T cell-independent manner.

(A-C) Bone marrow from wild-type and *Ptenu*^{-/-} mice were transplanted into *Ptenu*^{-/-} mice, respectively. 30 days after transplantation, mice were *i.v.* infected with 5×10^5 PFU LCMV-C113. On day 30 post infection, lungs were harvested from the mice, and lymphocytes were isolated.

(A) Percentages of CD8⁺ GP33-41⁺ cells were determined by flow cytometry analysis (n = 3 mice, mean ± SD, ns, not significant, **P* < 0.05).

(B) Expression levels of PD-1, TIM-3 and LAG-3 on the CD8⁺ GP33-41⁺ cells were determined by flow cytometry analysis (n = 3 mice, mean ± SD, ns, not significant, ***P* < 0.01, ****P* < 0.001).

(C) The lymphocytes were treated with 2 µg/ml GP33-41 peptide for 5 hours, and the production of TNFα and IFNγ of the CD8⁺ GP33-41⁺ cells were determined by flow cytometry analysis (n = 3 mice, mean ± SD, ns, not significant, ****P* < 0.001).

Statistical significance was assessed by one-way ANOVA followed by Tukey's multiple comparisons test (A-C).

Reviewers' Comments:

Reviewer #1:

Remarks to the Author:

The authors had addressed most concerns raised previously, however, the following two major concerns are still not addressed:

1. Instead of vaccine model, cancer immunotherapy models including anti-PD-1 or adoptive T-cell transfer should be used to indicate the immune escape.
2. The connections between Fig.5 and Figures 6, 7 are still very weak. The previous concern (The author should manipulate eIF2a or GPX4 expression in PTEN^{-/-} PTEN^{+/+} B16 tumor cell to test their effects on tumor growth and tumor immune response) is not responded. There is no data to show whether p-eIF2a and GPX4 regulated by PTEN⁺ contributes to the resistance to CD8⁺ T cell-mediated killing.

Besides, there are some other concerns:

- 1) Although the author showed PTEN^{-/-} MEF are more sensitive to ferroptosis, there is not enough evidence to conclude PTEN⁺ regulates ferroptosis especially mediated by classical ferroptosis inducers (erastin, RSL3) in B16 cells, since it is the only tumor model used in the whole paper.
- 2) The conclusion that PTEN⁺ inhibits tumor cell death mediated by effector T cells is still not definitive. The co-culture of T cell and tumor cell is a complicated system. In Fig.4F-G, author tried to claim that PTEN⁺ impaired T cell-mediated killing on target tumor cells, however, the methods including CCK8, ATP release and DCFDA staining are unable to distinguish tumor and T cells in the coculture system. It's possible that dead T cells interfere with the results.
- 3) PI staining to detect cell death is usually performed on unfixed live cells. Here the authors performed on fixed cell, which is mainly used for cell cycle analysis. And based on the plot of PI staining in manuscript, it seems that effector T cells or H₂O₂ treatment mainly cause cell cycle arrest in tumor cells.
- 4) What is the conclusion for Figure 5A? There is no obvious cell death and It's not consistent with Fig. 5B.
- 5) There is still a logical issue between clinical significance and experimental models. As stated in Fig.1, PTEN⁺ mutation shows clinical relevance in UCEC, but later on all experiments are done in B16, a mouse melanoma cell line.

Reviewer #2:

Remarks to the Author:

The authors have carefully responded to nearly all of the reviewer 2 criticisms and comments from the first review. Similarly the response to the other reviewers has been quite extensive.

Reviewer #3:

Remarks to the Author:

All my questions are appropriately addressed.

Reviewer #4:

Remarks to the Author:

In their paper, Sun et al studied the role of Ptena (aka Ptena/Pten-L/Pten-Long), a Pten splice variant that bears an N-terminal extension, in cancer prognosis and anti-tumor and anti-viral immune responses.

In many human cancers, PTEN displays phosphatase-inactivating and/or stop-gaining mutations, and the authors observed that the latter are associated with worse prognosis and reduced expression of immune response signatures in patients with uterine corpus endometrial carcinoma and colorectal cancer.

The authors found that stop-gaining mutations result in protein instability of canonical PTEN but not PTENa, which they hypothesize, has immune-suppressive function. It has been previously

shown that Ptena KO mice display increased immunopathology in a *P. aeruginosa* infection model (<https://doi.org/10.1016/j.immuni.2017.11.010>) but its role in viral infections and cancer is unknown.

The authors showed in a B16 cancer vaccine model that, compared to Ptena-null tumors, Ptena-expressing tumors grow faster, display lower immune cell infiltration, a lower ratio of lymphocytes to myeloid cells, and higher expression of exhaustion-related genes among CD8 TILs after vaccination; and conclude that Ptena-expression in tumors is associated with an impaired anti-tumor immune response.

In a chronic infection model using LCMV cl13, the authors show that Ptena KO mice displayed lower viral loads and higher frequency of virus-specific CD8 T cells in lungs, and that these T cells displayed lower exhaustion levels.

In addition, the authors provide evidence that Ptena has no intrinsic effect on T cells, but rather on cancer cells' death resistance. In vitro, (GP33 peptide-pulsed) B16 cells expressing Ptena displayed a higher susceptibility to T cell killing.

Finally, the authors show that Ptena increases cancer cell resistance to immunogenic cell death and H₂O₂ oxidative stress, by neutralizing intracellular ROS and impairing stress granule formation during oxidative stress.

The results are very interesting, novel, and highlight a potentially relevant role of the splice variant Ptena in tumor resistance to immune control.

I have the following major comments:

Evidence that tumor-specific CD8 TILs are more exhausted in Ptena-expressing tumors is not presented, and therefore, the claim that "PTENa promotes T cell exhaustion" or "presence of PTENa leads to T cell dysfunction" might be misleading.

Because T cell tumor-specificity was not evaluated, the in vivo data only demonstrate that Ptena is associated with 1) reduced TIL infiltration, and 2) higher levels of T cell activation/exhaustion markers "on average".

Perhaps the latter is not explained by tumor-specific T cells being more exhausted, but because of the reduced T cell infiltration making the pool of tumor-specific CD8 TILs (which express activation/exhaustion markers Pdc1, etc) account for a larger proportion of total TIL.

For instance, in Fig 2I, what is the percentage of PD1+ cells among CD8 TILs? (only mean fluorescence is shown)

In addition, Ptena expression is associated with higher tumor size and reduced CD8 T cell infiltration, therefore it impacts antigen to T cell ratio, and higher antigen dose can lead to higher exhaustion of Ag-specific T cells. Therefore, the higher levels of exhaustion might be a direct consequence of the reduced amounts of CD8 T cells. Can the authors clarify this?

Similarly, for LCMV chronic infection, Ptena KO mice present lower viral loads, which could explain the lower CD8 T cell exhaustion due to reduced antigen dose (e.g. <https://www.ncbi.nlm.nih.gov/pmc/articles/PMC4995073/>).

In other words, is there any evidence to claim that Ptena induces T cell exhaustion, independently from the reduction of antigen dose? Otherwise, it might be useful to discuss this.

The scRNA-seq experiment is insufficiently described and the analysis is not fully convincing.

i) The current interpretation that increased expression of activation/exhaustion markers (Pdc1, lag3, Havcr2, Hif1a, etc) and reduced expression of memory genes (Tcf7, Sell, Il7r etc) is linked to immune escape is not very convincing. Most tumor-specific CD8 TILs express Pdc1 and do not express naive/central memory genes Sell/Il7r, therefore, most likely, these differences are largely due to comparing tumor-specific vs non-specific T cells.

Consider that a substantial part of CD8 TILs in B16 tumors might not be tumor-specific, and even include naive T cells (e.g. <https://www.ncbi.nlm.nih.gov/pmc/articles/PMC2916130/>, <https://pubmed.ncbi.nlm.nih.gov/32313720/>)

To help interpreting these data, additional information should be shown, including the full table of differential gene expression analysis of CD8 TILs in mock vs Ptena is missing and should be provided.

In particular it would be key to see expression of cytotoxicity genes such as Tnfa, Ifng and Gzmb.

ii) Figure S2D does not allow verifying the cluster annotations. Which genes are expressed in the CD4 or CD8 T cell clusters? For instance, Cd4 and Gzmb seem to be expressed in both CD4 and CD8 clusters. "Treg genes" module seem to be equally expressed in CD4 and CD8 clusters, or even more highly expressed in CD8.

It would be important to show key cell type marker genes expression individually in each cluster, both as boxplot/violin plots and on the tSNE.

iii) There is insufficient description of the scRNA-seq experiments, answers to all these questions should be provided: which samples were used for scRNA-seq? How were these samples obtained and processed? From how many animals, at which timepoints, which average tumor sizes? Where the samples grouped into separate sequencing batches? How were cells sorted (FACS panel) ? how single-cell RNA-seq libraries were prepared? (protocol?)

iv) Where are these data available (both raw and processed)? These should be made publicly available in a repository such as NCBI GEO

v) Please indicate how the expression values shown in eg Fig 2G were calculated. In addition, data integration/batch-effect correction is a major transformation that was applied to these data and it might be important to mention it in the main text.

vi) In the methods a UMAP representation is described, but the plot is not shown

vii) In Figure 2G, it would be also important to show 'control genes' including Cd2, Cd3*, Cd8a, Cd8b, Cd4, Foxp3, to demonstrate that these two groups are equally well represented in terms of CD8 T cells. This is particularly important because batch-effect correction was applied to the data, which can introduce artifacts and mix different cell type populations.

Can the authors confirm that in vitro T cell killing of B16 cells is MHC-I mediated?

While the evidence for increased resistance to oxidative stress by Ptena expression in cancer cells seems convincing, and contribute to increased resistance to CTL killing, the fact that Ptena-expressing tumors are associated with reduced immune cell infiltration (CD45+) and T cell infiltration in particular, both in human samples (TCGA data) and in murine models, is not directly discussed.

Reduction of immunogenic cell death in Ptena-expressing tumors might explain the reduced T cell infiltration in vivo. Is this the authors' model? What evidence is supporting that? For instance, can they show that the increased growth of tumors treated with ferroptosis inhibitor NP-VE (i.e. where Ptena expression is irrelevant, Fig 5 F) is CD8 T cell dependent? (due to higher CD8 T cell infiltration levels?)

Minor comments

In Figure 1 C, for the GSEA it is recommended to use the full ranked gene list for measuring enrichment of a gene set instead of defining a threshold (here the top phosphate-inactive and the top stop-gain mutation correlated genes were chosen). See for example

https://software.broadinstitute.org/cancer/software/gsea/wiki/index.php/FAQ#What_is_the_difference_between_GSEA_and_an_overlap_statistic_.28hypergeometric.29_analysis_tool.3F).

Otherwise, please justify the use of a subsetted ranked gene list for GSEA. Also in the heatmap, "SLANF7" is not a valid gene name.

The following sentences should be corrected:

"Durable benefit, however, this therapy is limited to a minority of patients with cancer, and the clinical efficacy is often compromised along with tumor development"

"which is analogue to phosphatase-inactive mutations do"

"We next sought to determine whether T cell polarization is depends on T cell-intrinsic PTEN α ."

We are grateful to the reviewers for their comments and suggestions. During the past two months, we have generated a large amount of data and made a revision of the paper based on new data. The following is our point-by-point response to the reviewers' comments and questions.

A. Point-by-point responses to the comments by Reviewer #1

Reviewer #1 (Remarks to the Author):

The authors had addressed most concerns raised previously, however, the following two major concerns are still not addressed:

We would first like to express our sincere thanks to the reviewer for the positive comments of our study. We found the reviewer's comments to be very helpful and of value for improving the quality of our manuscript. We have addressed each of the comments as follows.

1. Instead of vaccine model, cancer immunotherapy models including anti-PD-1 or adoptive T-cell transfer should be used to indicate the immune escape.

[Response] We used the adoptive T-cell transfer assay to further assess the role of PTEN α in cancer immune escape. Mock or PTEN α -expressing B16-*Pten*^{-/-} cells were infected with LCMV-C113, and subcutaneously injected into nude mice. 10 days later, the activated CD8⁺ T cells were harvested from spleen in mice infected with LCMV-C113 on day-7 post infection, and subsequently transferred into the tumor-bearing nude mice. Similar procedures were repeated on day-14 and day-18 in B16 tumor-bearing mice (**Response Fig. 1A**). As shown in **Response Figure 1B**, viral infection hardly influenced tumor growth of Mock or PTEN α -expressing B16 cells in nude mice, and ectopic expression of PTEN α elicited no effects on cell proliferation, which is consistent to the data in C57BL/6 mice (**Response Fig. 1B**). Moreover, the transferred CD8⁺ T cells exhibited significantly antitumor effects on Mock rather than PTEN α -expressing B16 cells in nude mice (**Response Fig. 1B**). Notably, through analysis of CD8⁺ T cells infiltrating in spleens or tumors, we found that the amount of adoptive transferred CD8⁺ T cells were identical between the nude mice with PTEN α -expressing B16 cells and those with Mock B16 cells (**Response Fig. 1C**). Taken together, our data demonstrate that PTEN α blocks T cell-mediated cancer eradication and promotes cancer immune escape.

Accordingly, the main text information has been updated in the section of "Results" (Page 8, Line 16-22, and Page 9, 1-8).

Response Figure 1

Response Figure 1.

(A-C) Mock or PTEN α expressing B16-*Pten*^{-/-} cells were infected with 1 MOI LCMV-C113, and subcutaneously injected into the right flank of Balb/c nude mice. CD8⁺ T cells were harvested from spleens of C57BL/6 mice *i.v.* infected with 1×10^6 PFU LCMV-C113 on day 7 post infection. 10, 14 and 18 days after tumor inoculation, the CD8⁺ T cells were transferred into the tumor bearing nude mice through intravenous injection.

(A) An illustration of mice treatment. Also shown in **Figure 2h** in the manuscript.

(B) Tumor volumes were monitored overtime (Mock, n = 6 mice; PTEN α , n = 7 mice, mean \pm s.e.m, **P* < 0.05). Statistical significance between CD8⁺ T cells transferred groups were shown. Also shown in **Figure 2i** in the manuscript.

(C) On day 24 post tumor inoculation, immune cells were isolated from spleens and tumors of the nude mice, and subjected to flow cytometry analysis. Percentages of CD8⁺ T cells in CD45⁺ immune cells were shown (Mock, n = 3 mice; PTEN α , n = 4 mice, mean \pm SD, ns, not significant). Also shown in **Figure S3g** in the manuscript.

Statistical significance was assessed by two-tailed unpaired Student's t test (B and C). Data are pooled from two (B) independent experiments.

2. The connections between Fig.5 and Figures 6, 7 are still very weak. The previous concern (The author should manipulate eIF2 α or GPX4 expression in PTEN^{-/-} PTEN α ^{+/+} B16 tumor cell to test their effects on tumor growth and tumor immune response) is not responded. There is no data to show whether p-eIF2 α and GPX4 regulated by PTEN α contributes to the resistance to CD8⁺ T cell-mediated killing.

[Response] As suggested, we used shRNA to knockdown the endogenous Gpx4

expression in *Pten*^{-/-} B16 cells, subsequently stable transfected with PTEN α (**Response Fig. 2A**). To assess the role of PTEN α in T cell-mediated cytotoxicity, we first used higher concentration of CFSE to stain the viral-peptide pulsed B16 cells and low dose of CFSE to stain untreated B16 cells, followed by co-culturing with virus-specific CD8⁺ T cells. As shown in **Response Figure 2B**, ectopic expression of PTEN α increased cancer resistance to CD8⁺ T cell attack. Furthermore, loss of Gpx4 significantly increased the sensitivity of B16 cells to T cell killing with or without PTEN α (**Response Fig. 2B**), which indicate PTEN α elicited its immunoregulatory effects in a Gpx4-dependent manner. To further confirm this result *in vivo*, we also employed murine tumor vaccination model. As shown in **Response Figure 2C**, loss of Gpx4 neutralized the immunosuppressive effects of PTEN α and remarkably increased the effectiveness of tumor vaccination. Our data thus demonstrate that Gpx4 largely contributes to the stimulatory role of PTEN α in cancer immune escape.

Accordingly, the main text information has been updated in the section of “Results” (Page 18, Line 20-22, and Page 19, 1-4).

Response Figure 2

A

B

C

Response Figure 2.

(A-C) Endogenous Gpx4 expression in B16-*Pten*^{-/-} cells were knockdown using shRNA targeting Gpx4 or scramble shRNA. Then the cells were transfected to express Mock or PTEN α .

(A) Immunoblot analysis of PTEN α and Gpx4 expression in the cells with anti-PTEN and anti-Gpx4 antibodies, respectively.

(B) The B16 cells were pulsed with 2 μ g/ml GP₃₃₋₄₁ peptide or untreated, and stained with 2 μ M or 0.2 μ M CFSE, respectively. CD8⁺ T cells were harvested from spleens of LCMV-Cl13 infected C57BL/6 mice on day 7 post infection, and co-cultured with the B16 cells at the ratio of 15:1. B16 cells were collected and subjected to flow cytometry analysis. Percent cytotoxicity was calculated using the formula: Percent cytotoxicity (%) = (1 - CFSE^{hi} cells/CFSE^{low} cells) \times 100%. (n = 3 cell cultures, mean \pm SD, ****P* < 0.001, *****P* < 0.0001).

(C) C57BL/6 mice were immunized with irradiated B16 cells, and subcutaneously injected with 1 \times 10⁶ indicated B16 cells. The tumor volumes were monitored overtime (Scramble, n = 5 mice; shRNA, n = 6 mice, mean \pm s.e.m, **P* < 0.05). Statistical significance between scramble groups were shown. Also shown in **Figure 7i** in the manuscript.

Statistical significance was assessed by one-way ANOVA followed by Tukey's multiple comparisons test (B) or two-tailed unpaired Student's t test (C).

Besides, there are some other concerns:

1) Although the author showed PTEN α -/- MEF are more sensitive to ferroptosis, there is no enough evidence to conclude PTEN α regulates ferroptosis especially mediated by classical ferroptosis inducers (erastin, RSL3) in B16 cells, since it is the only tumor model used in the whole paper.

[Response] As suggested, we used ferroptosis inducer RSL3 to treat the Mock or PTEN α expressing B16-*Pten*^{-/-} cells. As shown in **Response Figure 3**, ectopic expression of PTEN α increased cell resistance to RSL3 treatment as compared with Mock B16 cells, which is consistent with the data in MEFs.

Accordingly, the main text information has been updated in the section of "Results" (Page 14, Line 11-13).

Response Figure 3

Response Figure 3.

Mock or PTEN α expressing B16-*Pten*^{-/-} cells were treated with 1 μ M RSL3 for 12 hours, and stained with PI. The cells were subjected to flow cytometry analysis, and the death cell rates were used for statistical analysis (n = 3 cell cultures, mean \pm SD, ****P* < 0.001). UT, untreated. Also shown in **Figure S8g** in the manuscript.

Statistical significance was assessed by two-tailed unpaired Student's t test.

2) The conclusion that PTEN α inhibits tumor cell death mediated by effector T cells is still not definitive. The co-culture of T cell and tumor cell is a complicated system. In Fig.4F-G, author tried to claim that PTEN α impaired T cell-mediated killing on target tumor cells, however, the methods including CCK8, ATP release and DCFDA staining are unable to distinguish tumor and T cells in the coculture system. It's possible that dead T cells interfere with the results.

[Response] We thank the reviewer for this question. Inspired by experimental design of *in vivo* killing assay, we used high doses of CFSE to stain the viral-peptide pulsed B16-*Pten*^{-/-} cells and low dose of CFSE to stain untreated B16-*Pten*^{-/-} cells, followed by co-culturing with virus-specific CD8⁺ T cells. Cytotoxicity was calculated by comparing the remaining CFSE^{hi} and CFSE^{low} cells. As shown in **Response Figure 4**, ectopic expression of PTEN α increased B16 cell resistance to CD8⁺ T cell attack. Our data thus demonstrate that presence of PTEN α promotes cancer immune resistance without affecting T cell viability or function.

Response Figure 4

Response Figure 4.

Mock or PTEN α expressing B16-*Pten*^{-/-} cells were pulsed with 2 μ g/ml GP₃₃₋₄₁ peptide or untreated, and stained with 2 μ M or 0.2 μ M CFSE, respectively. C57BL/6 mice were *i.v.* infected with LCMV-C113, and CD8⁺ T cells were harvested from spleens of the mice on day 7 post infection, using as effector. The CD8⁺ T cells were co-cultured with the B16 cells at the ratio of 15:1 for 20 hours. Percent cytotoxicity was calculated using the formula: Percent cytotoxicity (%) = (1 - CFSE^{hi} cells/CFSE^{low} cells) \times 100%. (n = 3 cell cultures, mean \pm SD, ***P* < 0.01). Statistical significance was assessed by two-tailed unpaired Student's t test.

3) PI staining to detect cell death is usually performed on unfixed live cells. Here the authors performed on fixed cell, which is mainly used for cell cycle analysis. And based on the plot of PI staining in manuscript, it seems that effector T cells or H₂O₂ treatment mainly cause cell cycle arrest in tumor cells.

[Response] Indeed, PI staining can also be used to analyze cell death (PMID: 17406435). Instead of the peak of cells with normal (diploid) DNA content (which is analyzed in cell cycle analysis), we analyzed the sub-G1 peak, which represents dying cells. To further confirm these results, we also used PI staining on unfixed live cells to measure the late stage of cell death. C57BL/6 mice were primed with LCMV to induce virus-specific T cells, which subsequently co-cultured with virus-infected or un-infected target cells in presence or absence of PTEN α . As shown in **Response Figure 5**, ectopic expression of PTEN α limited T cell-mediated cytotoxicity rather than cell cycle progression.

Response Figure 5

Response Figure 5.

CD8⁺ T cells were isolated from the spleens of mice untreated or *i.v.* infected with 1×10^6 LCMV-C113 for 7 days, and used as the effector cells for *in vitro* cytotoxicity assay. Mock or PTEN α expressing B16-*Pten*^{-/-} cells were pulsed with GP₃₃₋₄₁ peptide and used as target cells. Effector cells and target cells were incubated at indicated ratios. 20 hours later, the cells were harvested, and directly stained with PI (Invitrogen, following manufactory's instruction), followed by flow cytometry analysis. Percentages of PI⁺ cells in CD8⁺ cells were used for statistical analysis (n = 3 cell

cultures, mean \pm SD, $**P < 0.01$).

Statistical significance was assessed by two-tailed unpaired Student's t test.

4) What is the conclusion for Figure 5A? There is no obvious cell death and It's not consistent with Fig. 5B.

[Response] We regret the irregularity of this display noted by the reviewer, and this has been corrected in the revised version (**Response Figure 6**).

Response Figure 6

Response Figure 6.

Microscopy analysis of H₂O₂ (500 μM) treated Mock or PTENα expressing B16-*Pten*^{-/-} cells. UT, untreated. Also shown in **Figure 5a** in the manuscript.

5) There is still a logical issue between clinical significance and experimental models. As stated in Fig.1, PTENα mutation shows clinical relevance in UCEC, but later on all experiments are done in B16, a mouse melanoma cell line.

[Response] The following effects were made to address this concern. In addition to UCEC, PTENα mutations also exhibit clinical relevance in colon cancers. We thus employed CT26 cells (murine colorectal cancer cell line) to assess the role of PTENα in tumor vaccination model. We first used shRNA to knockdown the endogenous PTEN expression in CT26 cells, and subsequently stably transfected with the vector encoding Mock or PTENα (**Response Fig. 7A**). In accordance with the data in B16 cells, ectopic expression of PTENα elicited immunosuppressive effects and promoted tumor development in immunized mice (**Response Fig. 7B**). Taken together, our data identify the regulatory role of PTENα in host antitumor immunity.

Accordingly, the main text information has been updated in the section of "Results" (Page 9, Line 20-22, and Page 10, 1-5).

Response Figure 7

A

B

Response Figure 7.

(A and B) PTEN expression in CT26 cells were knocked down by shRNA targeting PTEN, and then the cells were transfected with empty vector or vector encoding PTEN α .

(A) Immunoblot analysis of PTEN α expression in the CT26 cells. Also shown in **Figure S3i** in the manuscript.

(B) Balb/c mice were untreated or subcutaneously injected with 1×10^5 irradiated CT26 cells in the left flank. 10 days later, the mice were re-injected with 1×10^6 Mock or PTEN α CT26-sh-PTEN cells, and the tumor volumes were monitored overtime (Unimmunized, n = 5 mice; Immunized-Mock, n = 7 mice; Immunized-PTEN α , n = 6 mice, mean \pm s.e.m, * $P < 0.05$). Statistical significance between immunized groups were shown. Also shown in **Figure S3j** in the manuscript.

Statistical significance was assessed by two-tailed unpaired Student's t test (B).

B. Point-by-point responses to the comments by Reviewer #2

Reviewer #2 (Remarks to the Author):

The authors have carefully responded to nearly all of the reviewer 2 criticisms and comments from the first review. Similarly the response to the other reviewers has been quite extensive.

[Response] We are grateful for the reviewer's positive comments and affirmation of our study.

C. Point-by-point responses to the comments by Reviewer #3

Reviewer #3 (Remarks to the Author):

All my questions are appropriately addressed.

[Response] We appreciate the reviewer's appreciation of our study.

D. Point-by-point responses to the comments by Reviewer #4

Reviewer #4 (Remarks to the Author): with expertise in scRNAseq, T cells, LCMV

In their paper, Sun et al studied the role of Ptena (aka Ptena/Pten-L/Pten-Long), a Pten splice variant that bears an N-terminal extension, in cancer prognosis and anti-tumor and anti-viral immune responses.

In many human cancers, PTEN displays phosphatase-inactivating and/or stop-gaining mutations, and the authors observed that the latter are associated with worse prognosis and reduced expression of immune response signatures in patients with uterine corpus endometrial carcinoma and colorectal cancer.

*The authors found that stop-gaining mutations result in protein instability of canonical PTEN but not PTENa, which they hypothesize, has immune-suppressive function. It has been previously shown that Ptena KO mice display increased immunopathology in a *P. aeruginosa* infection model (<https://doi.org/10.1016/j.immuni.2017.11.010>) but its role in viral infections and cancer is unknown.*

The authors showed in a B16 cancer vaccine model that, compared to Ptena-null tumors, Ptena-expressing tumors grow faster, display lower immune cell infiltration, a lower ratio of lymphocytes to myeloid cells, and higher expression of exhaustion-related genes among CD8 TILs after vaccination; and conclude that Ptena-expression in tumors is associated with an impaired anti-tumor immune response.

In a chronic infection model using LCMV cl13, the authors show that Ptena KO mice displayed lower viral loads and higher frequency of virus-specific CD8 T cells in lungs, and that these T cells displayed lower exhaustion levels.

In addition, the authors provide evidence that Ptena has no intrinsic effect on T cells, but rather on cancer cells' death resistance. In vitro, (GP33 peptide-pulsed) B16 cells expressing Ptena displayed a higher susceptibility to T cell killing.

Finally, the authors show that Ptena increases cancer cell resistance to immunogenic cell death and H₂O₂ oxidative stress, by neutralizing intracellular ROS and impairing stress granule formation during oxidative stress.

The results are very interesting, novel, and highlight a potentially relevant role of the splice variant Ptena in tumor resistance to immune control.

We are thankful to the reviewer's comments and instructions for improving our data and manuscript. In the revised version, results have been better presented to be in accordance with publication standards, and figures have been rearranged to repair illegibility. We have done an additional series of experiments and addressed all the comments as below:

I have the following major comments:

Evidence that tumor-specific CD8 TILs are more exhausted in Ptena-expressing tumors is not presented, and therefore, the claim that "PTEN α promotes T cell exhaustion" or "presence of PTEN α leads to T cell dysfunction" might be misleading.

Because T cell tumor-specificity was not evaluated, the in vivo data only demonstrate that Ptena is associated with 1) reduced TIL infiltration, and 2) higher levels of T cell activation/exhaustion markers "on average".

Perhaps the latter is not explained by tumor-specific T cells being more exhausted, but because of the reduced T cell infiltration making the pool of tumor-specific CD8 TILs (which express activation/exhaustion markers Pcd1, etc) account for a larger proportion of total TIL.

For instance, in Fig 2I, what is the percentage of PD1+ cells among CD8 TILs? (only mean fluorescence is shown)

[Response] In light of the reviewer's suggestion, we performed the scRNAseq with two replicates, and identified 12 clusters for 10X data based on graph-based clustering (**Response Fig. 8A**). To further confirm the stability and accuracy of our data, we examined the canonical marker genes related with major cell populations, including T cells, B cells and other diverse myeloid-lineage cells (**Response Fig. 8B**). In accordance with the flow cytometry data, the amounts of lymphocytes were reduced in PTEN α -expressing tumors as relative to control tumors (**Response Fig. 8C**).

To uncover the potential functional subtypes of CD8⁺ T cell population, we performed unsupervised clustering and identified 4 clusters for CD8⁺ T cells, each with its unique signature genes (**Response Fig. 8D** and **8E**). In contrast to other 3 clusters, the fourth cluster of CD8⁺ T cells, CD8T-Ctla4, expressed high levels of exhaustion markers *Ctla4*, *Pcd1*, *Lag3* and *Hif1 α* , thus representing exhausted CD8⁺ T cells (**Response Fig. 8E**). As expected, the percentage of C4_CD8T-Ctla4 was remarkably increased in PTEN α -expressing tumors as compared with control tumors (**Response Fig. 8F** and **8G**). Our data thus demonstrate that presence of PTEN α promotes tumor-reactive T cell dysfunction.

Accordingly, the main text information has been updated in the section of "Results" (Page 7, Line 11-22, and Page 8, 1-8).

Response Figure 8

A

B

C

D

E

Response Figure 8.

(A-G) C57BL/6 mice were immunized subcutaneously with 1×10^5 irradiated B16 cells. 10 days post immunization, mice were re injected with 1×10^6 live Mock or PTEN α expressing *Pten*^{-/-} B16 cells through subcutaneously injection. TILs were isolated when the tumor volume of Mock-immunized group reaching 200 mm³, and CD45⁺ cells were sorted, using for 10x scRNA-seq.

(A) Cells were identified as 12 clusters utilizing graph-based clustering.

(B) The differentially expressed genes in the clusters were used for drawing the heat map. Also shown in **Figure S2d** in the manuscript.

(C) Proportion of the 12 clusters were shown. Also shown in **Figure S2f** in the manuscript.

(D) Cd8T cells were further analyzed and identified as 4 clusters utilizing graph-based clustering. Also shown in **Figure 2d** in the manuscript.

(E) Heat map analysis of the differentially expressed genes in the 4 clusters of Cd8T cells. Also shown in **Figure S3a** in the manuscript.

(F and G) Umap and proportion of the 4 clusters of Cd8T cells were shown. Also shown in **Figure 2f and 2g** in the manuscript.

In addition, Ptena expression is associated with higher tumor size and reduced CD8 T cell infiltration, therefore it impacts antigen to T cell ratio, and higher antigen dose can lead to higher exhaustion of ag-specific T cells. Therefore, the higher levels of exhaustion might be a direct consequence of the reduced amounts of CD8 T cells. Can the authors clarify this?

[Response] To determine whether the reduced amounts of tumor-infiltrating lymphocytes (TILs) are the cause or consequence of T cell exhaustion in PTEN α -expressing tumor, we used the adoptive T-cell transfer assay. Mock or PTEN α -expressing B16-*Pten*^{-/-} cells were infected with LCMV-C113, and subcutaneously injected into nude mice. 10 days later, the CD8⁺ T cells were harvested from spleen in mice infected with LCMV-C113 on day-7 post infection, and subsequently transferred into the tumor-bearing nude mice. Similar procedures were repeated on day-14 and day-18 in B16 tumor-bearing mice (**Response Fig. 9A**). As shown in **Response Figure 9B**, viral infection hardly influenced tumor growth of

Mock or PTEN α -expressing B16 cells in nude mice, and ectopic expression of PTEN α elicited no effects on cell proliferation, which is consistent to the data in C57BL/6 mice (**Response Fig. 9B**). Moreover, the transferred CD8⁺ T cells exhibited significantly antitumor effects on Mock rather than PTEN α -expressing B16 cells in nude mice (**Response Fig. 9B**). Notably, through analysis of CD8⁺ T cells infiltrating in spleen or tumors, we found that the amount of adoptive transferred CD8⁺ T cells were identical between the nude mice with PTEN α -expressing B16 cells and those with Mock B16 cells (**Response Fig. 9C**). Taken together, our data demonstrate that presence of PTEN α promotes T cell dysfunction, subsequently limiting the amount of CD8⁺ T cells.

Accordingly, the main text information has been updated in the section of “Results” (Page 8, Line 16-22, and Page 9, 1-8).

Response Figure 9

Response Figure 9.

(A-C) Mock or PTEN α expressing B16-*Pten*^{-/-} cells were infected with 1 MOI LCMV-C113, and subcutaneously injected into the right flank of Balb/c nude mice. CD8⁺ T cells were harvested from spleens of C57BL/6 mice *i.v.* infected with 1×10^6 PFU LCMV-C113 on day 7 post infection. 10, 14 and 18 days after tumor inoculation, the CD8⁺ T cells were transferred into the tumor bearing nude mice through intravenous injection.

(A) An illustration of mice treatment. Also shown in **Figure 2h** in the manuscript.

(B) Tumor volumes were monitored overtime (Mock, n = 6 mice; PTEN α , n = 7 mice, mean \pm s.e.m, **P* < 0.05). Statistical significance between CD8⁺ T cells transferred groups were shown. Also shown in **Figure 2i** in the manuscript.

(C) On day 24 post tumor inoculation, immune cells were isolated from spleens and

tumors of the nude mice, and subjected to flow cytometry analysis. Percentages of CD8⁺ T cells in CD45⁺ immune cells were shown (Mock, n = 3 mice; PTEN α , n = 4 mice, mean \pm SD, ns, not significant). Also shown in **Figure S3g** in the manuscript. Statistical significance was assessed by two-tailed unpaired Student's t test (B and C). Data are pooled from two (B) independent experiments.

Similarly, for LCMV chronic infection, Ptena KO mice present lower viral loads, which could explain the lower CD8 T cell exhaustion due to reduced antigen dose (e.g. <https://www.ncbi.nlm.nih.gov/pmc/articles/PMC4995073/>).

In other words, is there any evidence to claim that Ptena induces T cell exhaustion, independently from the reduction of antigen dose? Otherwise, it might be useful to discuss this.

[Response] No, we do not have any evidence to claim that PTEN α induces T cell exhaustion in an antigen dose-independent manner. Conversely, our data support the notion that PTEN α impairs T cell-mediated eradication of tumor cell, which in turn triggers persistent antigenic stimulation, eventually resulting in T cell exhaustion. As suggested, we will discuss the role of PTEN α in modulation of T cell dysfunction in revised manuscript.

Accordingly, the main text information has been updated in the section of "Discussion" (Page 20, Line 18-22, and Page 21, 1-7).

The scRNA-seq experiment is insufficiently described and the analysis is not fully convincing. i) The current interpretation that increased expression of activation/exhaustion markers (Pdc1, lag3, Havcr2, Hif1a, etc) and reduced expression of memory genes (Tcf7, Sell, IL7r etc) is linked to immune escape is not very convincing. Most tumor-specific CD8 TILs express Pdc1 and do not express naive/central memory genes Sell/IL7r, therefore, most likely, these differences are largely due to comparing tumor-specific vs non-specific T cells. Consider that a substantial part of CD8 TILs in B16 tumors might not be tumor-specific, and even include naive T cells (e.g. <https://www.ncbi.nlm.nih.gov/pmc/articles/PMC2916130/>, <https://pubmed.ncbi.nlm.nih.gov/32313720/>)

To help interpreting these data, additional information should be shown, including the full table of differential gene expression analysis of CD8 TILs in mock vs Ptena is missing and should be provided.

In particular it would be key to see expression of cytotoxicity genes such as Tnfa, Ifng and Gzmb.

[Response] As suggested, we performed unsupervised clustering and identified the fourth cluster of CD8⁺ T cells, CD8T-Ctla4, expressed high levels of exhaustion markers *Ctla4*, *Pdc1*, *Lag3* and *Hif1a*, thus representing exhausted CD8⁺ T cells. As expected, the percentage of C4_CD8T-Ctla4 was remarkably increased in PTEN α -expressing tumors as compared with Mock tumors (**Response Fig. 10A-C**).

Our data thus demonstrate that presence of PTEN α promotes tumor-reactive T cell dysfunction. Interestingly, we also noticed that the genes encoding effector cytokines including *Ifng*, *Gzmb* and *Gzmk* were also increased in C4_CD8T-Ctla4 (**Response Fig. 10D**), which is consistent to the published scRNAseq data from tumor-infiltrating CD8⁺ T cells in human liver cancers (PMID: 28622514) (**Response Fig. 10E**).

In order to further assess the effector function of CD8⁺ tumor-reactive T cells, we isolated CD8⁺ T cells from Mock or PTEN α -expressing tumors. As shown in **Response Figure 11A**, the expression of PD-1 and TIM-3 on the surface of CD8⁺ T cells were upregulated in PTEN α -expressing tumors as relative to those in control tumors. Furthermore, we also measured the production of IFN γ in the tumor-reactive CD8⁺ T cells in the treatment of PMA/ionomycin. As shown in **Response Figure 11B**, the production of IFN γ were weakened in CD8⁺ T cells from PTEN α -expressing tumors as relative to those in control tumors. Given the less amounts of cluster 1 and cluster 2 of CD8⁺ T cells in PTEN α -expressing tumors, our data thus indicate that cluster 1 and cluster 2 of CD8⁺ T cells are the main source of effector cytokines upon exposure to antigen, despite higher level of genes encoding effector cytokines in exhausted T cell subset.

Accordingly, the main text information has been updated in the section of “Results” (Page 8, Line 1-15).

Response Figure 10

Response Figure 10.

(A-D) C57BL/6 mice were immunized subcutaneously with 1×10^5 irradiated B16 cells. 10 days post immunization, mice were reinjected with 1×10^6 live Mock or PTEN α transfected *Pten*^{-/-} B16 cells through subcutaneously injection. TILs were isolated when the tumor volume of Mock-immunized group reaching 200 mm³, and CD45⁺ cells were sorted, using for 10x scRNA-seq.

(A and B) Cd8T cells were analyzed and identified as 4 clusters utilizing graph-based clustering. Also shown in **Figure 2d and 2g** in the manuscript.

(C) Proportion of the 4 clusters were shown. Also shown in **Figure 2f** in the manuscript.

(D) Violin plot of differentiated genes in the 4 clusters of Cd8T cells. The color of each violin is the average expression value of a given gene. Also shown in **Figure 2e** in the manuscript.

(E) Gene expressions in infiltrating T cells in liver cancer from tumor patients downloaded from <http://hcc.cancer-pku.cn> (PMID: 28622514). C4_CD8-LAYN expressed high levels of *Ctla4*, *Pdcd1* and *Tox* was clustered as exhausted T cells.

Response Figure 11

A

B

Response Figure 11.

(A and B) C57BL/6 mice were immunized subcutaneously with 1×10^5 irradiated B16 cells. 10 days post immunization, mice were reinjected with 1×10^6 live Mock or PTEN α transfected *Pten*^{-/-} B16 cells through subcutaneously injection. The tumor-infiltrating lymphocytes (TILs) were isolated when the tumor volume of Mock-immunized group reaching 200 mm³, and the cells were subjected to flow cytometry analysis.

(A) The mean fluorescence intensity of PD-1 and TIM-3 in CD8⁺ T cells were assessed by flow cytometry analysis (n = 4 mice, mean \pm SD, *P < 0.05). Also shown in **Figure S3c-e** in the manuscript.

(B) The lymphocytes were treated with PMA (100 ng/ml) and ionomycin (500 ng/ml) for 5 hours, and production of IFN γ in CD8⁺ T cells was determined by flow cytometry analysis (n = 4 mice, mean \pm SD, *P < 0.05). Also shown in **Figure S3f** in the manuscript.

Statistical significance was assessed by two-tailed unpaired Student's t test (A and B).

ii) Figure S2D does not allow verifying the cluster annotations. Which genes are expressed in the CD4 or CD8 T cell clusters? For instance, Cd4 and Gzmb seem to be expressed in both CD4 and CD8 clusters. "Treg genes" module seem to be equally expressed in CD4 and CD8 clusters, or even more highly expressed in CD8.

It would be important to show key cell type marker genes expression individually in each cluster, both as boxplot/violin plots and on the tSNE.

[Response] As suggested, we have used umap to examine the canonical marker genes related with major cell populations, including T cells, B cells and other diverse myeloid-lineage cells to further confirm the stability and accuracy of our data.

Response Figure 12

Response Figure 12.

(A) Umap projection of 12 clusters of cells. The color of Each cell based on the relative normalized expression of indicated genes.

(B) Umap projection of 4 clusters of Cd8T cells. The color of Each cell based on the

relative normalized expression of indicated genes.

iii) There is insufficient description of the scRNA-seq experiments, answers to all these questions should be provided: which samples were used for scRNA-seq? How were these samples obtained and processed? From how many animals, at which timepoints, which average tumor sizes? Where the samples grouped into separate sequencing batches? How were cells sorted (FACS panel) ? how single-cell RNA-seq libraries were prepared? (protocol?) description of the scRNA-seq experiments

[Response] We thank the reviewer for this comment. We have revised our manuscript and described the steps for sample collection and protocols for scRNAseq experiments in detail in the section of materials and methods in light of reviewer's suggestion.

Accordingly, the main text information has been updated in the section of "Materials and Methods" (Page 33, Line 10-22, and Page 34, and Page 35, Line 1-4).

iv) Where are these data available (both raw and processed)? These should be made publicly available in a repository such as NCBI GEO

[Response] The scRNAseq data have been deposited in the GEO database with the accession code GSE178258. Reviewer token for the scRNAseq data is klijaymibdmrnuh.

v) Please indicate how the expression values shown in eg Fig 2G were calculated. In addition, data integration/batch-effect correction is a major transformation that was applied to these data and it might be important to mention it in the main text.

[Response] We agree with the reviewer that we should study the alteration of T cell subsets rather than the overall tumor-infiltrating T cells to avoid the artifacts and thus mix different cell type population. We thus substitute the clustering analysis for the dot plot analysis. In addition, we have mentioned the data integration/batch-effect correction in the main text in our revised manuscript.

The main text information has been updated in the section of "Results" (Page 7, Line 13-14).

vi) In the methods a UMAP representation is described, but the plot is not shown

[Response] We regret for this inaccurate statement. We have deposited both uMAP and t-SNE representations in main text and supplementary Figures in our revised manuscript.

vii) In Figure 2G, it would be also important to show 'control genes' including Cd2, Cd3, Cd8a, Cd8b, Cd4, Foxp3, to demonstrate that these two groups are equally well represented in terms of CD8 T cells. This is particularly important because batch-effect correction was*

applied to the data, which can introduce artifacts and mix different cell type populations.

[Response] As suggested, we have shown the control genes in the violin plot. As shown in **Response Figure 13**, all Cd8T cell clusters express high levels of *Cd2*, *Cd3d*, *Cd3e*, *Cd8a* and *Cd8b1*, while hardly express *Cd4* and *Foxp3*.

Response Figure 13

Response Figure 13.

Violin plot of differentiated genes in the 4 clusters of Cd8T cells. The color of each violin is the average expression value of a given gene. Also shown in **Figure 2e** in the manuscript.

Can the authors confirm that in vitro T cell killing of B16 cells is MHC-I mediated?

[Response] To confirm that T cell killing of B16 cells is in MHC-I-dependent manner, we thus used LCMV to infect Balb/c (H2-k^d) or C57BL/6 (H2-k^b) mice. On the 7-day post-infection, the virus-specific CD8⁺ T cells were isolated from Balb/c (H2-k^d) or C57BL/6 (H2-k^b) mice, separately. These two types of virus-specific CD8⁺ T cells subsequently co-cultured with viral-peptide pulsed or untreated *Pten*^{-/-} B16 cells (H2-k^b) in presence or absence of PTEN α . As shown in **Response Figure 14**, only the MHC alleles-matched CD8⁺ T cells (from H2-k^b mice) killed viral-peptide pulsed target cells but not untreated cells (H2-k^b). Conversely, the virus-specific CD8⁺ T cells were isolated from Balb/c (H2-k^d) elicited little effects on the viability of target cells (H2-k^b) (**Response Fig. 14**). Our data thus demonstrate that MHC-restriction is required for T cell killing of B16 cells.

Response Figure 14

Response Figure 14.

CD8⁺ T cells were isolated from the spleens of C57BL/6 or Balb/c mice untreated or *i.v.* infected with 1×10^6 LCMV-Cl13 for 7 days, and used as the effector cells for *in vitro* cytotoxicity assay. Mock or PTEN α expressing B16-*Pten*^{-/-} cells were pulsed with GP₃₃₋₄₁ peptide and used as target cells. Effector cells and target cells were incubated at indicated ratios. 20 hours later, the cells were harvested, and directly stained with PI (Invitrogen, following manufactory's instruction), followed by flow cytometry analysis. Percentages of PI⁺ cells in CD8⁻ cells were used for statistical analysis (n = 3 cell cultures, mean \pm SD, ns, not significant, *****P* < 0.0001). Statistical significance was assessed by two-tailed unpaired Student's t test.

While the evidence for increased resistance to oxidative stress by Ptena expression in cancer cells seems convincing, and contribute to increased resistance to CTL killing, the fact that Ptena-expressing tumors are associated with reduced immune cell infiltration (CD45+) and T cell infiltration in particular, both in human samples (TCGA data) and in murine models, is not directly discussed.

[Response] We agree with the reviewer, and we have discussed the relationship between the stimulatory role of PTEN α in cancer immune resistance and the reduced amounts of tumor-infiltrating lymphocytes in PTEN α -expressing tumors in the section of discussion in our revised manuscript.

Accordingly, the main text information has been updated in the section of "Discussion" (Page 20, Line 18-22, and Page 21, 1-7).

Reduction of immunogenic cell death in Ptna-expressing tumors might explain the reduced T cell infiltration in vivo. Is this the authors' model? What evidence is supporting that? For instance, can they show that the increased growth of tumors treated with ferroptosis inhibitor NP-VE (i.e. where Ptena expression is irrelevant, Fig 5 F) is CD8 T cell dependent? (due to higher CD8 T cell infiltration levels?)

[Response] Yes, it is. Our proposed model is that reduction of immunogenic cell death (ICD) caused by PTEN α presence triggers persistent antigenic stimulation, consequently promoting exhausted T cell formation. Moreover, PTEN α restricts the release of danger-associated molecular patterns (DAMPs) through limitation of immunogenic cell death, which in turn blocks the recruitment of T cells in PTEN α -expressing tumor.

As suggested, we have subcutaneously injected Mock or PTEN α -expressing B16-*Pten*^{-/-} cells in the presence of ferroptosis inhibitor NP-VE. As shown in **Response Figure 15A**, injection of NP-VE restricted the amount of CD45⁺ immune cells infiltrating in tumors as compared with untreated Mock tumors. Moreover, presence of PTEN α exhibited little effects on the recruitment of immune cells in tumors treated with NP-VE (**Response Fig. 15A**), suggesting that blockade of ICD is essential for the immunosuppressive effects of PTEN α . Subsequent flow cytometry analysis further revealed that NP-VE treatment or PTEN α presence restricted both T and B cell recruitment in tumor (**Response Fig. 15B and 15C**). In accordance with the suppressive effects of NP-VE treatment on T cell recruitment, increased expression of PD1 and reduced production of IFN γ was also detected in CD8⁺ T cells isolated from mock tumors treated with NP-VE (**Response Fig. 15D and 15E**). Taken together, our data thus demonstrate that reduction of immunogenic cell death is essential for PTEN α in both promotion of T cell dysfunction and suppression of immune cell recruitment.

Accordingly, the main text information has been updated in the section of “Results” (Page 15, Line 10-21).

Response Figure 15

Response Figure 15.

(A-E) C57BL/6 mice were immunized subcutaneously with 1×10^5 irradiated B16 cells. 10 days post immunization, mice were re injected with 1×10^6 live Mock or PTEN α transfected *Pten*^{-/-} B16 cells through subcutaneously injection in the presence or absence of 50 μ l of 5 M NP-VE. Tumors were collected when the Mock-UT group reached 200 mm³, and tumor infiltrating immune cells were isolated.

(A) The cell counts of CD45⁺ cells were used for statistical analysis (n = 3 mice, mean \pm SD, ns, not significant, * $P < 0.05$). Also shown in Figure S9a in the manuscript.

(B and C) Cell counts of B220⁺ cells and CD4⁺ and CD8⁺ T cells were assessed by flow cytometry analysis (n = 3 mice, mean \pm SD, ns, not significant, ** $P < 0.01$, *** $P < 0.001$). Also shown in Figure S9b and S9c in the manuscript.

(D) The mean fluorescence intensity of PD-1 in CD8⁺ T cells were determined by flow cytometry analysis (n = 3 mice, mean ± SD, ns, not significant, **P < 0.01). Also shown in Figure S9d in the manuscript.

(E) The cells were treated with PMA and ionomycin for 5 hours, and production of IFN γ in CD8⁺ T cells were assessed by flow cytometry analysis (n = 3 mice, mean ± SD, ns, not significant, **P < 0.01). Also shown in Figure S9e in the manuscript. Statistical significance was assessed by two-tailed unpaired Student's t test (A,C-E).

Minor comments

In Figure 1 C, for the GSEA it is recommended to use the full ranked gene list for measuring enrichment of a gene set instead of defining a threshold (here the top phosphate-inactive and the top stop-gain mutation correlated genes were chosen). See for example https://software.broadinstitute.org/cancer/software/gsea/wiki/index.php/FAQ#What_is_the_difference_between_GSEA_and_an_overlap_statistic_.28hypergeometric.29_analysis_to_ol.3F). Otherwise, please justify the use of a subsetted ranked gene list for GSEA. Also in the heatmap, "SLANF7" is not a valid gene name.

[Response] As suggested, we used the full ranked genes to measure enrichment of signaling pathways. Due to space limitation, we only listed top 20 genes in Response Figure 16. In addition, we regret this error, "SLANF7" has been changed to "SLAMF7".

Response Figure 16

Response Figure 16.

UCEC patients with matched diagnosis age (40-60 yrs), aneuploidy score (<3) and neoplasm histologic grade (G3) were screened. RNA-seq data of these patients were acquired from the TCGA database, and subjected to GSEA. ES is abbreviation of Enrichment Score, and NES represents Normalized Enrichment Score. Also shown in Figure 1c in the manuscript.

The following sentences should be corrected:

"Durable benefit, however, this therapy is limited to a minority of patients with cancer, and the clinical efficacy is often compromised along with tumor development"

"which is analogue to phosphatase-inactive mutations do"

"We next sought to determine whether T cell polarization is depends on T cell-intrinsic PTENa."

[Response] We apologize for the inaccurate statement. We have corrected the above sentences in our revised manuscript and highlighted the changes.

The main text information has been updated in the section of "Introduction" (Page 3, Line 16-18) and "Results" (Page 5, Line 8-9, and Page 11, Line 17-18).

Reviewers' Comments:

Reviewer #1:

Remarks to the Author:

The authors have appropriately responded to all of my comments.

Reviewer #4:

Remarks to the Author:

In their revised manuscript, the authors have addressed most of my comments. In particular, they improved the quality of the scRNA-seq data and their analysis, which is now much clearer.

Although the authors did not directly address my comment regarding tumor-specificity of T cells, they performed an adoptive T cell transfer experiment in nude mice showing that PTEN α -expression makes the tumor more resistant to T cell killing, supporting their model.

I believe that the authors have presented enough evidence, but that some changes are still required in the manuscript:

1) do not claim "tumor-reactive" when that was not measured:

"Our data thus demonstrate that presence of PTEN α promotes tumor-reactive T cell dysfunction." Tumor-reactivity was not assessed

"Furthermore, we also measured the production of IFN γ in the tumor-reactive CD8 $^+$ T cells in the treatment of PMA/ionomycin. As shown in supplementary Figure 3f, the production of IFN γ were weakened in CD8 $^+$ T cells from PTEN α -expressing tumors as relative to those in control tumors"

However it seems the authors did not evaluate tumor-reactive CD8 T cells.

Where are the PMA/ionomycin assays described?

How were tumor-reactive CD8 T cells isolated?

2) The authors claim that cluster 1 and cluster 2 of CD8 $^+$ T cells are the main population exerting tumor control:

"As shown in Response Figure 11B, the production of IFN γ were weakened in CD8 $^+$ T cells from PTEN α -expressing tumors as relative to those in control tumors. Given the less amounts of cluster 1 and cluster 2 of CD8 $^+$ T cells in PTEN α -expressing tumors, our data thus indicate that cluster 1 and cluster 2 of CD8 $^+$ T cells are the main source of effector cytokines upon exposure to antigen"

This is not supported by data, and on the contrary, it seems unlikely as these cells (cluster 1 and 2) do not express cytotoxic molecules in vivo (Ifng, Gzmb) and are most likely not tumor specific (Pcd1 neg, Lag3 neg, Sell high, Ccr7 high, etc.). The claim that these cells are exerting tumor control is not convincing.

Instead, the scRNA-seq data showed that the exhausted CD8 T cells seem to be polyfunctional and are more likely tumor-specific. The authors observe that "Interestingly, we also noticed that the genes encoding effector cytokines including Ifng, Gzmb and Gzmk were also increased in C4_CD8T-Ctla4 (Response Fig. 10D)".

The exhausted polyfunctional population might be therefore exerting tumor control. Even if their relative proportion among CD8 is lower in PTEN α , they seem to be in higher numbers with respect to tumor mass, and this is more relevant than their relative frequency.

3) I did not find in Methods the description of the new adoptive T cell transfer experiment, including information on how they infected B16 cells with LCMV."

Dear Dr. Danelli,

Thank you for your decision on our manuscript entitled “PTEN α functions as an immune suppressor and promotes immune resistance in *PTEN*-mutant cancer” (Manuscript ID: NCOMMS-20-38788B) that we submitted to *Nature Communications*. We have now made a revision of the paper according to the reviewer’s suggestions and the editor’s requests. The following is our point-by-point response to the reviewer’s comments.

A. Point-by-point responses to the comments by Reviewer #1

Reviewer #1 (Remarks to the Author):

The authors have appropriately responded to all of my comments.

[Response] We appreciate the reviewer’s appreciation of our study.

B. Point-by-point responses to the comments by Reviewer #4

Reviewer #4 (Remarks to the Author):

In their revised manuscript, the authors have addressed most of my comments. In particular, they improved the quality of the scRNA-seq data and their analysis, which is now much clearer.

Although the authors did not directly address my comment regarding tumor-specificity of T cells, they performed an adoptive T cell transfer experiment in nude mice showing that PTEN α -expression makes the tumor more resistant to T cell killing, supporting their model.

I believe that the authors have presented enough evidence, but that some changes are still required in the manuscript:

[Response] We are grateful for the reviewer’s instructions to improve our manuscript. Now we have carefully revised our manuscript, and have addressed each of the comments as follows.

1) do not claim “tumor-reactive” when that was not measured:

"Our data thus demonstrate that presence of PTEN α promotes tumor-reactive T cell dysfunction."

Tumor-reactivity was not assessed

"Furthermore, we also measured the production of IFN γ in the tumor-reactive CD8+ T

cells in the treatment of PMA/ionomycin. As shown in supplementary Figure 3f, the production of IFN γ were weakened in CD8⁺ T cells from PTEN α -expressing tumors as relative to those in control tumors"

However it seems the authors did not evaluate tumor-reactive CD8 T cells.

[Response] As suggested, we have replaced the statement of “tumor-reactive” with “tumor-infiltrating” to describe T cells that we analyzed.

Accordingly, the main text information has been updated in the section of “Results” (Page 8, Line 11).

Where are the PMA/ionomycin assays described?

How were tumor-reactive CD8 T cells isolated?

[Response] The specific protocols including PMA/ionomycin assay and isolation of tumor-infiltrating CD8⁺ T cells have been described in the section of methods in our revised manuscript (Page 30, Line 20-22, Page 31, Line 1-3 and Page 32, Line 14-17).

2) The authors claim that cluster 1 and cluster 2 of CD8⁺ T cells are the main population exerting tumor control:

"As shown in Response Figure 11B, the production of IFN γ were weakened in CD8⁺ T cells from PTEN α -expressing tumors as relative to those in control tumors. Given the less amounts of cluster 1 and cluster 2 of CD8⁺ T cells in PTEN α -expressing tumors, our data thus indicate that cluster 1 and cluster 2 of CD8⁺ T cells are the main source of effector cytokines upon exposure to antigen"

This is not supported by data, and on the contrary, it seems unlikely as these cells (cluster 1 and 2) do not express cytotoxic molecules in vivo (Ifng, Gzmb) and are most likely not tumor specific (Pdcd1 neg, Lag3 neg, Sell high, Ccr7 high, etc.). The claim that these cells are exerting tumor control is not convincing.

Instead, the scRNA-seq data showed that the exhausted CD8 T cells seem to be polyfunctional and are more likely tumor-specific. The authors observe that “Interestingly, we also noticed that the genes encoding effector cytokines including Ifng, Gzmb and Gzmk were also increased in C4_CD8T-Ctla4 (Response Fig. 10D)”.

The exhausted polyfunctional population might be therefore exerting tumor control. Even if their relative proportion among CD8 is lower in PTEN α , they seem to be in higher numbers with respect to tumor mass, and this is more relevant than their relative frequency.

[Response] We thank the reviewer for this question. Compared with the cluster 3 and 4 T cells, genes encoding both inhibitory receptors and effector cytokines were decreased in the cluster 1 and 2 T cells. Conversely, genes associated with T cell

stemness including *Sell/CD62l*, *Ccr7* and *Tcf7* were selectively upregulated in these cells (Cluster 1 and 2). According to the previous studies (PMID: 33230330, 19525962), the stem cell-like memory T cells (Tscm) ($CD44^{low}CD62L^{high}CD127^{high}CCR7^{+}TCF7^{high}$) that expressed lower level of effector cytokines exhibited enhanced anti-tumor capacities compared with other T cell subsets. Based on the gene expression profile, we found that the cluster 1 and 2 T cells possessed similar characteristic of Tscm (PMID: 28060797). It is thus conceivable that both reduced self-renewing multipotent $CD8^{+}$ Tscm and increased percentages of exhausted T cells cooperatively contribute to immune escape of PTEN α -expressing tumors.

Indeed, we agree with the reviewer's comments that the exhausted T cells are polyfunctional. Actually, some recent publications have revealed that exhausted T cells can be defined as four subsets linked in a hierarchical developmental pathway. The "progenitor exhausted" cells are still polyfunctional, and an intermediate subset even re-engaged some effector biology (PMID: 30778252, 32396847). Therefore, it is also possible that existence of PTEN α promotes the development of the exhausted T cells, leading to different proportions of the exhausted T cells, thereby affecting tumor clearance. We will further analyze the potential of each cluster of $CD8^{+}$ T cells in our future work.

3) I did not find in Methods the description of the new adoptive T cell transfer experiment, including information on how they infected B16 cells with LCMV

[Response] As suggested, we have described the specific steps of adoptive T cell transfer experiment in the section of methods in the revised manuscript (Page 26, Line 3-8).